# SeaS: Few-shot Industrial Anomaly Image Generation with Separation and Sharing Fine-tuning

## Abstract

Current segmentation methods typically require many training images and precise masks, while insufficient anomaly images hinder their application in industrial scenarios. To address such an issue, we explore producing diverse anomalies and accurate pixel-wise annotations. By observing the real production lines, we find that anomalies vary randomly in shape and appearance, whereas products hold globally consistent patterns with slight local variations. Such a characteristic inspires us to develop a Separation and Sharing Fine-tuning (SeaS) approach using only a few abnormal and some normal images. Firstly, we propose the Unbalanced Abnormal (UA) Text Prompt tailored to industrial anomaly generation, consisting of one product token and several anomaly tokens. Then, for anomaly images, we propose a Decoupled Anomaly Alignment (DA) loss to bind the attributes of the anomalies to different anomaly tokens. Re-blending such attributes may produce never-seen anomalies, achieving a high diversity of anomalies. For normal images, we propose a Normal-image Alignment (NA) loss to learn the products' key features that are used to synthesize products with both global consistency and local variations. The two training processes are separated but conducted on a shared U-Net. Finally, SeaS produces high-fidelity annotations for the generated anomalies by fusing discriminative features of U-Net and high-resolution VAE features. The extensive evaluations on the challenging MVTec AD and MVTec 3D AD dataset (RGB images) demonstrate the effectiveness of our approach. For anomaly image generation, on MVTec AD dataset, we achieve 1.88 on IS and 0.34 on IC-LPIPS, while on the MVTec 3D AD dataset, we obtain 1.95 on IS and 0.30 on IC-LPIPS. For the downstream task, by using our generated anomaly image-mask pairs, three common segmentation methods achieve an average 11.17% improvement on IoU on MVTec AD dataset, and a 15.49% enhancement in IoU on the MVTec 3D AD dataset. The source code will be released publicly available.

## 1 Introduction

Existing segmentation approaches require a large number of anomaly images with mask annotations, while the scarcity of anomaly images obstructs their application in industrial scenarios. To solve this problem, generative methods for industrial scenarios have emerged to expand the training set of segmentation models.

To the best of our knowledge, generation approaches (Zavrtanik et al., 2021)in industrial scenarios can be broadly classified into two categories: **Anomaly Generation (AG)** and **Anomaly Image Generation (AIG)**. AG methods (Li et al., 2021; Zavrtanik et al., 2021; Schlüter et al., 2022; Hu et al., 2024)generate anomalies only and merge them into the real normal images using different strategies, e.g., CutPaste (Li et al., 2021)pastes a cropped normal region to normal images, which simulates anomalies by misalignment. AnomalyDiffusion (Hu et al., 2024)generates anomalies by a diffusion model, and edits anomalies onto the normal images guided by the anomaly masks, as shown in Fig. 1(a). However, AG methods require anomaly masks as inputs, which easily suffer from low fidelity and consistency in generation if these masks are unreasonably positioned. In contrast to AG, as shown in Fig. 1(b), AIG approaches take a step further, generating anomalies and the industrial products that they lie in simultaneously. Therefore, AIG faces greater challenges

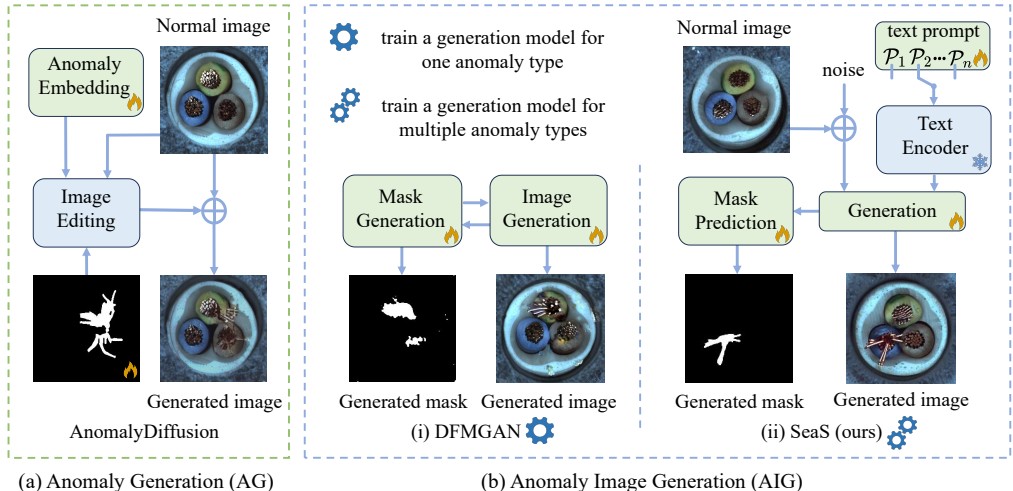

Figure 1: **(a) Anomaly Generation (AG)** method only generates anomaly and edits it onto normal images guided by the anomaly masks. **(b) Anomaly Image Generation (AIG)** methods generate anomalies and the industrial products that they lie in simultaneously. (i) DFMGAN trains a generation model for each anomaly type. (ii) SeaS trains a shared generation model for multiple anomaly types.

due to the requirement of high-fidelity of both anomalies and products. Previously, AIG methods (Duan et al., 2023; Zhang et al., 2021a; Niu et al., 2020) are mainly based on Generative Adversarial Network (GAN). However, they are limited by insufficient generalization of GAN, lacking specific control over products/anomalies, and inaccurate masks.

In real industrial manufacturing, the products in an individual production line are almost similar to each other, while the anomalies are unforeseeable in shape and appearance. **Such an observation reveals a differentiated characteristic, i.e., the products satisfy global consistency with minor variations in local details, while the anomalies hold randomness,** which is rarely discussed in existing AG or AIG approaches. Motivated by such a characteristic, we propose a **Se**pa**ration and S**haring Fine-tuning method, short by SeaS, a controllable AIG method based on Stable Diffusion (Rombach et al., 2022). The key idea is to employ Unbalanced Abnormal (UA) Text Prompts containing a set of tokens that characterize products and anomalies separately, so that the anomaly tokens align with the anomaly semantics for diverse generations, and a product token expresses a globally consistent product surface. Specifically, to learn highly-diverse anomalies, we first propose a Decoupled Anomaly Alignment (DA) loss to bind the attributes of the anomalies to different anomaly tokens. Recombining the decoupled attributes may produce anomalies that have never been seen in the training dataset, therefore increasing the diversity of the generated anomalies. Secondly, to learn globally-consistent patterns from products, we propose the Normal-image Alignment (NA) loss. It enables the network to learn the key features of the product from normal images and fine-tune a learnable embedding. Such an embedding ensures the preservation of global consistency amidst local detail variations. Thirdly, **according to the experimental analysis, we find that existing methods leverage the low-resolution features to predict the mask, which may introduce a large amount of boundary uncertainty.** Thus, we propose a Refined Mask Prediction (RMP) branch to produce pixel-wise anomaly annotations for other downstream tasks. It combines the discriminative U-Net features and high-resolution VAE features to generate accurate and crisp masks in a progressive way. Extensive experiments on AIG and downstream anomaly segmentation tasks show that SeaS outperforms the existing industrial anomaly generation methods. On MVTec AD dataset, our model achieves **1.88** on IS metric and **0.34** on IC-LPIPS. Furthermore, training on the image-mask pairs generated by SeaS, the downstream segmentation models achieve improvements of average **+5.53%** AP and **+11.17%** IoU. On MVTec AD 3D dataset (RGB images), our method attains **1.95** on IS metric and **0.30** on IC-LPIPS. Using the image-mask pairs generated by SeaS to train the downstream segmentation models, we exhibit average improvements of **+12.13%** AP and **+15.49%** IoU.

In summary, the key contribution of our approach lies in:

- We reveal different characteristics of products and anomalies, which motivates us to propose SeaS, a novel AIG method. It independently learns products and anomalies on a shared U-Net and ensures the randomness of anomalies and global consistency of products.

- We propose a Refined Mask Prediction branch to produce accurate and crisp pixel-wise annotations for generated anomalies, which combines the advantages of the discriminative U-Net features and the high-resolution VAE decoder features.

- Extensive experiments on anomaly image generation and downstream anomaly segmentation tasks show that SeaS outperforms the existing industrial anomaly generation methods.

## 2 RELATED WORK

**Anomaly Image Generation.** Early non-generative methods (DeVries & Taylor, 2017; Li et al., 2021; Zavrtanik et al., 2021) use data augmentation techniques to create anomaly images. Data augmentation techniques lack consistency in anomaly images, resulting in low fidelity. AG methods (Li et al., 2021; Zavrtanik et al., 2021; Schlüter et al., 2022; Hu et al., 2024; Gui et al., 2024) only generate anomalies and merge them into the real normal images. NSA (Schlüter et al., 2022) uses Poisson Image Editing (Pérez et al., 2003) to facilitate the fusion of the cropped normal region. However, AG methods require anomaly masks as inputs. If these masks are positioned in an unreasonable manner, the generated images will have low fidelity and consistency. Previous AIG methods (Duan et al., 2023; Zhang et al., 2021a; Niu et al., 2020) are mainly based on GAN. Defect-GAN (Zhang et al., 2021a) cannot generate masks. The masks produced by DFMGAN (Duan et al., 2023) often do not align accurately with anomalies, limiting their utility in training segmentation models. We propose a controllable AIG model based on Stable Diffusion to generate high-fidelity and diverse anomaly images with accurate masks.

**Fine-tuning Diffusion Models.** Fine-tuning is a potent strategy for enhancing specific capabilities of pre-trained diffusion models (Gal et al., 2022; Zhang et al., 2023b; Brooks et al., 2023). Personalized methods (Ruiz et al., 2023; Gal et al., 2022; Chen et al., 2024) utilize a small set of images to fine-tune the diffusion model, thereby generating images of the same object. Several methods for multi-concept image fine-tuning (Kumari et al., 2023; Xiao et al., 2023; Avrahami et al., 2023; Han et al., 2023; Jin et al., 2024) use cross-attention maps to align embeddings with individual concepts in the image. Nevertheless, they do not consider the diversity requirements between different concepts, which is important for industrial anomaly image generation. Thus, we propose a separation and sharing fine-tuning strategy for the different diversity needs of anomalies and products, which independently learns products and anomalies on a shared U-Net.

**Mask Prediction with Generation Method.** Previous methods on mask prediction for generated images are mainly based on GANs (Zhang et al., 2021b; Li et al., 2022). However, these approaches do not guarantee the generation of accurate masks for exceedingly small datasets. Based on Stable Diffusion (Rombach et al., 2022), some recent methods, i.e., DiffuMask (Wu et al., 2023b), DatasetDM (Wu et al., 2023a) and DatasetDiffusion (Nguyen et al., 2024), produce masks by exploiting the potential of the cross-attention maps. However, due to the low resolution of the cross-attention maps, they are directly interpolated to a higher resolution to match the image size without any auxiliary information, which leads to significant boundary uncertainty. We incorporate the high-resolution features from the VAE decoder as auxiliary information for resolution retrieving, fusing them with the discriminative features of U-Net decoder to generate accurate high-resolution masks.

## 3 METHOD

The training phase of the proposed Separation and Sharing (SeaS) Fine-tuning strategy is shown in Fig. 2. In Sec. 3.1, we introduce the preliminaries of our approach. In Sec. 3.2, we first design an Unbalanced Abnormal Text Prompt, which contains a set of tokens that characterize products and anomalies separately. Subsequently, we propose the Decoupled Anomaly Alignment (DA) loss to bind the anomaly image regions to the anomaly tokens, and leverage Normal-image Alignment (NA) loss to empower the product token to express globally-consistent normal product surface. The two training processes are implemented separately for abnormal and normal images but on a shared U-Net architecture. Then, based on the well-trained U-Net, we design a Refined Mask Prediction

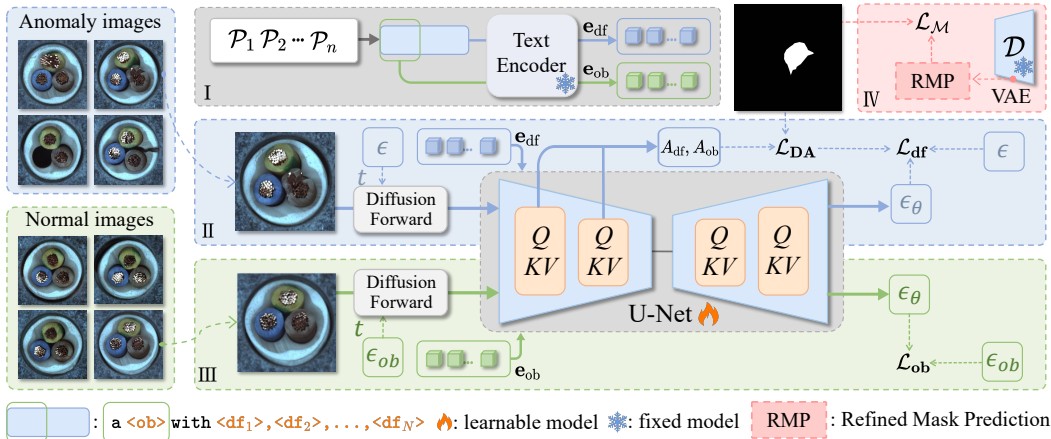

Figure 2: **Overall framework of SeaS.** It consists of four parts: (I) the Unbalanced Abnormal Text Prompt, (II) the Decoupled Anomaly Alignment for aligning the anomaly tokens $<\texttt{df}_n>$ to the anomaly area of abnormal images, (III) the Normal-image Alignment for maintaining authenticity through normal images, and (IV) the Refined Mask Prediction branch for generating accurate masks.

branch to generate accurate masks corresponding to the generated anomaly images in Sec. 3.3. Finally, we detail the generation of the abnormal image-mask pairs in Sec. 3.4.

## 3.1 PRELIMINARIES

**Stable Diffusion.** Given an input image $x_0$, Stable Diffusion (Rombach et al., 2022) firstly transforms $x_0$ into a latent space as $z = \varepsilon(x_0)$, and then adds a randomly sampled noise $\epsilon \sim N(0, \mathbf{I})$ into $z$ as $\hat{z}_t = \alpha_t z + \beta_t \epsilon$, where $t$ is the randomly sampled timestep. Then, the U-Net is employed to predict the noise $\epsilon$. Let $c_\theta(\mathcal{P})$ be the CLIP text encoder that maps conditioning text prompt $\mathcal{P}$ into a conditioning vector $\mathbf{e}$. The training loss of Stable Diffusion can be stated as follows:

$$\mathcal{L}_{\text{SD}} = \mathbb{E}_{z=\varepsilon(x_0), \mathcal{P}, \epsilon \sim N(0,\mathbf{I}), t}\left[||\epsilon - \epsilon_\theta(\hat{z}_t, t, \mathbf{e})||_2^2\right] \tag{1}$$

where $\epsilon_\theta$ is the predicted noise.

**Cross-Attention Map in U-Net.** Aiming to control the generation process, the conditioning mechanism is implemented by calculating cross-attention between the conditioning vector $\mathbf{e} \in \mathbb{R}^{Z \times C_1}$ and image features $\mathbf{v} \in \mathbb{R}^{r \times r \times C_2}$ of the U-Net inner layers (Hertz et al., 2022; Chefer et al., 2023; Xie et al., 2023). The cross-attention map $A^{m,l} \in \mathbb{R}^{r \times r \times Z}$ can be calculated as:

$$A^{m,l} = \text{softmax}(\frac{QK^\top}{\sqrt{d}}), Q = \phi_q(\mathbf{v}), K = \phi_k(\mathbf{e}) \tag{2}$$

where $Q \in \mathbb{R}^{r \times r \times C}$ denotes a query projected by a linear layer $\phi_q$ from $\mathbf{v}$, $r$ is the resolution of the feature map in U-Net, and $l$ is the index of the U-Net inner layer. $K \in \mathbb{R}^{Z \times C}$ denotes a key through another linear layer $\phi_k$ from $\mathbf{e}$, and $Z$ is the number of text embeddings after padding.

## 3.2 SEPARATION AND SHARING FINE-TUNING

**Unbalanced Abnormal Text Prompt.** Through the experimental observation, we found that the typical text prompt, like `a photo of a bottle with defect` (Jeong et al., 2023), or `damaged bottle` (Zhou et al., 2024b), is suboptimal for industrial anomaly generation. The fixed generic semantic words, e.g., `damaged`, `defect`, may fail to align with a few training images that contain specific defect types. Therefore, we design the Unbalanced Abnormal (UA) Text Prompt for each anomaly type of each product, i.e.,

$$\mathcal{P} = \texttt{a <ob> with <df}_1\texttt{>,<df}_2\texttt{>,...,<df}_N\texttt{>}$$

where `<ob>` and `<df`$_n$`>` ($n \in \{1, 2, ..., N\}$) are the tokens of the industrial products (short for Normal Token) and the defects (short for Anomaly Token) respectively. We use a set of $N$ Anomaly Tokens for each anomaly type, with different sets corresponding to different anomaly types. As

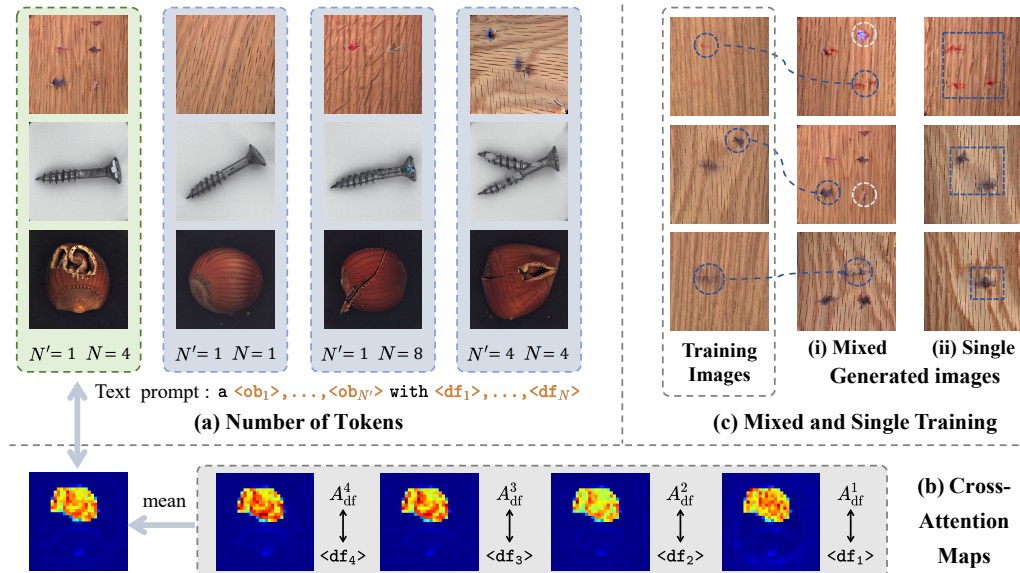

Figure 3: (a) Generated images with the different number of tokens. (b) Cross-attention maps. (c) Examples of diverse generated images.

shown in Fig. 3, in SeaS, we separately employ normal images to train the embedding corresponding to `<ob>`, and abnormal images to train the embeddings corresponding to `<df_n>`. Experimental observations indicate that one `<ob>` is sufficient to express the normal product, while multiple `<df_n>` are necessary for controlling the generation of the anomalies. As shown in Fig. 3(a), when we use the UA prompt $\mathcal{P}$ (the dotted green box in (a)), the cross-attention maps in (b) show that different tokens have different responses in the abnormal regions, which indicates that they focus on different attributes of the anomalies, and performing the average operation on the cross-attention maps produces never-seen anomalies. When we use only one `<df>`, it is difficult to align it to several different anomalies that belong to the same category. Therefore, during inference, if the denoised anomaly feature has a larger distance to `<df>`, it will be assigned a smaller response by the U-Net, which leads to the "anomaly missing" phenomenon, e.g., the generated images in the case of ($N' = 1, N = 1$). In addition, if we utilize a large number of `<df_n>`, we find that each `<df_n>` may focus on some local properties of an anomaly, such a case increases the diversity but may reduce the authenticity of the anomalies, as shown in the case $N' = 1, N = 8$. Similarly, if we use multiple learnable `<ob>`, e.g., $N' = 4, N = 4$, each `<ob>` pays attention to the local character of the product, which may reduce the authenticity of the product.

**Decoupled Anomaly Alignment.** Given a few abnormal images $x_{df}$ and their corresponding masks, we aim to align the anomaly tokens `<df_n>` to the anomaly area of $x_{df}$ by tuning the U-Net and the learnable embedding corresponding to `<df_n>`. Therefore, we propose the Decoupled Anomaly Alignment (DA) loss, i.e.,

$$\mathcal{L}_{\mathbf{DA}} = \sum_{l=1}^{L}(||\frac{1}{N}\sum_{n=1}^{N} A_{df}^{n,l} - M^l||^2 + ||A_{ob}^l \odot M^l||^2) \qquad (3)$$

where $A_{df}^{n,l} \in \mathbb{R}^{r \times r \times 1}$ is the cross-attention map corresponding to the $n$-th anomaly token `<df_n>`, $N$ is the number of anomaly token in $\mathcal{P}$. $L$ is the total number of U-Net layers used in alignment. $M^l$ is the binary mask with $r \times r$ resolution, where the abnormal area is 1 and the background is 0. $A_{ob}^l \in \mathbb{R}^{r \times r \times 1}$ is the cross-attention map corresponding to the normal token `<ob>`, $\odot$ is the element-wise product. DA loss performs the mandatory decoupling of the anomaly and the product. The first term of DA loss is to align the abnormal area to `<df_n>` according to the mask $M^l$. The second term of DA loss reduces the response value of $A_{ob}^l$ in the abnormal area, which prevents `<ob>` from aligning to the abnormal area of $x_{df}$. Further analysis of how the DA loss ensures the diversity of anomalies is provided in Appendix A.2. Therefore, the total loss for the anomaly image $x_{df}$ is:

$$\mathcal{L}_{\mathbf{df}} = \mathcal{L}_{\mathbf{DA}} + ||\epsilon_{df} - \epsilon_\theta(\hat{z}_{df}, t_{df}, \mathbf{e}_{df})||_2^2 \qquad (4)$$

In second term of Eq. 4, we use random noises $\epsilon_{\text{df}}$ and timesteps $t_{\text{df}}$ to perform forward diffusion on abnormal images $x_{\text{df}}$, then obtain the noisy latent $\hat{z}_{\text{df}}$. The conditioning vector $\mathbf{e}_{\text{df}} \in \mathbb{R}^{Z \times C_1}$ is used to guide the U-Net in predicting noise, and then calculate the loss with the noise $\epsilon_{\text{df}}$.

**Normal-image Alignment.** As we discussed, increasing the number of the normal token `<ob>` leads to a higher diversity, while may reduce the authenticity of the generated image and destruct global consistency. However, aligning only one `<ob>` to a few of the training images may suffer from the issue of overfitting. Therefore, we add a Normal-image Alignment (NA) loss to overcome such a dilemma, which is stated as follows,

$$\mathcal{L}_{\mathbf{ob}} = ||\epsilon_{\mathbf{ob}} - \epsilon_\theta(\hat{z}_{\mathbf{ob}}, t_{\mathbf{ob}}, \mathbf{e}_{\mathbf{ob}})||_2^2 \tag{5}$$

Instead of aligning the normal region of $x_{\text{df}}$ to `<ob>`, in calculating the NA loss, we use random noises $\epsilon_{\text{ob}}$ and timesteps $t_{\text{ob}}$ to perform forward diffusion on the normal product images $x_{\text{ob}}$. Then the noisy latent $\hat{z}_{ob}$ and the embedding $\mathbf{e}_{\text{ob}}$ corresponding to the normal tokens of $\mathcal{P}$, i.e., "a `<ob>`", are input into the U-Net in predicting noise, and then calculate the NA loss with $\epsilon_{\text{ob}}$.

**Mixed Training.** Based on the separated DA loss for abnormal images and NA loss for the normal images, the objective of Separation and Sharing Fine-tuning is expressed as:

$$\mathcal{L} = \mathcal{L}_{\mathbf{df}} + \mathcal{L}_{\mathbf{ob}} \tag{6}$$

In the training process, instead of training a single U-Net model for each anomaly type, we train a unified U-Net model for each product. Specifically, given a product image set, which contains $G$ anomaly categories and some normal images with their corresponding masks. We group all the abnormal images of a product into a unified set $X_{\text{df}} = \{x_{\text{df}}^1, x_{\text{df}}^2, .., x_{\text{df}}^H\}$. For each anomaly type, we use $\mathcal{P}$ with different sets of anomaly tokens. In addition, we sample a fixed number of normal images to consist of the normal training set $X_{\text{ob}} = \{x_{\text{ob}}^1, x_{\text{ob}}^2, .., x_{\text{ob}}^P\}$. During each step of our fine-tuning process, we sample same number of images from both $X_{\text{df}}$ and $X_{\text{ob}}$, and mixed them into a batch. We found that such a mixed training strategy not only alleviates the overfitting caused by the limited number of each anomaly type, but also increases the diversity of the anomaly image, while still maintaining reasonable authenticity, as is shown in Fig. 3(c), (i) indicates that the model with mixed training may generate new anomalies, e.g., the anomalies inside the dotted white line. In contrast, the anomalies in (ii) overfit the training images. More ablation studies on the mixed training strategy are shown in Tab. 23 in appendix A.8.

### 3.3 REFINED MASK PREDICTION

High-fidelity pixel-wise annotations of anomalies play an important role in boosting segmentation models. However, existing methods, such as DFMGAN (Duan et al., 2023) and AnomlayDiffusion (Hu et al., 2024), produce anomaly masks that are not tightly matched with generated anomalies, which is insufficient for training segmentation model. To address this issue, we design a cascaded Refined Mask Prediction (RMP) branch, which is grafted onto the U-Net trained according to SeaS (mentioned in Sec. 3.2). As shown in Fig. 4, RMP consists of two steps, firstly capturing discriminative features from U-Net and secondly combining it with high-resolution features of VAE decoder to generate anomaly-matched masks.

**Coarse Feature Extraction.** The first step aims to extract a coarse but highly-discriminative feature for anomalies from the U-Net decoder. Specifically, let $F_1 \in \mathbb{R}^{32 \times 32 \times 1280}$ and $F_2 \in \mathbb{R}^{64 \times 64 \times 640}$ denote the output feature of "`up-2`" and "`up-3`" layers of the decoder in U-Net, respectively. We first leverage a $1 \times 1$ convolution block to compress the channel of $F_1$ and $F_2$ to $\overline{F}_1 \in \mathbb{R}^{32 \times 32 \times 128}$ and $\overline{F}_2 \in \mathbb{R}^{64 \times 64 \times 64}$, respectively. Then, we upsample $\overline{F}_1$ to $64 \times 64$ resolution and concatenate it with $\overline{F}_2$. Finally, four transformer layers are employed to fuse the concatenated features and obtain a unified coarse feature $\hat{F} \in \mathbb{R}^{64 \times 64 \times 192}$.

**Mask Refinement Module.** Directly upsampling the coarse feature $\hat{F}$ to high resolution will result in a loss of anomaly details. Therefore, we design the Mask Refinement Module (MRM) to refine the coarse feature $\hat{F}$ in a progressive manner. As shown in Fig. 4, each MRM takes in two features, i.e., the high-resolution features from VAE and the discriminative feature to be refined. Firstly, the discriminative feature is upsampled to align with the high-resolution features of VAE. To preserve the discriminative ability, the upsampled feature is processed through two chained convolution blocks for capturing multi-scale anomaly features and a $1 \times 1$ convolution for capturing local

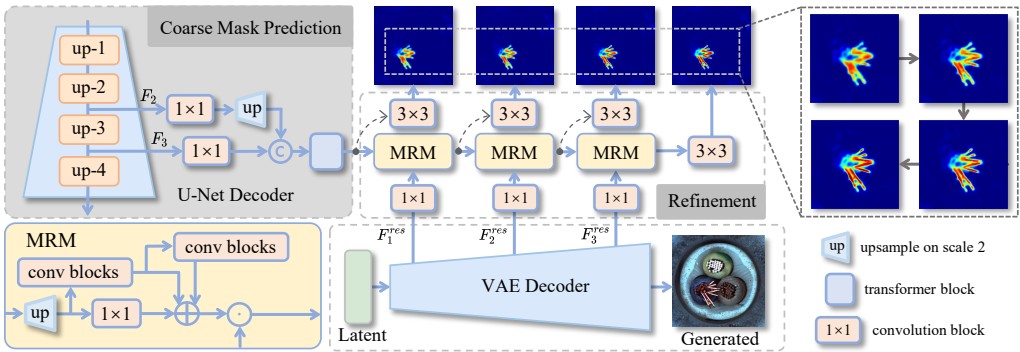

Figure 4: The **Refined Mask Prediction (RMP)** branch during inference. The Coarse Feature Extraction utilizes features from the up-2 and up-3 layers of U-Net Decoder to extract coarse features. The cascaded Mask Refinement Module (MRM) further obtains the mask accurately aligned with the anomaly with the assistance of high-resolution features of the VAE Decoder.

features. These features are then summed, and multiplied with the VAE features element-wisely to enhance the anomalies' boundary. Finally, MRM employs a $3 \times 3$ convolution to fuse the added features and output a refined feature.

To refine $\hat{F}$, we employ three MRMs positioned in sequence. Each MRM takes the previous MRM's output as the discriminant feature to be refined, while the first MRM takes $\hat{F}$ as the discriminative input. For another input of each MRM, we use the outputs from the 1-st, 2-nd, and 3-rd "up-blocks" of the VAE decoder respectively. In this way, the features obtained by the last MRM have the advantages of both high resolution and high discriminability. Finally, we use a $3 \times 3$ convolution and a softmax to generate the refined anomaly mask $\hat{M}'_{df} \in \mathbb{R}^{512 \times 512 \times 2}$ using the output of the last MRM.

**Loss Functions.** During training, we use $x_{df}$ and $x_{ob}$ as inputs. For $x_{df}$ we obtain the coarse mask $\hat{M}_{df} \in \mathbb{R}^{64 \times 64 \times 2}$ from the Coarse Feature Extraction and $\hat{M}'_{df}$ after the MRMs. Similarly, for $x_{ob}$ we obtain the $\hat{M}_{ob} \in \mathbb{R}^{64 \times 64 \times 2}$ from Coarse Feature Extraction and directly upsample it to the original resolution, denoted as $\hat{M}'_{ob} \in \mathbb{R}^{512 \times 512 \times 2}$. Then we conduct the supervision on both low-resolution and high-resolution predictions as,

$$\mathcal{L}_{\mathcal{M}} = \mathcal{F}(\hat{M}_{df}, \mathbf{M}_{df}) + \mathcal{F}(\hat{M}_{ob}, \mathbf{M}_{ob}) + \mathcal{F}(\hat{M}'_{df}, \mathbf{M}'_{df}) + \mathcal{F}(\hat{M}'_{ob}, \mathbf{M}'_{ob}) \tag{7}$$

where $\mathcal{F}$ indicates the Focal Loss (Lin et al., 2017). $\mathbf{M}_{ob} \in \mathbb{R}^{64 \times 64 \times 1}$ and $\mathbf{M}'_{ob} \in \mathbb{R}^{512 \times 512 \times 1}$ are used to suppress noise in normal images, with each pixel value set to 0. $\mathbf{M}_{df} \in \mathbb{R}^{64 \times 64 \times 1}$ and $\mathbf{M}'_{df} \in \mathbb{R}^{512 \times 512 \times 1}$ are the ground truth masks of abnormal images. More ablation studies on the effect of normal images in training RMP branch are shown in Tab. 27 and Fig. 16 in appendix A.8.

### 3.4 INFERENCE

During the generation of the abnormal image-mask pairs, aiming further to ensure the global consistency of the abnormal image, we random select a normal image $x_{ob}$ from $X_{ob}$ as input, and add random noise to $x_{ob}$, which resulting in an initial noisy latent $\hat{z}_0$. Next, $\hat{z}_0$ is input into the U-Net for noise prediction, with the process guided by the conditioning vector $\mathbf{e}_{df}$ (mentioned in Eq. 4). In the final three denoising steps, the RMP branch (Sec. 3.3) leverages the features from the U-Net decoder and VAE decoder to generate the final anomaly mask. Specifically, we average the refined anomaly mask from these steps to obtain the refined mask $\hat{M}'_{df} \in \mathbb{R}^{512 \times 512 \times 2}$. Then we take the threshold $\tau$ for the second channel of $\hat{M}'_{df}$ to segment the final anomaly mask $M_{df} \in \mathbb{R}^{512 \times 512 \times 1}$. The effect of $\tau$ on the downstream segmentation models is shown in Tab. 29 in appendix A.8. In the last denoising step, the output of the generation model is used as the generated abnormal image.

Table 1: Comparison on IS and IC-LPIPS on MVTec AD. Bold indicates the best performance, while underlined denotes the second-best result.

| Category | CDC | | Crop& Paste | | SDGAN | | Defect-GAN | | DFMGAN | | Anomaly Diffusion | | Ours | |
|---|---|---|---|---|---|---|---|---|---|---|---|---|---|---|
| | IS ↑ | IC-L ↑ | IS ↑ | IC-L ↑ | IS ↑ | IC-L ↑ | IS ↑ | IC-L ↑ | IS ↑ | IC-L ↑ | IS ↑ | IC-L ↑ | IS ↑ | IC-L ↑ |
| bottle | 1.52 | 0.04 | 1.43 | 0.04 | 1.57 | 0.06 | 1.39 | 0.07 | _1.62_ | 0.12 | 1.58 | _0.19_ | **1.78** | **0.21** |
| cable | 1.97 | 0.19 | 1.74 | 0.25 | 1.89 | 0.19 | 1.70 | 0.22 | 1.96 | 0.25 | **2.13** | _0.41_ | _2.09_ | **0.42** |
| capsule | 1.37 | 0.06 | 1.23 | 0.05 | 1.49 | 0.03 | **1.59** | 0.04 | **1.59** | 0.11 | **1.59** | _0.21_ | _1.56_ | **0.26** |
| carpet | **1.25** | 0.03 | 1.17 | 0.11 | 1.18 | 0.11 | _1.24_ | 0.12 | 1.23 | 0.13 | 1.16 | _0.24_ | 1.13 | **0.25** |
| grid | 1.97 | 0.07 | 2.00 | 0.12 | 1.95 | 0.10 | 2.01 | 0.12 | 1.97 | _0.13_ | _2.04_ | **0.44** | **2.43** | **0.44** |
| hazelnut | _1.97_ | 0.05 | 1.74 | 0.21 | 1.85 | 0.16 | 1.87 | 0.19 | 1.93 | _0.24_ | **2.13** | **0.31** | 1.87 | **0.31** |
| leather | 1.80 | 0.07 | 1.47 | 0.14 | 2.04 | 0.12 | **2.12** | 0.14 | _2.06_ | 0.17 | 1.94 | **0.41** | 2.03 | _0.40_ |
| metal_nut | 1.55 | 0.04 | 1.56 | 0.15 | 1.45 | 0.28 | 1.47 | 0.30 | 1.49 | **0.32** | **1.96** | 0.30 | _1.64_ | _0.31_ |
| pill | 1.56 | 0.06 | 1.49 | 0.11 | 1.61 | 0.07 | 1.61 | 0.10 | **1.63** | 0.16 | 1.61 | _0.26_ | _1.62_ | **0.33** |
| screw | 1.13 | 0.11 | 1.12 | 0.16 | 1.17 | 0.10 | 1.19 | 0.12 | 1.12 | 0.14 | _1.28_ | _0.30_ | **1.52** | **0.31** |
| tile | 2.10 | 0.12 | 1.83 | 0.20 | 2.53 | 0.21 | 2.35 | 0.22 | 2.39 | 0.22 | _2.54_ | **0.55** | **2.60** | _0.50_ |
| toothbrush | 1.63 | 0.06 | 1.30 | 0.08 | 1.78 | 0.03 | _1.85_ | 0.03 | 1.82 | 0.18 | 1.68 | _0.21_ | **1.96** | **0.25** |
| transistor | 1.61 | 0.13 | 1.39 | 0.15 | **1.76** | 0.13 | 1.47 | 0.13 | _1.64_ | _0.25_ | 1.57 | **0.34** | 1.51 | **0.34** |
| wood | 2.05 | 0.03 | 1.95 | 0.23 | 2.12 | 0.25 | 2.19 | 0.29 | 2.12 | 0.35 | _2.33_ | _0.37_ | **2.77** | **0.46** |
| zipper | 1.30 | 0.05 | 1.23 | 0.11 | 1.25 | 0.10 | 1.25 | 0.10 | 1.29 | _0.27_ | _1.39_ | 0.25 | **1.63** | **0.30** |
| Average | 1.65 | 0.07 | 1.51 | 0.14 | 1.71 | 0.13 | 1.69 | 0.15 | 1.72 | 0.20 | _1.80_ | _0.32_ | **1.88** | **0.34** |

# 4 EXPERIMENTS

## 4.1 EXPERIMENTAL SETTINGS

**Implementation Details.** We train SeaS by fine-tuning the pre-trained Stable Diffusion v1-4 (Rombach et al., 2022). In AIG experiments, we use 60 normal images and 1/3 abnormal images with their corresponding masks per anomaly type for training. During inference, we generate 1,000 abnormal image-mask pairs for a single anomaly type. More details are given in appendix A.3.

**Datasets.** We conduct experiments on MVTec AD dataset (Bergmann et al., 2019) and MVTec 3D AD dataset (only RGB images) (Bergmann. et al., 2022). MVTec AD dataset contains 15 product categories, each with up to 8 different anomalies, making it suitable for simulating real-world industrial scenarios. MVTec 3D AD dataset includes 10 product categories, each with up to 4 different anomalies. It contains more challenges, i.e., lighting condition variations, product pose variations. Due to the page limitation, results on MVTec 3D AD dataset are given in appendix A.4.

**Evaluation Metrics.** For AIG, we leverage 2 metrics: the Inception Score (IS) and the Intra-cluster pairwise LPIPS distance (IC-LPIPS)(Ojha et al., 2021). The scarcity of abnormal images hampers the reliability of FID (Heusel et al., 2017) and KID (Bińkowski et al., 2018), as overfitted model (Duan et al., 2023) achieves higher scores. For pixel-level anomaly segmentation and image-level anomaly detection, we report 3 metrics: Area Under Receiver Operator Characteristic curve (AUROC), Average Precision (AP) and $F_1$-score at optimal threshold ($F_1$-max). In addition, we report Intersection over Union (IoU) for segmentation.

## 4.2 COMPARISON IN ANOMALY IMAGE GENERATION

**Comparison Methods.** We compare SeaS with the current AG and AIG methods on generation fidelity and diversity, such as CDC (Ojha et al., 2021), Crop&Paste (Lin et al., 2021), SDGAN (Niu et al., 2020), Defect-GAN (Zhang et al., 2021a), DFMGAN (Duan et al., 2023) and AnomalyDiffusion (Hu et al., 2024). Then we use Crop&Paste, DRAEM (Zavrtanik et al., 2021), DFMGAN, AnomalyDiffusion and our method to generate anomaly image-mask pairs. These pairs are used to train BiSeNet V2 (Yu et al., 2021), UPerNet (Xiao et al., 2018) and LFD (Zhou et al., 2024a) respectively. Different from AnomalyDiffusion (Hu et al., 2024), which trains one segmentation model per product, we train a unified segmentation model for all the products. We also compare the segmentation results based on SeaS with the state-of-the-art unsupervised anomaly detection methods, such as RealNet (Zhang et al., 2024) and HVQ-Trans (Lu et al., 2023), in appendix A.5.

**Anomaly image generation quality.** In Tab. 1, we compare SeaS with some state-of-the-art AG and AIG methods on generation fidelity (IS) and diversity (IC-LPIPS). SeaS achieves 1.88 on IS and 0.34 on IC-LPIPS, which demonstrates that our method generates anomaly images with higher fidelity and diversity. We exhibit the generated anomaly images in Fig. 5, the anomaly images

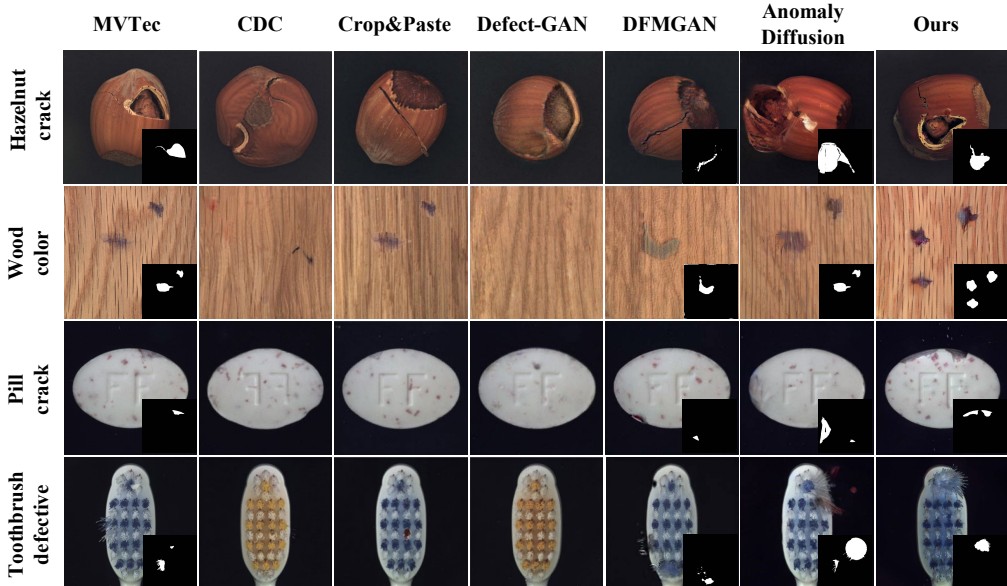

MVTec   CDC   Crop&Paste   Defect-GAN   DFMGAN   Anomaly Diffusion   Ours

Hazelnut crack

Wood color

Pill crack

Toothbrush defective

Figure 5: Visualization of the generation results on MVTec AD. The sub-image in the lower right corner is the generated mask, none means that the method cannot generate masks.

Table 2: Comparison on anomaly segmentation on MVTec AD.

| Model | DRAEM | | | | DFMGAN | | | | AnomalyDiffusion | | | | Ours | | | |
|---|---|---|---|---|---|---|---|---|---|---|---|---|---|---|---|---|
| | AUROC | AP | $F_1$-max | IoU | AUROC | AP | $F_1$-max | IoU | AUROC | AP | $F_1$-max | IoU | AUROC | AP | $F_1$-max | IoU |
| BiSeNet V2 | 81.37 | 38.90 | 42.62 | 39.39 | 94.57 | 60.42 | 60.54 | 45.83 | 96.27 | 64.50 | 62.27 | 42.89 | 97.21 | 69.21 | 66.37 | 55.28 |
| UPerNet | 83.21 | 42.78 | 45.97 | 42.03 | 92.33 | 57.01 | 56.91 | 46.64 | 96.87 | 69.92 | 66.95 | 50.80 | 97.87 | 74.42 | 70.70 | 61.24 |
| LFD | 76.41 | 40.99 | 43.91 | 35.61 | 94.91 | 67.06 | 65.09 | 45.49 | 96.30 | 69.77 | 66.99 | 45.77 | 98.09 | 77.15 | 72.52 | 56.47 |
| Average | 80.33 | 40.89 | 44.17 | 39.01 | 93.94 | 61.50 | 60.85 | 45.99 | 96.48 | 68.06 | 65.40 | 46.49 | **97.72** | **73.59** | **69.86** | **57.66** |

Table 3: Comparison on image-level anomaly detection on MVTec AD.

| Model | DRAEM | | | DFMGAN | | | AnomalyDiffusion | | | Ours | | |
|---|---|---|---|---|---|---|---|---|---|---|---|---|
| | AUROC | AP | $F_1$-max | AUROC | AP | $F_1$-max | AUROC | AP | $F_1$-max | AUROC | AP | $F_1$-max |
| BiSeNet V2 | 89.87 | 93.51 | 89.97 | 90.90 | 94.43 | 90.33 | 90.08 | 94.84 | 91.84 | 96.00 | 98.14 | 95.43 |
| UPerNet | 89.45 | 93.92 | 89.66 | 90.74 | 94.43 | 90.37 | 96.62 | 98.61 | 96.21 | 98.29 | 99.20 | 97.34 |
| LFD | 87.94 | 93.41 | 88.65 | 91.08 | 95.40 | 90.58 | 95.15 | 97.78 | 94.66 | 95.88 | 97.89 | 95.15 |
| Average | 89.09 | 93.61 | 89.43 | 90.91 | 94.75 | 90.43 | 93.95 | 97.08 | 94.24 | **96.72** | **98.41** | **95.97** |

generated by our method have higher fidelity (e.g., *hazelnut_crack*). Compared with other methods, SeaS can generate images with different types, colors, and shapes of anomalies rather than overfitting to the training images (e.g., *wood_color* and *pill_crack*). The masks generated by our method are also precisely aligned with the anomaly regions (e.g., *toothbrush_defective*). More qualitative and quantitative anomaly image generation results are in appendix A.6.

**Anomaly segmentation and detection.** We generate 1,000 image-mask pairs for each anomaly type, and use the image-mask pairs of all products along with all the training normal images to train the unified segmentation model, rather than training separate segmentation models for each product. We test the models on the rest images of the testing set of MVTec AD, which are not included in the training set for generation. The results are given in Tab. 2. All the methods are trained using the same number of images and the same training settings, detailed in appendix A.7. The segmentation results consistently demonstrate that our method outperforms others across all the segmentation models, with an 11.17% average improvement on IoU. We show the segmentation anomaly maps in Fig. 6. By using our generated image-mask pairs to train BiSeNet V2, there are fewer false positives in *wood_combined* and fewer false negatives in *bottle_contamination* and *carpet_cut*. In addition, we use the maximum value of the segmentation anomaly map as the image-level anomaly score for anomaly detection. We report the image-level metrics in Tab. 3, and our method achieves a 2.77% gain on image-AUROC. More qualitative comparison results on anomaly segmentation are in appendix A.9 and appendix A.10.

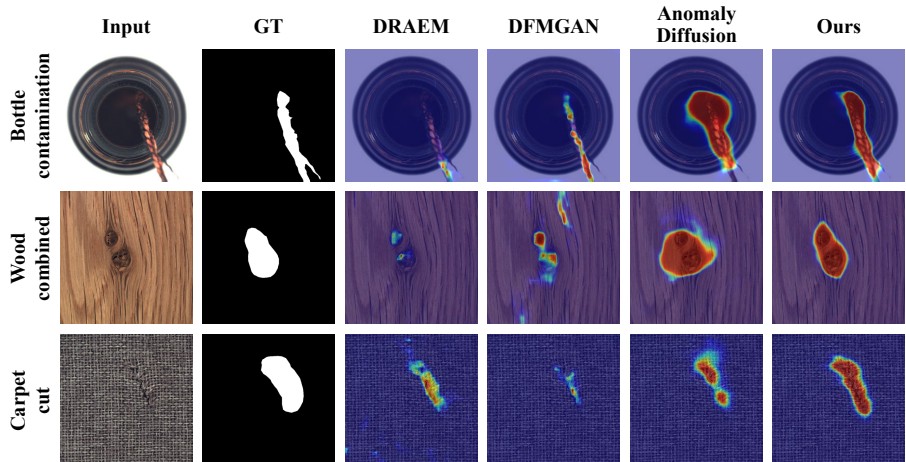

Figure 6: Qualitative anomaly segmentation results with BiSeNet V2 on MVTec AD.

## 4.3 ABLATION STUDY

**Anomaly image generation.** We train additional models to assess the effect of each component: **(a)** the model with typical text prompt with fixed generic semantic words (short for with TP in Tab. 4); **(b)** the model without mixing the different types of anomaly images in the same product; **(c)** the model without NA loss; **(d)** the model without the second term of DA loss in Eq. 3 (short for ST in Tab. 4); **(e)** our complete model. We use these models to generate 1,000 anomaly image-mask pairs per anomaly type and train BiSeNet V2 for anomaly segmentation. In Tab. 4, the results show that omitting any component leads to a decrease in the fidelity and diversity of the generated images, as well as a decrease in the segmentation results. These validate the effectiveness of the components we proposed. More ablation studies on SeaS are shown in appendix A.8.

**Refined Mask Prediction branch.** To verify the validity of the components in the RMP branch, we conduct ablation studies on MRM, the progressive manner to refine coarse feature (short for PM in Tab. 5) and coarse mask supervision (short for CMS in Tab. 5). **1)** the model without any components, which means we do not use MRM to fuse the high-resolution features in RMP, but directly obtain the mask from the coarse features $\hat{F} \in \mathbb{R}^{64 \times 64 \times 192}$ through convolution and bilinear interpolation upsampling; **2)** the model with MRM; **3)** the model utilizing the MRM in a progressive manner to refine coarse features; **4)** our complete model. We report the BiSeNet V2 results in Tab. 5, which demonstrates that each component in the RMP is indispensable for downstream anomaly segmentation. More ablation studies about RMP are in appendix A.8.

Table 4: Ablation on the generation model.

| Method | IS | IC-L | AUROC | AP | $F_1$-max | IoU |
|---|---|---|---|---|---|---|
| (a) with TP | 1.72 | 0.33 | 94.72 | 57.16 | 55.67 | 50.46 |
| (b) w/o Mixed | 1.79 | 0.32 | 95.82 | 66.07 | 64.50 | 53.11 |
| (c) w/o NA | 1.67 | 0.31 | 96.20 | 66.03 | 64.09 | 53.97 |
| (d) w/o ST | 1.86 | 0.33 | 96.44 | 67.73 | 65.23 | 54.99 |
| (e) All (Ours) | 1.88 | 0.34 | 97.21 | 69.21 | 66.37 | 55.28 |

Table 5: Ablation on the RMP branch.

| MRM | PM | CMS | AUROC | AP | $F_1$-max | IoU |
|---|---|---|---|---|---|---|
| | | | 97.00 | 65.28 | 62.56 | 53.93 |
| ✓ | | | 94.54 | 60.52 | 59.06 | 49.42 |
| ✓ | ✓ | | 94.04 | 62.04 | 59.82 | 50.44 |
| ✓ | ✓ | ✓ | **97.21** | **69.21** | **66.37** | **55.28** |

## 5 CONCLUSION

In this paper, we propose a novel few-shot industrial anomaly image generation method named SeaS. We explore an implicit characteristic that the anomalies exhibit randomness in shape and appearance, while the products maintain global consistency with minor variations in local details. We design a Separation and Sharing Fine-tuning strategy for industrial anomaly image generation, and a Refined Mask Prediction branch to obtain a fine-grained mask. Our method surpasses existing methods on both AIG and downstream anomaly segmentation tasks.

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

# A APPENDIX

## A.1 OVERVIEW

This supplementary material consists of:

- Analysis on decoupled anomaly alignment loss and multiple tokens (Sec. A.2).
- More implementation details (Sec. A.3).
- Anomaly image and mask generation results, downstream segmentation results on MVTec 3D AD dataset (RGB images) (Sec. A.4).
- More quantitative comparison with unsupervised anomaly detection methods (Sec. A.5).
- More qualitative and quantitative results of anomaly image generation (Sec. A.6).
- More details of downstream segmentation model implementation and usage (Sec. A.7).
- More ablation studies (Sec. A.8), including ablation studies on the Unbalanced Abnormal Text Prompt design, the Separation and Sharing Fine-tuning loss, the minimum size requirement for training images, the training strategy of SeaS, the cross-attention maps for Decoupled Anomaly Alignment, the features for Coarse Feature Extraction, the features of VAE for Refined Mask Prediction, the normal image supervision for Refined Mask Prediction, the Mask Refinement Module, and the threshold for mask binarization.
- Qualitative comparison results of segmentation models trained on image-mask pairs generated by different anomaly generation methods (Sec. A.9).
- Qualitative comparison results of different segmentation models trained on image-mask pairs generated by SeaS (Sec. A.10).
- Anomaly image and mask generation results, downstream segmentation results on VisA dataset (Sec. A.11).
- Explanation of discriminative features in U-Net decoder (Sec. A.12).
- Comparison with the Textual Inversion (Sec. A.13).
- More experiments on lighting conditions (Sec. A.14).
- More experiments on replacing generation strategies (Sec. A.15).
- More visualization on recombining the decoupled attributes for unseen anomalies (Sec. A.16).

## A.2 ANALYSIS ON DECOUPLED ANOMALY ALIGNMENT LOSS AND MULTIPLE TOKENS

Here we give a more detailed analysis of the learning process of the DA loss. According to Eq. 3, intuitively, the DA loss may pull the anomaly tokens similar to each other. However, the U-Net in Stable Diffusion uses multi-head attention, which ensures different anomaly tokens cover different attributes of the anomalies. In Eq. 3, the cross-attention map is the multiply of the feature map of U-Net and the anomaly tokens. In the implementation of multi-head attention, both the learnable embedding of the anomaly token and the U-Net feature are decomposed into several groups along the channel dimension. E.g., the conditioning vector $e_a \in \mathbb{R}^{1 \times C_1}$, which is corresponding to anomaly token, is divided into $\{e_{a,i} \in \mathbb{R}^{1 \times \frac{C_1}{q}} | i \in [1, q]\}$, and the image feature $v \in \mathbb{R}^{r \times r \times C_2}$ is divided into $\{v_i \in \mathbb{R}^{1 \times \frac{C_2}{q}} | i \in [1, q]\}$, where $q$ is the number of heads in the multi-head attention. Then the corresponding groups are multiplied, and the outputs of all the heads are averaged. The attention map $A$ of $e_a$ is calculated by:

$$A = \frac{1}{q} \sum_{i=1}^{q} \text{softmax}(\frac{Q_i K_{a,i}^\top}{\sqrt{d}}), Q_i = \phi_q(v_i), K_{a,i} = \phi_k(e_{a,i}). \tag{8}$$

Therefore, in the defect region, the DA loss only ensures the average of each head tends to 1, but does not require the anomaly tokens to be the same with each other. In addition, each $e_a$ is different from each other, and is combined by $e_{a,i}$. **The update direction of each $e_{a,i}$ is related to $v_i$ and covers some features of the defect, it encompasses the attributes of anomalies from various perspectives, thereby providing diversified information.**

## A.3 More implementation details

**More training details.** For the Unbalanced Abnormal Text Prompt, we set the number $N$ of multiple $<\text{df}_n>$ to 4 and the number $N'$ of $<\text{ob}>$ to 1, these parameters are fixed across all product classes. For example, for the normal token $<\text{ob}>$, given the lookup $\mathcal{U} \in \mathbb{R}^{b \times 768}$, where $b$ is the number of text embeddings stored by the pre-trained text encoder, we use a placeholder string `"ob1"` as the input. Firstly, `"ob1"` is converted to a token ID $s_{\text{ob1}} \in \mathbb{R}^{1 \times 1}$ in the tokenizer. Secondly, $s_{\text{ob1}} \in \mathbb{R}^{1 \times 1}$ is converted to a one-hot vector $\mathcal{S}_{\text{ob1}} \in \mathbb{R}^{1 \times (b+1)}$. Thirdly, one learnable new embedding $g \in \mathbb{R}^{1 \times 768}$ corresponding to $s_{\text{ob1}}$ is inserted to the lookup $\mathcal{U}$, resulting in $\mathcal{U}' \in \mathbb{R}^{(b+1) \times 768}$. Here, $g \in \mathbb{R}^{1 \times 768}$ is the learnable embedding of $<\text{ob}>$. **These embeddings and U-Net are learnable during the fine-tuning process.**

**Training image generation model.** For each product, we perform $800 \times G$ steps for fine-tuning, where $G$ represents the number of anomaly categories of the product. The batch size of training image generation model is set to 4. During each step of our fine-tuning process, we sample 2 images from the abnormal training set $X_{\text{df}}$, and 2 images from the normal training set $X_{\text{ob}}$. We utilize the AdamW (Loshchilov & Hutter, 2018) optimizer with a learning rate of U-Net is $4 \times 10^{-6}$. The learning rate of the text embedding is $4 \times 10^{-5}$.

**Training Refined Mask Prediction branch.** We design a cascaded Refined Mask Prediction (RMP) branch, which is grafted onto the U-Net trained according to SeaS. For each product, we perform $800 \times G$ steps for the RMP model, where $G$ represents the number of anomaly types for the product. The batch size of training the RMP branch is set to 4. During each step of our fine-tuning process, we sample 2 images with their corresponding masks from the abnormal training set $X_{\text{df}}$, and 2 images from the normal training set $X_{\text{ob}}$. The masks used to suppress noise in normal images has each pixel value set to 0. The learning rate of the RMP model is $5 \times 10^{-4}$.

**Metrics.** For anomaly image generation, we report 2 metrics: the Inception Score (IS) and Intra-cluster pairwise LPIPS Distance (IC-LPIPS). The Inception Score (IS), proposed in (Barratt & Sharma, 2018), serves as an independent metric to evaluate the fidelity and diversity of generated images, by measuring the mutual information between input samples and their predicted classes. The IC-LPIPS (Ojha et al., 2021) is used to evaluate the diversity of generated images, which quantifies the perceptual similarity between image patches in the same cluster. For pixel-level anomaly segmentation and image-level anomaly detection, we report 3 metrics: Area Under Receiver Operator Characteristic curve (AUROC), Average Precision (AP), and $F_1$-score at the optimal threshold ($F_1$-max). **All of these metric are calculated using the *scikit-learn* library.** In addition, we calculate the Intersection over Union (IoU) to more accurately evaluate the anomaly segmentation result.

**Resource requirement and time consumption.** We conduct our training on NVIDIA Tesla A100 40G GPU. Specifically, we use a single A100 to train a generation model sequentially for each product category, with each training process occupying about 20G of GPU memory. Since each anomaly type requires isolate training, the training time depends on the total amount of anomaly types across all products. For example, the product *metal_nut* contains 4 anomaly types, and each needs around 35 minutes. The generation model for *metal_nut* spends 2 hours and 20 minutes on training in total. For the RMP branch, each anomaly type needs around 25 minutes. Hence, it takes 1 hour and 40 minutes to train *metal_nut*. More details are given in Tab. 6, where $K$ is the total number of anomaly types across all products. The comparison on time consumption is shown in Tab. 7. For the MVTecAD datasets with 73 anomaly types, our training takes 73 hours, which is shorter than the 249 hours required by AnomalyDiffusion and the 414 hours required by DFMGAN. In terms of inference time, SeaS costs 720 ms per image, which is shorter than the 3830 ms per image required by the Diffusion-based method AnomalyDiffusion. The inference time of the GAN-based method DFMGAN is 48ms per image.

Table 6: Computational resource and training time.

| Stage | Time (minutes per product) | GPU(MB) | Overall Time |
|---|---|---|---|
| Generation Model | $35 \times K$ | 20242 | 42 hours and 35 minutes |
| RMP branch | $25 \times K$ | 23280 | 30 hours and 25 minutes |

Table 7: Comparison on time consumption.

| Model | Overall Training Time(hours) | Inference Time (ms) |
|---|---|---|
| DFMGAN (Duan et al., 2023) | 414 | 48 |
| AnomalyDiffusion (Hu et al., 2024) | 249 | 3830 |
| Ours | 73 | 720 |

## A.4 ADDITIONAL DATASET RESULTS

We perform experimental evaluations on the RGB images of the MVTec 3D AD Dataset (Bergmann. et al., 2022), which includes 10 product categories, each with up to 4 different anomalies. It encompasses several common challenges, such as variations in lighting conditions and product poses, which are crucial for validating the robustness of image generation methods. The experimental settings are the same as those in Sec. 4.1 and Sec. A.3.

Table 8: Comparison on IS and IC-LPIPS on MVTec 3D AD. Bold indicates the best performance.

| Category | DFMGAN (Duan et al., 2023) | | AnomalyDiffusion (Hu et al., 2024) | | Ours | |
|---|---|---|---|---|---|---|
| | IS ↑ | IC-L ↑ | IS ↑ | IC-L ↑ | IS ↑ | IC-L ↑ |
| bagel | 1.07 | 0.26 | 1.02 | 0.22 | **1.28** | **0.29** |
| cable_gland | 1.59 | **0.25** | 1.79 | 0.19 | **2.21** | 0.19 |
| carrot | 1.94 | **0.29** | 1.66 | 0.17 | **2.07** | 0.22 |
| cookie | 1.80 | 0.31 | 1.77 | 0.29 | **2.07** | **0.38** |
| dowel | **1.96** | **0.37** | 1.60 | 0.20 | 1.95 | 0.26 |
| foam | 1.50 | 0.17 | 1.77 | 0.30 | **2.20** | **0.39** |
| peach | 2.11 | **0.34** | 1.91 | 0.23 | **2.40** | 0.28 |
| potato | **3.05** | **0.35** | 1.92 | 0.17 | 1.98 | 0.22 |
| rope | 1.46 | 0.29 | 1.28 | 0.25 | **1.53** | **0.41** |
| tire | 1.53 | 0.25 | 1.35 | 0.20 | **1.81** | **0.31** |
| Average | 1.80 | 0.29 | 1.61 | 0.22 | **1.95** | **0.30** |

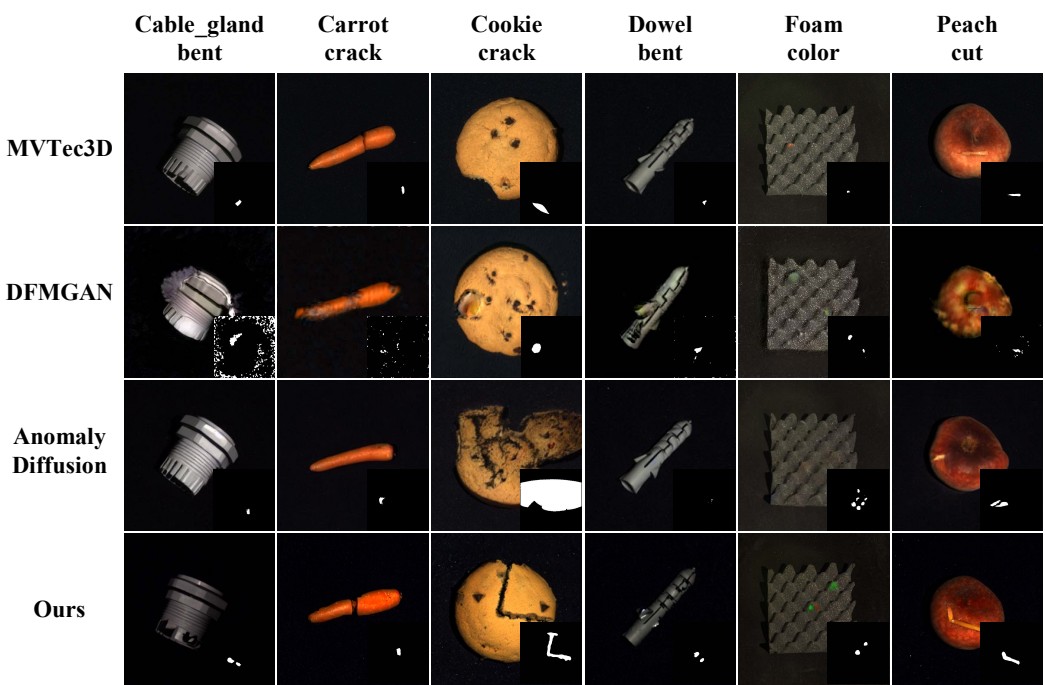

Figure 7: Visualization of the generation results on MVTec 3D AD. The sub-image in the lower right corner is the generated mask.

**Comparison Methods**

In terms of compared approaches, since existing state-of-the-art approaches, e.g., DFMGAN(Duan et al., 2023), AnomalyDiffusion (Hu et al., 2024), conducted the experimental evaluations only on MVTec AD dataset (Bergmann et al., 2019), we evaluate them on MVTec 3D AD dataset (Bergmann. et al., 2022) using their official source codes.

**Anomaly image generation quality**

As presented in Tab. 8, SeaS achieves scores of 1.95 on IS and 0.30 on IC-LPIPS, demonstrating our method's ability to generate anomaly images with superior fidelity and diversity. The generated anomaly images are shown in Fig. 7. SeaS can generate images with diverse anomalies, avoiding overfitting to the training images (e.g., *cookie_crack* and *foam_color*), while ensuring the fidelity of the generated images (e.g., *cable_gland_bent*). Additionally, the masks generated by our method are accurately aligned with the anomalies (e.g., *peach_cut*).

**Anomaly segmentation and detection**

Tab. 9 shows the comparisons on downstream supervised segmentation trained by the generated images. It consistently demonstrates that our method outperforms others across all the segmentation models, with a 15.49% average improvement on IoU. The segmentation anomaly maps are shown in Fig. 8. There are fewer false positives (e.g., *potato_combined*) and fewer false negatives (e.g., *bagel_contamination*), when the BiSeNet V2 is trained on the image-mask pairs generated by our method. In addition, we report the image-level metrics in Tab. 10, and our method achieves a 6.74% gain on image-AUROC. Tab. 11 shows the comparisons of anomaly detection methods HVQ-Trans (Lu et al., 2023), Shape-guided (Chu et al., 2023), and FOD (Yao et al., 2023a) on anomaly segmentation tasks. Supervised segmentation models achieve better performance than most unsupervised AD methods on small-scale networks, with an IoU of 39.00% on LFD (0.936M). The pixel-level AUROC, which is sensitive to false negatives but less sensitive to false positives, of the Shape-guided method is higher. However, our observation indicates that the Shape-guided method has a high number of false positives. This significantly degrades the segmentation metrics, resulting in low pixel-level AP, $F_1$-max, and IoU scores. For industrial anomaly detection, an effective method should achieve a balance between false positives and false negatives.

Table 9: Comparison on anomaly segmentation on MVTec 3D AD.

| Model | DFMGAN (Duan et al., 2023) | | | | AnomalyDiffusion (Hu et al., 2024) | | | | Ours | | | |
|---|---|---|---|---|---|---|---|---|---|---|---|---|
| | AUROC | AP | $F_1$-max | IoU | AUROC | AP | $F_1$-max | IoU | AUROC | AP | $F_1$-max | IoU |
| BiSeNet V2 (Yu et al., 2021) | 75.89 | 15.02 | 21.73 | 15.68 | 92.39 | 15.15 | 20.09 | 14.70 | 90.41 | 26.04 | 32.61 | 28.55 |
| UPerNet (Xiao et al., 2018) | 75.12 | 19.54 | 26.04 | 18.78 | 88.48 | 28.95 | 35.81 | 25.04 | 91.93 | 38.51 | 43.53 | 38.56 |
| LFD (Zhou et al., 2024a) | 72.15 | 9.54 | 14.29 | 14.81 | 92.68 | 24.29 | 32.74 | 19.90 | 91.61 | 40.25 | 43.47 | 39.00 |
| Average | 74.39 | 14.70 | 20.69 | 16.42 | 91.18 | 22.80 | 29.55 | 19.88 | **91.32** | **34.93** | **39.87** | **35.37** |

Table 10: Comparison on image-level anomaly detection on MVTec 3D AD.

| Model | DFMGAN (Duan et al., 2023) | | | AnomalyDiffusion (Hu et al., 2024) | | | Ours | | |
|---|---|---|---|---|---|---|---|---|---|
| | AUROC | AP | $F_1$-max | AUROC | AP | $F_1$-max | AUROC | AP | $F_1$-max |
| BiSeNet V2 (Yu et al., 2021) | 61.88 | 81.80 | 84.44 | 61.49 | 81.35 | 85.36 | 73.60 | 87.75 | 85.82 |
| UPerNet (Xiao et al., 2018) | 67.56 | 84.53 | 84.99 | 76.56 | 90.42 | 87.35 | 82.75 | 92.59 | 88.72 |
| LFD (Zhou et al., 2024a) | 62.23 | 82.17 | 85.38 | 77.06 | 89.44 | 87.20 | 78.96 | 91.22 | 87.28 |
| Average | 63.89 | 82.83 | 84.94 | 71.70 | 87.07 | 86.64 | **78.44** | **90.52** | **87.27** |

Table 11: Comparison with anomaly detection methods on MVTec 3D AD.

| Model | Parameters | Image-level | | | Pixel-level | | | |
|---|---|---|---|---|---|---|---|---|
| | | AUROC | AP | $F_1$-max | AUROC | AP | $F_1$-max | IoU |
| SeaS + BiSeNet V2 (Yu et al., 2021) | 3.341M | 73.60 | 87.75 | 85.82 | 90.41 | 26.04 | 32.61 | 28.55 |
| SeaS + UPerNet (Xiao et al., 2018) | 64.042M | **82.57** | **92.59** | **88.72** | 91.93 | 38.51 | **43.53** | 38.56 |
| SeaS + LFD (Zhou et al., 2024a) | **0.936M** | 78.96 | 91.22 | 87.28 | 91.61 | **40.25** | 43.47 | **39.00** |
| HVQ-Trans (Lu et al., 2023) | 8.45M | 68.15 | 84.38 | 85.20 | 96.40 | 24.59 | 17.23 | 20.51 |
| Shape-guided (Chu et al., 2023) | 4.13M | 79.07 | 91.05 | **88.72** | **98.45** | 26.69 | 34.16 | 34.12 |
| FOD (Yao et al., 2023a) | 3.58M | 71.66 | 86.83 | 86.57 | 97.03 | 14.70 | 20.99 | 23.31 |

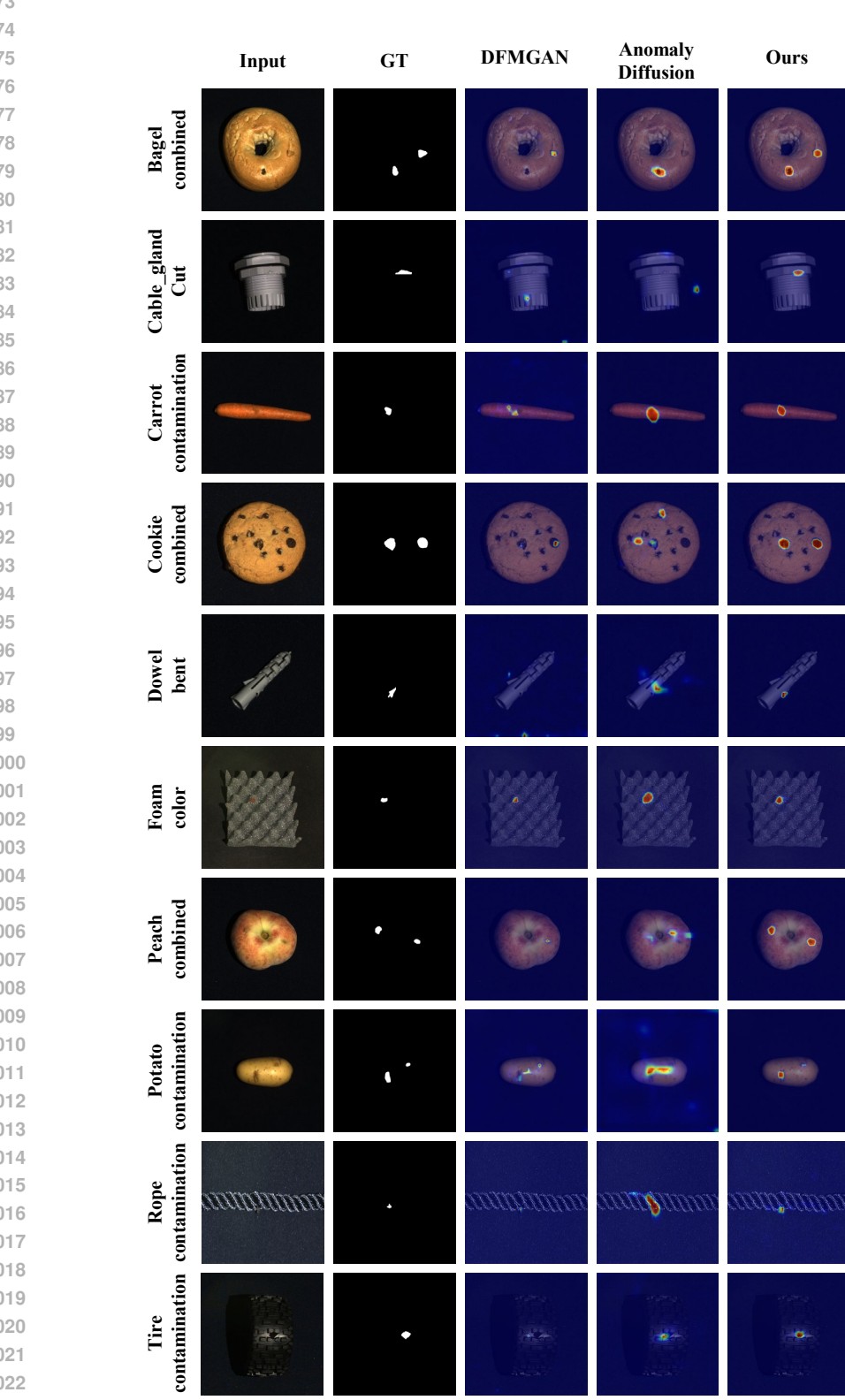

Figure 8: Qualitative anomaly segmentation results with BiSeNet V2 on MVTec 3D AD.

**More qualitative anomaly image generation results**

We provide further qualitative results of every category on the MVTec 3D AD dataset, from Fig. 9 to Fig. 10. We report the anomaly image generation results of SeaS for varying types of anomalies. The first column represents the generated anomaly images, the second column represents the corresponding generated masks.

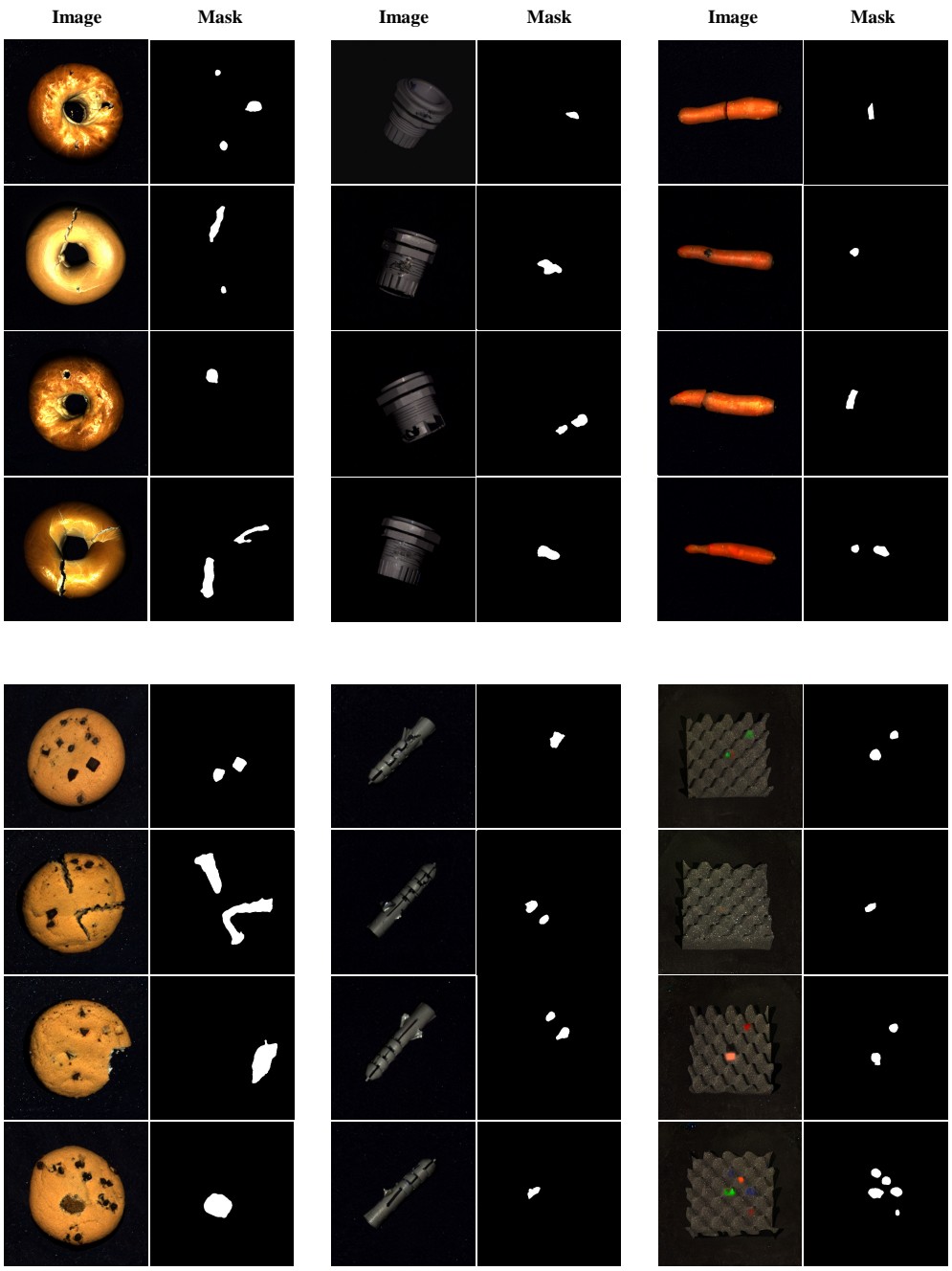

Figure 9: Qualitative results of our anomaly image generation results on MVTec 3D AD. In the first row, from left to right are the results for *bagel*, *cable_gland*, and *carrot* categories. In the second row, from left to right are the results for *cookie*, *dowel*, and *foam* categories.

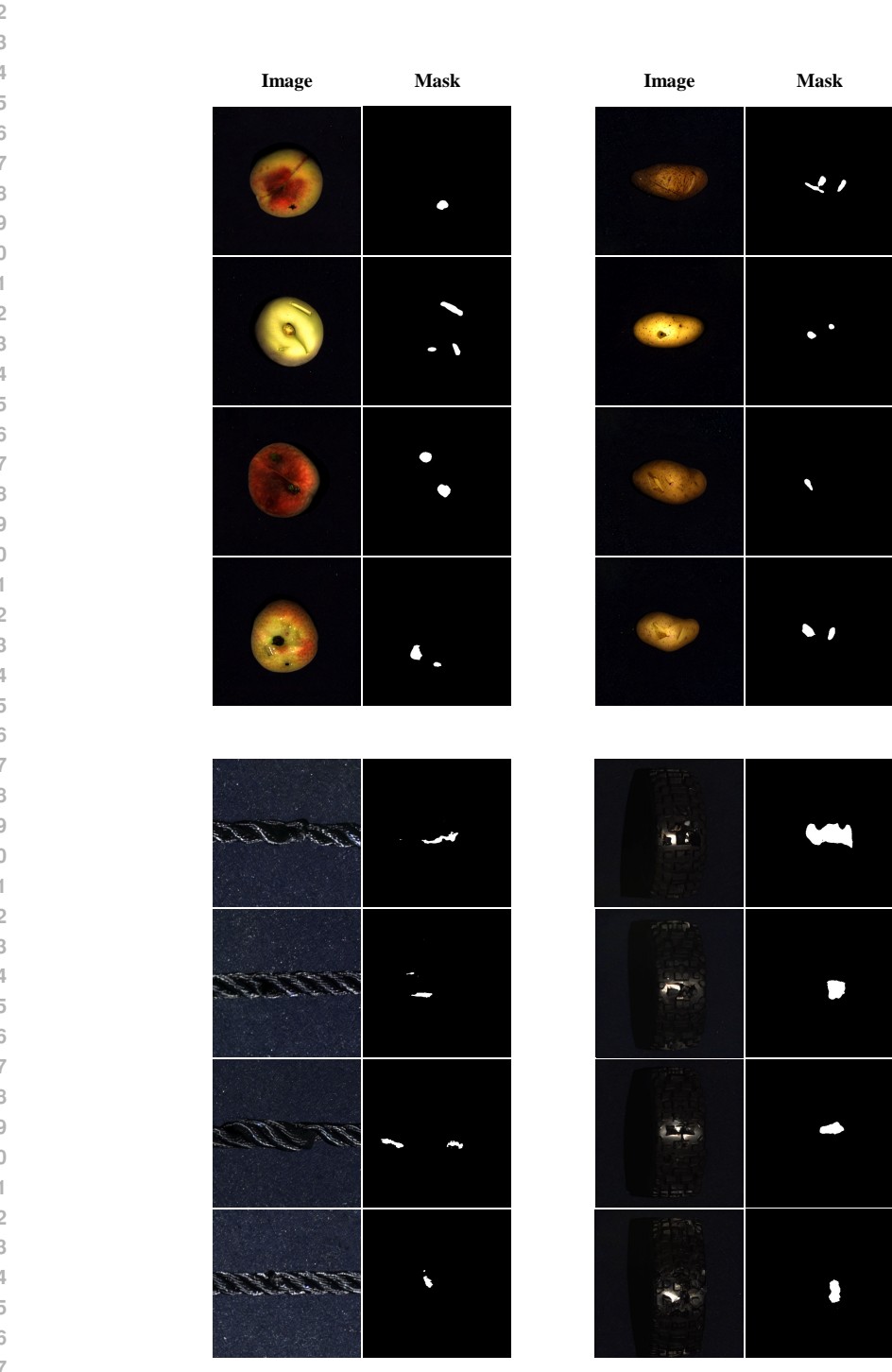

Figure 10: Qualitative results of our anomaly image generation results on MVTec 3D AD. In the first row, from left to right are the results for *peach*, and *potato* categories. In the second row, from left to right are the results for *rope*, and *tire* categories.

**More qualitative segmentation results with different segmentation models**

In this section, we provide further qualitative results with the anomaly segmentation models on the MVTec 3D AD dataset. As shown in Fig. 11, we report the segmentation results of different segmentation models trained on image-mask pairs generated by SeaS.

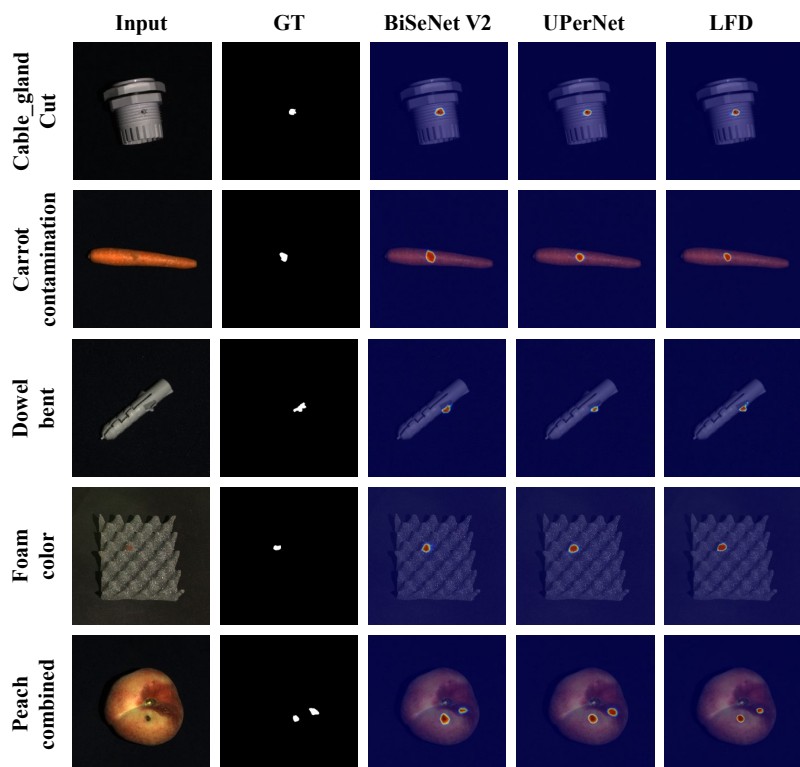

Figure 11: Qualitative comparison results with the anomaly segmentation models on MVTec 3D AD. In the figure, from top to bottom are the results for *cable_gland*, *carrot*, *dowel*, *foam* and *peach* categories.

## A.5 MORE QUANTITATIVE COMPARISON WITH UNSUPERVISED ANOMALY DETECTION METHODS

We compare our method with the state-of-the-art unsupervised anomaly detection methods, i.e., RealNet (Zhang et al., 2024), HVQ-Trans (Lu et al., 2023), DiAD (He et al., 2024), and PRN (Zhang et al., 2023a). The performance is different from the results reported in the paper, since as we mentioned in Sec 4.1, we use 2/3 anomaly images and all good images in the testing set of MVTec AD as the testing set in all the experiments, while the original results are achieved on the whole testing set. As shown in Tab. 12, Real-Net contains 591M parameters, around 177 times larger than BiSeNet V2, and 631 times larger than LFD, while the pixel-level AP and IoU measures of Real-Net are even worse than those of BiSeNet V2 and LFD. Although the pixel-level AUROC metric, which is more sensitive to false negatives than to false positives, is slightly higher for Real-Net, we observe that it generates a high number of false positives, substantially reducing pixel-level AP, $F_1$-max, and IoU scores. The results of UperNet outperform the DiAD method on all measures, despite having only 1/24 of the parameters. For effective industrial anomaly detection, a method must balance false positives and false negatives. The supervised anomaly segmentation models greatly outperform HVQ-Trans (8.45M parameters) and PRN in AP, F1-max measure, and IoU, even though BiSeNet V2 and LFD are much smaller. These comparisons reveal that the small segmentation model achieves good performance using the generated images of SeaS, which is important for practical industrial applications.

Table 12: Comparison with anomaly detection methods on MVTec AD.

| Model | Parameters | Image-level | | | Pixel-level | | | |
|---|---|---|---|---|---|---|---|---|
| | | AUROC | AP | $F_1$-max | AUROC | AP | $F_1$-max | IoU |
| SeaS + BiSeNet V2 (Yu et al., 2021) | 3.341M | 96.00 | 98.14 | 95.43 | 97.21 | 69.21 | 66.37 | 55.28 |
| SeaS + UPerNet (Xiao et al., 2018) | 64.042M | **98.29** | **99.20** | 97.34 | 97.87 | 74.42 | 70.70 | **61.24** |
| SeaS + LFD (Zhou et al., 2024a) | **0.936M** | 95.88 | 97.89 | 95.15 | 98.09 | **77.15** | **72.52** | 56.47 |
| Real-Net (Zhang et al., 2024) | 591M | 98.19 | 98.99 | **97.88** | **98.84** | 68.09 | 66.46 | 53.99 |
| HVQ-Trans (Lu et al., 2023) | 8.45M | 96.38 | 98.09 | 95.30 | 97.60 | 47.95 | 53.32 | 45.03 |
| DiAD (He et al., 2024) | 1525M | 97.20 | 99.00 | 96.50 | 96.80 | 52.60 | 55.50 | - |
| PRN (Zhang et al., 2023a) | - | 91.60 | 96.60 | 92.40 | 96.90 | 66.20 | 64.70 | - |

## A.6 MORE QUALITATIVE AND QUANTITATIVE ANOMALY IMAGE GENERATION RESULTS

**More qualitative generation results**

We provide further qualitative results of every category on the MVTec AD dataset, from Fig. 12 to Fig. 14. We report the anomaly image generation results of SeaS for varying types of anomalies. The first column represents the generated anomaly images, the second column represents the corresponding generated masks, and the third column represents the masks generated without using the Mask Refinement Module.

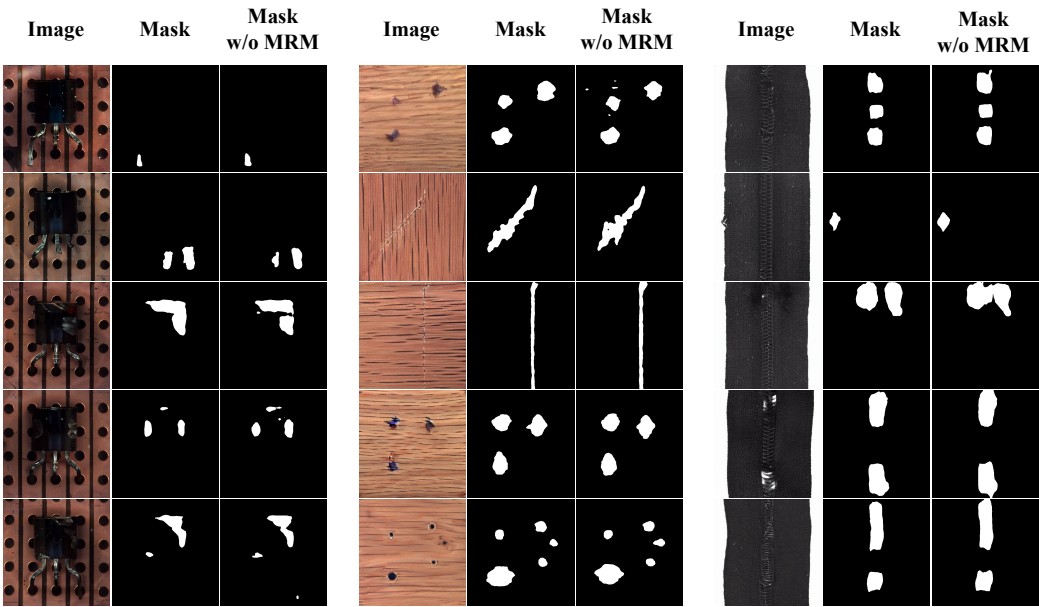

Figure 12: Qualitative results of our anomaly image generation results on MVTec AD. In the first row, from left to right are the results for *transistor*, *wood*, and *zipper* categories.

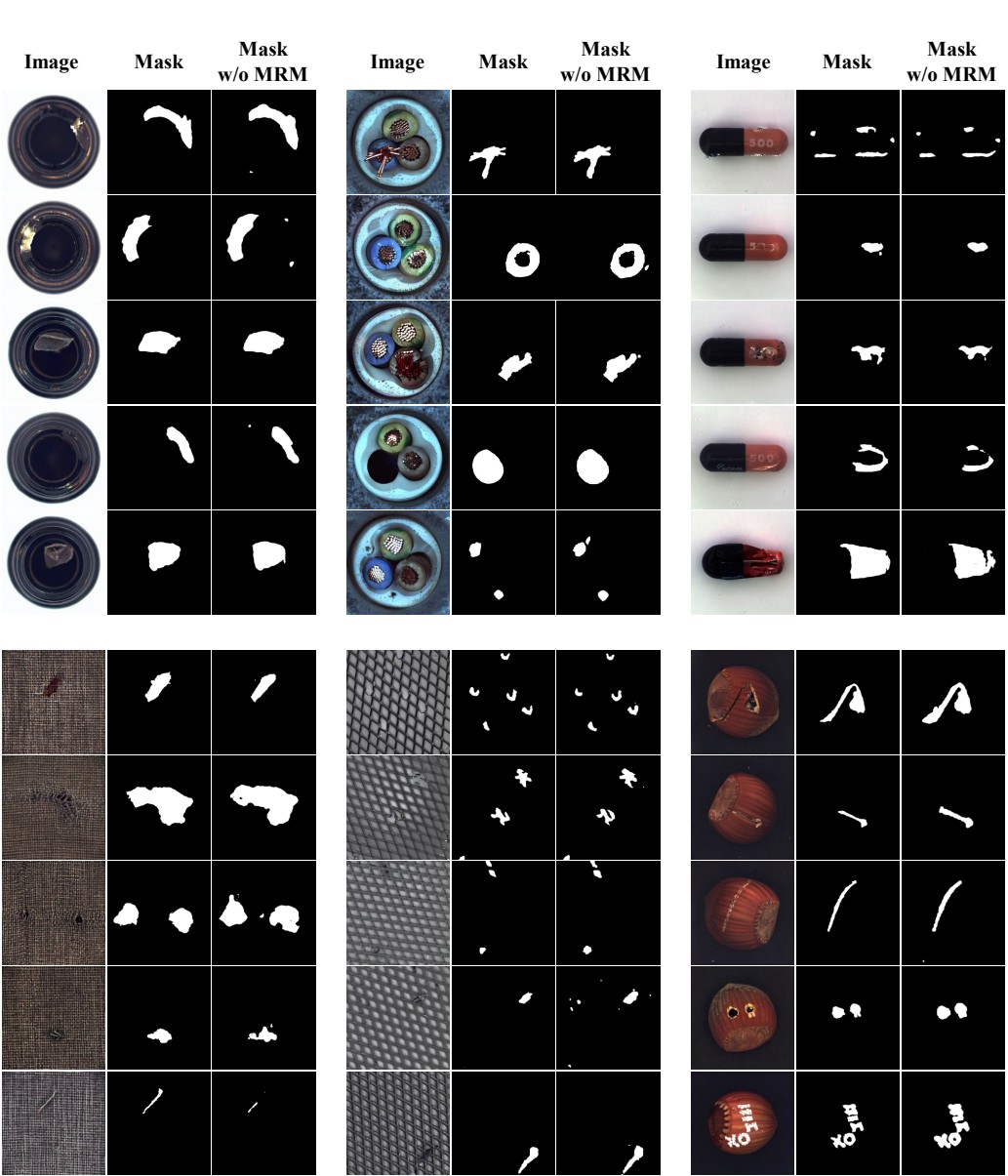

Figure 13: Qualitative results of our anomaly image generation results on MVTec AD. In the first row, from left to right are the results for *bottle*, *cable*, and *capsule* categories. In the second row, from left to right are the results for *carpet*, *grid*, and *hazelnut* categories.

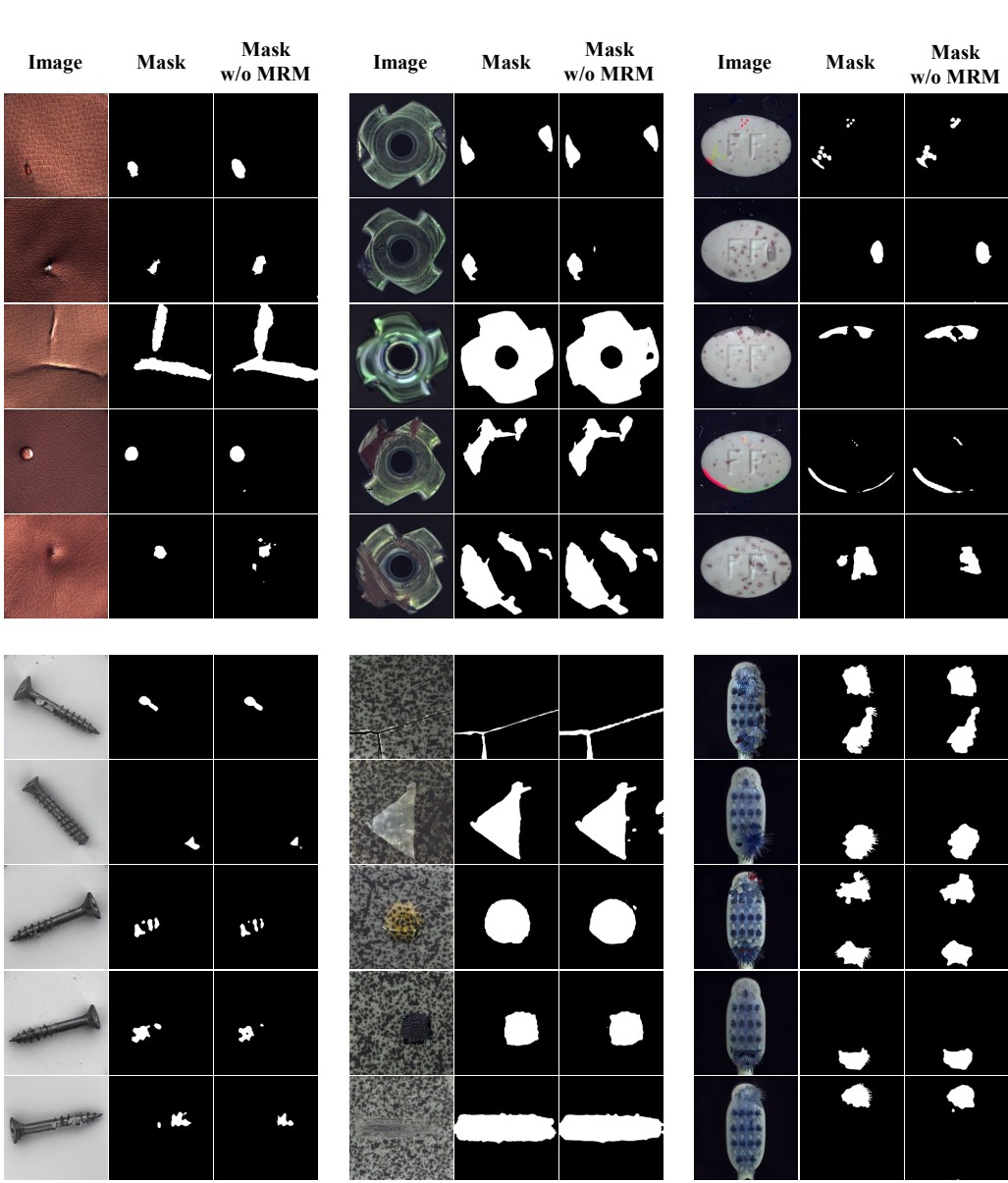

Figure 14: Qualitative results of our anomaly image generation results on MVTec AD. In the first row, from left to right are the results for *leather*, *metal_nut*, and *pill* categories. In the second row, from left to right are the results for *screw*, *tile*, and *toothbrush* categories.

**More quantitative results**

In this section, we report the detailed generation results of SeaS for each category on the MVTec AD datasets, compared with DRAEM (Zavrtanik et al., 2021), DFMGAN (Duan et al., 2023) and AnomalyDiffusion (Hu et al., 2024) which are presented from Tab. 13 to Tab. 18

Table 13: Comparison on anomaly segmentation on BiSeNet V2.

| Category | DRAEM | | | | DFMGAN | | | | AnomalyDiffusion | | | | Ours | | | |
|---|---|---|---|---|---|---|---|---|---|---|---|---|---|---|---|---|
| | AUROC | AP | $F_1$-max | IoU | AUROC | AP | $F_1$-max | IoU | AUROC | AP | $F_1$-max | IoU | AUROC | AP | $F_1$-max | IoU |
| bottle | 77.42 | 45.25 | 48.21 | 45.77 | 89.34 | 64.67 | 62.78 | 44.71 | 99.00 | 88.02 | 80.53 | 68.25 | 99.46 | 93.43 | 85.59 | 75.86 |
| cable | 65.28 | 14.73 | 23.09 | 19.44 | 93.87 | 67.98 | 64.74 | 44.02 | 92.84 | 69.86 | 66.32 | 46.49 | 89.85 | 72.07 | 71.58 | 53.24 |
| capsule | 63.71 | 13.31 | 20.00 | 37.88 | 74.88 | 16.43 | 23.01 | 29.97 | 92.71 | 38.11 | 40.67 | 19.44 | 86.33 | 24.64 | 30.54 | 39.70 |
| carpet | 97.27 | 69.99 | 68.28 | 54.31 | 94.53 | 42.53 | 47.44 | 39.88 | 98.65 | 73.10 | 65.83 | 43.25 | 99.61 | 82.30 | 72.94 | 55.52 |
| grid | 93.86 | 25.36 | 35.46 | 35.57 | 96.86 | 24.40 | 37.40 | 29.93 | 80.59 | 8.08 | 16.79 | 14.26 | 99.36 | 37.91 | 42.50 | 39.80 |
| hazelnut | 77.48 | 41.52 | 48.36 | 54.58 | 99.87 | 96.75 | 90.07 | 71.68 | 97.71 | 63.34 | 59.87 | 43.12 | 97.82 | 78.55 | 73.09 | 68.47 |
| leather | 99.76 | 67.02 | 64.96 | 52.37 | 97.50 | 51.10 | 52.26 | 50.67 | 99.30 | 57.49 | 59.62 | 43.94 | 98.91 | 59.84 | 58.62 | 45.82 |
| metal_nut | 73.26 | 32.26 | 32.77 | 48.68 | 99.39 | 97.59 | 92.52 | 70.40 | 99.03 | 95.67 | 88.69 | 58.8 | 99.69 | 98.29 | 93.23 | 74.40 |
| pill | 60.02 | 9.33 | 17.17 | 11.67 | 97.09 | 83.98 | 79.26 | 36.39 | 99.44 | 93.16 | 86.62 | 41.18 | 98.31 | 76.97 | 68.00 | 55.43 |
| screw | 82.23 | 17.78 | 24.08 | 22.15 | 97.94 | 37.10 | 41.01 | 31.63 | 94.08 | 17.95 | 25.90 | 20.00 | 97.64 | 40.20 | 45.35 | 38.43 |
| tile | 98.09 | 82.9 | 76.43 | 63.48 | 99.65 | 97.08 | 91.16 | 75.94 | 97.79 | 85.58 | 78.28 | 60.46 | 99.67 | 97.29 | 91.48 | 75.75 |
| toothbrush | 92.65 | 36.73 | 45.92 | 23.90 | 99.70 | 51.32 | 54.05 | 23.38 | 98.43 | 49.64 | 54.08 | 26.53 | 97.15 | 46.09 | 49.02 | 28.56 |
| transistor | 62.48 | 14.83 | 20.57 | 21.85 | 84.31 | 45.34 | 46.07 | 30.00 | 98.85 | 85.27 | 77.95 | 49.83 | 96.75 | 69.52 | 66.11 | 57.24 |
| wood | 92.89 | 70.82 | 68.34 | 58.05 | 98.32 | 64.82 | 63.11 | 58.99 | 96.78 | 63.38 | 60.31 | 45.73 | 98.38 | 80.81 | 74.03 | 56.22 |
| zipper | 84.18 | 41.68 | 45.65 | 41.08 | 97.29 | 65.18 | 63.24 | 49.93 | 98.81 | 78.89 | 72.66 | 62.03 | 99.23 | 80.27 | 73.41 | 64.80 |
| Average | 81.37 | 38.90 | 42.62 | 39.39 | 94.57 | 60.42 | 60.54 | 45.83 | 96.27 | 64.5 | 62.27 | 42.89 | 97.21 | 69.21 | 66.37 | 55.28 |

Table 14: Comparison on image-level anomaly detection on BiSeNet V2.

| Category | DRAEM | | | DFMGAN | | | AnomalyDiffusion | | | Ours | | |
|---|---|---|---|---|---|---|---|---|---|---|---|---|
| | AUROC | AP | $F_1$-max | AUROC | AP | $F_1$-max | AUROC | AP | $F_1$-max | AUROC | AP | $F_1$-max |
| bottle | 95.93 | 98.21 | 93.18 | 96.74 | 98.75 | 95.35 | 98.14 | 99.34 | 97.67 | 100.00 | 100.00 | 100.00 |
| cable | 80.79 | 83.53 | 79.22 | 79.47 | 85.00 | 74.13 | 95.37 | 96.71 | 92.91 | 94.61 | 96.39 | 89.83 |
| capsule | 91.88 | 97.62 | 92.41 | 85.51 | 95.16 | 89.82 | 84.06 | 95.01 | 89.74 | 88.81 | 96.92 | 89.21 |
| carpet | 98.21 | 99.24 | 95.31 | 91.42 | 96.29 | 88.89 | 90.55 | 96.41 | 90.32 | 98.16 | 99.31 | 97.56 |
| grid | 96.43 | 98.50 | 95.00 | 99.64 | 99.82 | 97.56 | 81.19 | 89.92 | 83.95 | 99.17 | 99.63 | 98.73 |
| hazelnut | 97.92 | 98.65 | 94.62 | 100.00 | 100.00 | 100.00 | 93.39 | 95.74 | 90.91 | 100.00 | 100.00 | 100.00 |
| leather | 100.00 | 100.00 | 100.00 | 98.31 | 99.23 | 95.24 | 100.00 | 100.00 | 100.00 | 95.83 | 98.38 | 95.93 |
| metal_nut | 96.38 | 99.00 | 96.83 | 97.37 | 99.16 | 94.66 | 99.01 | 99.66 | 97.71 | 100.00 | 100.00 | 100.00 |
| pill | 74.68 | 89.88 | 89.00 | 84.86 | 95.27 | 91.00 | 90.38 | 97.43 | 91.35 | 96.59 | 99.12 | 95.24 |
| screw | 71.15 | 83.52 | 83.15 | 74.95 | 85.50 | 80.72 | 58.18 | 75.32 | 81.25 | 77.24 | 89.55 | 80.60 |
| tile | 99.68 | 99.82 | 98.28 | 99.47 | 99.74 | 99.12 | 99.44 | 99.39 | 98.50 | 100.00 | 100.00 | 100.00 |
| toothbrush | 82.50 | 90.00 | 85.11 | 78.33 | 87.73 | 83.72 | 78.33 | 89.26 | 79.17 | 90.42 | 94.49 | 89.47 |
| transistor | 73.87 | 68.65 | 60.87 | 79.52 | 75.77 | 69.57 | 94.40 | 94.68 | 94.34 | 99.23 | 98.39 | 94.92 |
| wood | 97.24 | 98.99 | 96.30 | 98.87 | 99.46 | 97.67 | 90.48 | 94.12 | 93.33 | 100.00 | 100.00 | 100.00 |
| zipper | 91.31 | 97.01 | 90.24 | 98.97 | 99.64 | 97.56 | 98.89 | 99.62 | 97.56 | 100.00 | 100.00 | 100.00 |
| Average | 89.87 | 93.51 | 89.97 | 90.90 | 94.43 | 90.33 | 90.08 | 94.84 | 91.84 | 96.00 | 98.14 | 95.43 |

Table 15: Comparison on anomaly segmentation on UPerNet.

| Category | DRAEM | | | | DFMGAN | | | | AnomalyDiffusion | | | | Ours | | | |
|---|---|---|---|---|---|---|---|---|---|---|---|---|---|---|---|---|
| | AUROC | AP | $F_1$-max | IoU | AUROC | AP | $F_1$-max | IoU | AUROC | AP | $F_1$-max | IoU | AUROC | AP | $F_1$-max | IoU |
| bottle | 82.87 | 51.45 | 52.25 | 54.58 | 87.94 | 56.89 | 56.56 | 45.41 | 99.54 | 93.01 | 85.94 | 75.31 | 99.28 | 91.73 | 84.53 | 78.73 |
| cable | 56.64 | 13.59 | 22.42 | 16.29 | 87.52 | 64.30 | 65.61 | 41.02 | 91.00 | 68.12 | 67.49 | 51.84 | 91.08 | 76.25 | 74.63 | 59.00 |
| capsule | 60.95 | 11.12 | 18.34 | 27.18 | 67.92 | 12.31 | 20.32 | 30.47 | 97.64 | 51.90 | 51.66 | 37.00 | 92.09 | 39.60 | 43.89 | 50.18 |
| carpet | 98.83 | 79.57 | 72.53 | 61.50 | 95.85 | 36.05 | 34.52 | 48.10 | 99.45 | 82.13 | 72.55 | 53.17 | 99.67 | 82.01 | 73.53 | 60.60 |
| grid | 95.66 | 36.69 | 43.55 | 36.52 | 97.49 | 29.67 | 36.15 | 31.37 | 94.22 | 28.97 | 38.50 | 32.93 | 99.18 | 44.94 | 48.28 | 44.21 |
| hazelnut | 77.69 | 41.57 | 47.13 | 58.88 | 99.36 | 79.76 | 71.10 | 72.90 | 97.77 | 70.48 | 67.93 | 54.47 | 99.54 | 81.84 | 75.48 | 73.30 |
| leather | 99.70 | 65.49 | 63.01 | 62.20 | 80.97 | 17.60 | 26.21 | 30.17 | 99.48 | 63.46 | 60.54 | 48.70 | 99.42 | 68.26 | 65.52 | 57.01 |
| metal_nut | 65.37 | 23.26 | 27.39 | 42.56 | 98.44 | 95.64 | 91.48 | 64.92 | 98.62 | 95.11 | 88.62 | 61.31 | 99.70 | 98.33 | 92.90 | 76.07 |
| pill | 64.46 | 11.33 | 20.28 | 13.10 | 97.58 | 83.74 | 80.02 | 42.33 | 99.33 | 95.04 | 88.77 | 49.18 | 98.59 | 81.16 | 74.26 | 62.62 |
| screw | 90.88 | 23.64 | 31.49 | 24.71 | 97.49 | 53.83 | 53.02 | 42.05 | 93.89 | 36.60 | 42.68 | 34.08 | 98.97 | 52.02 | 51.65 | 46.61 |
| tile | 96.25 | 79.31 | 74.18 | 66.79 | 99.79 | 97.29 | 91.11 | 77.46 | 94.70 | 73.34 | 67.79 | 58.54 | 99.67 | 95.89 | 90.71 | 77.89 |
| toothbrush | 93.86 | 46.93 | 58.92 | 26.76 | 97.42 | 51.09 | 59.23 | 28.33 | 97.52 | 60.67 | 59.46 | 33.98 | 98.50 | 63.62 | 63.07 | 42.09 |
| transistor | 78.20 | 26.52 | 30.34 | 26.07 | 82.07 | 36.31 | 39.48 | 27.44 | 94.26 | 73.68 | 69.50 | 53.64 | 93.88 | 70.37 | 68.12 | 56.98 |
| wood | 95.03 | 77.07 | 74.07 | 64.67 | 97.90 | 69.02 | 62.21 | 63.10 | 96.09 | 70.10 | 64.38 | 51.44 | 99.28 | 85.28 | 76.28 | 65.09 |
| zipper | 91.74 | 54.11 | 53.69 | 48.57 | 97.28 | 71.60 | 66.64 | 54.54 | 99.54 | 86.18 | 78.50 | 66.47 | 99.17 | 85.01 | 77.57 | 68.21 |
| Average | 83.21 | 42.78 | 45.97 | 42.03 | 92.33 | 57.01 | 56.91 | 46.64 | 96.87 | 69.92 | 66.95 | 50.80 | 97.87 | 74.42 | 70.70 | 61.24 |

Table 16: Comparison on image-level anomaly detection on UPerNet.

| Category | DRAEM | | | DFMGAN | | | AnomalyDiffusion | | | Ours | | |
|---|---|---|---|---|---|---|---|---|---|---|---|---|
| | AUROC | AP | $F_1$-max | AUROC | AP | $F_1$-max | AUROC | AP | $F_1$-max | AUROC | AP | $F_1$-max |
| bottle | 98.49 | 99.41 | 97.62 | 94.19 | 97.86 | 93.18 | **100.00** | **100.00** | **100.00** | **100.00** | **100.00** | **100.00** |
| cable | 73.06 | 77.88 | 71.94 | 85.64 | 90.03 | 80.33 | **95.58** | **97.06** | **92.56** | 94.40 | 96.38 | 92.44 |
| capsule | 85.62 | 95.66 | 89.02 | 81.04 | 94.26 | 87.01 | **96.00** | **98.77** | **95.48** | 94.43 | 98.44 | 92.21 |
| carpet | 97.64 | 99.00 | 95.08 | 96.72 | 98.58 | 93.75 | 98.68 | 99.53 | 98.36 | **99.94** | **99.97** | **99.20** |
| grid | 97.62 | 98.99 | 97.44 | 98.33 | 99.13 | 96.30 | 96.67 | 98.73 | 97.44 | **99.76** | **99.88** | **98.73** |
| hazelnut | 97.14 | 97.74 | 92.63 | 99.84 | 99.87 | 97.96 | 99.17 | 99.43 | 97.87 | **100.00** | **100.00** | **100.00** |
| leather | **100.00** | **100.00** | **100.00** | 79.91 | 90.70 | 81.75 | **100.00** | **100.00** | **100.00** | 100.00 | 100.00 | 100.00 |
| metal_nut | 86.79 | 96.02 | 87.39 | 98.30 | 99.38 | 97.71 | 98.65 | 99.62 | 98.41 | **99.72** | **99.91** | **99.21** |
| pill | 67.07 | 87.37 | 88.07 | 88.54 | 96.56 | 92.39 | 91.23 | 97.78 | 90.91 | **98.28** | **99.58** | **97.92** |
| screw | 80.04 | 90.95 | 81.03 | 89.01 | 94.54 | 88.24 | 85.06 | 93.87 | 85.33 | **93.47** | **97.07** | **90.45** |
| tile | 99.20 | 99.51 | 98.25 | 99.68 | 99.81 | 99.13 | 99.68 | 99.81 | 99.13 | **100.00** | **100.00** | **100.00** |
| toothbrush | 86.67 | 94.19 | 88.89 | 75.00 | 86.99 | 80.00 | 90.00 | 95.13 | 90.00 | **95.00** | **97.65** | **94.74** |
| transistor | 79.29 | 74.68 | 69.09 | 83.04 | 73.59 | 74.19 | **100.00** | **100.00** | **100.00** | 99.52 | 99.16 | 96.43 |
| wood | 98.25 | 99.29 | 96.47 | 93.36 | 95.60 | 95.45 | 98.62 | 99.49 | 97.62 | **99.87** | **99.94** | **98.82** |
| zipper | 94.86 | 98.13 | 92.02 | 98.48 | 99.51 | 98.14 | **100.00** | **100.00** | **100.00** | 100.00 | 100.00 | 100.00 |
| Average | 89.45 | 93.92 | 89.66 | 90.74 | 94.43 | 90.37 | 96.62 | 98.61 | 96.21 | **98.29** | **99.20** | **97.34** |

Table 17: Comparison on anomaly segmentation on LFD.

| Category | DRAEM | | | | DFMGAN | | | | AnomalyDiffusion | | | | Ours | | | |
|---|---|---|---|---|---|---|---|---|---|---|---|---|---|---|---|---|
| | AUROC | AP | $F_1$-max | IoU | AUROC | AP | $F_1$-max | IoU | AUROC | AP | $F_1$-max | IoU | AUROC | AP | $F_1$-max | IoU |
| bottle | 78.76 | 50.40 | 51.05 | 39.74 | 90.41 | 61.51 | 58.49 | 40.19 | 98.71 | 89.64 | 81.55 | 67.10 | **99.28** | **92.65** | **84.86** | **73.82** |
| cable | 67.64 | 19.41 | 25.36 | 19.19 | 96.49 | 79.40 | **75.25** | 53.47 | **97.89** | **79.85** | 72.75 | 53.69 | 94.53 | 75.41 | 72.70 | **55.98** |
| capsule | 88.48 | 34.60 | 39.62 | 31.48 | 91.82 | **56.11** | **58.56** | 32.50 | 95.80 | 38.17 | 48.92 | 32.04 | 91.80 | 49.76 | 53.69 | **41.14** |
| carpet | 83.51 | 37.69 | 41.86 | 41.22 | 89.10 | 48.04 | 49.89 | 39.46 | 94.83 | 53.15 | 51.79 | 42.21 | **99.10** | **82.74** | **74.51** | **57.56** |
| grid | 92.13 | 45.75 | 48.84 | 27.66 | 89.18 | 34.89 | 41.21 | 19.21 | 85.19 | 24.32 | 34.76 | 18.22 | **98.78** | **62.24** | **58.44** | **41.69** |
| hazelnut | 59.97 | 28.87 | 37.82 | 30.38 | **99.36** | **95.16** | **89.80** | **76.43** | 98.54 | 77.39 | 70.42 | 45.97 | 98.97 | 88.00 | 81.77 | 73.39 |
| leather | 97.38 | 66.36 | 63.43 | 53.13 | 97.82 | 51.86 | 52.25 | 48.09 | 98.99 | 65.73 | 62.85 | 42.65 | **99.11** | **76.49** | **69.30** | **56.51** |
| metal_nut | 63.34 | 35.42 | 35.24 | 47.51 | 98.16 | 95.16 | 90.99 | 63.02 | **99.38** | **97.34** | **91.63** | 64.59 | 99.23 | 96.66 | 91.42 | **75.15** |
| pill | 36.18 | 8.93 | 12.79 | 13.62 | 95.80 | 75.90 | 70.31 | 31.73 | **98.96** | **92.51** | **85.35** | 50.04 | 98.11 | 79.63 | 72.54 | **56.73** |
| screw | 91.03 | 27.05 | 32.95 | 19.03 | 93.96 | 38.00 | 41.69 | 30.88 | 92.68 | 44.64 | 49.17 | 34.08 | **98.27** | **52.40** | **52.32** | **41.02** |
| tile | 91.77 | 80.02 | 77.19 | 56.27 | 97.37 | 88.79 | 82.05 | 66.30 | 92.98 | 79.59 | 73.52 | 55.08 | **99.38** | **96.24** | **89.90** | **75.50** |
| toothbrush | 55.94 | 24.87 | 35.54 | 12.75 | 95.17 | 55.21 | 53.95 | 28.83 | **98.31** | **68.60** | **66.14** | **29.67** | 96.97 | 54.84 | 53.19 | 27.91 |
| transistor | 55.81 | 19.36 | 24.38 | 32.58 | 97.68 | **89.68** | **84.18** | 46.98 | 98.20 | 83.97 | 75.84 | 44.22 | **98.80** | 84.32 | 77.02 | **55.57** |
| wood | 90.04 | 73.42 | 71.25 | 59.55 | 97.47 | 77.72 | 70.91 | 58.77 | 95.68 | 67.54 | 63.06 | 42.78 | **98.60** | **88.57** | **81.46** | **62.94** |
| zipper | 94.15 | 62.65 | 61.39 | 50.12 | 93.80 | 58.43 | 56.82 | 46.44 | 98.42 | 84.05 | 77.08 | 64.14 | **99.15** | **86.67** | **79.09** | **69.37** |
| Average | 76.41 | 40.99 | 43.91 | 35.61 | 94.91 | 67.06 | 65.09 | 45.49 | 96.30 | 69.77 | 66.99 | 45.77 | **98.01** | **77.77** | **72.81** | **57.62** |

Table 18: Comparison on image-level anomaly detection on LFD.

| Category | DRAEM | | | DFMGAN | | | AnomalyDiffusion | | | Ours | | |
|---|---|---|---|---|---|---|---|---|---|---|---|---|
| | AUROC | AP | $F_1$-max | AUROC | AP | $F_1$-max | AUROC | AP | $F_1$-max | AUROC | AP | $F_1$-max |
| bottle | 96.40 | 98.56 | 94.12 | 96.98 | 98.76 | 95.35 | **100.00** | **100.00** | **100.00** | 100.00 | 100.00 | 100.00 |
| cable | 70.91 | 78.06 | 71.52 | 90.98 | 94.21 | 88.14 | **99.52** | **99.55** | **97.71** | 92.05 | 94.95 | 88.70 |
| capsule | 85.80 | 95.77 | 87.80 | 86.32 | 95.99 | 88.46 | 83.25 | 94.62 | 89.44 | **93.80** | **98.19** | **93.42** |
| carpet | 81.28 | 92.34 | 84.67 | 88.02 | 95.33 | 87.60 | 86.00 | 93.42 | 87.22 | **97.98** | **99.22** | **96.67** |
| grid | **97.14** | 98.63 | 92.86 | 85.48 | 92.61 | 85.71 | 93.69 | 97.08 | 91.14 | 96.79 | **98.76** | **96.10** |
| hazelnut | 85.73 | 89.00 | 81.72 | 99.90 | 99.91 | 98.97 | 98.28 | 98.60 | 95.83 | **100.00** | **100.00** | **100.00** |
| leather | 98.81 | 99.28 | 98.44 | 95.93 | 98.15 | 93.65 | 99.90 | 99.90 | 99.20 | **100.00** | **100.00** | **100.00** |
| metal_nut | 94.67 | 98.39 | 93.44 | 96.16 | 98.57 | 96.18 | **99.01** | **99.65** | **98.46** | 98.58 | 99.54 | 97.64 |
| pill | 66.39 | 89.29 | 88.07 | 82.85 | 94.40 | 92.00 | 94.15 | 98.42 | 94.47 | **98.16** | **99.50** | **96.84** |
| screw | 73.92 | 85.38 | 82.72 | 82.60 | 92.15 | 82.22 | 81.54 | 91.32 | 82.05 | **87.83** | **94.39** | **85.54** |
| tile | 95.80 | 97.49 | 93.33 | 98.94 | 99.43 | 96.55 | 98.25 | 99.13 | 95.65 | **99.36** | **99.69** | **99.12** |
| toothbrush | 83.33 | 90.29 | 84.44 | 77.08 | 87.68 | 80.95 | **100.00** | **100.00** | **100.00** | 87.92 | 94.08 | 87.80 |
| transistor | 90.06 | 89.06 | 81.48 | 88.04 | 85.06 | 77.78 | 97.38 | 96.57 | 92.86 | **98.10** | **96.90** | **94.55** |
| wood | 99.50 | 99.78 | 97.62 | 99.87 | 99.94 | 98.82 | 97.24 | 98.70 | 96.47 | **100.00** | **100.00** | **100.00** |
| zipper | 99.39 | 99.77 | 97.56 | 97.07 | 98.78 | 96.25 | 99.01 | 99.71 | 99.39 | **100.00** | **100.00** | **100.00** |
| Average | 87.94 | 93.41 | 88.65 | 91.08 | 95.40 | 90.58 | 95.15 | 97.78 | 94.66 | **96.70** | **98.35** | **95.76** |

## A.7 MORE DETAILS OF THE SEGMENTATION MODELS

As mentioned in the experiment part, we choose three segmentation models (BiSeNet V2 (Yu et al., 2021), UPerNet (Xiao et al., 2018), LFD (Zhou et al., 2024a)) to verify the validity of the generated image-mask pairs on the downstream anomaly segmentation as well as detection tasks. **For BiSeNet V2 and UPerNet, we generally follow the implementation provided by MMsegmentation. For LFD, we also use the official implementation.**

Specifically, for BiSeNet V2, we choose a backbone structure of a detail branch of three stages with 64, 64 and 128 channels and a semantic branch of four stages with 16, 32, 64 and 128 channels respectively, with a decode head and four auxiliary heads (corresponding to the number of stages in the semantic branch). As for UPerNet, we choose ResNet-50 as the backbone, with a decode head and an auxiliary head.

**In training segmentation models for downstream tasks, we adopt a training strategy of training a unified segmentation model for all classes of products, rather than training separate segmentation models for each class.** Experimental results are shown in Tab. 19, which indicate that the performance of the unified segmentation model surpasses that of multiple individual segmentation models.

Table 19: Ablation on the training strategy of segmentation models.

| Models | Multiple Models | | | | Unified Model | | | |
|---|---|---|---|---|---|---|---|---|
| | AUROC | AP | $F_1$-max | IoU | AUROC | AP | $F_1$-max | IoU |
| BiSeNet V2 | 96.00 | 67.68 | 65.87 | 54.11 | 97.21 | 69.21 | 66.37 | 55.28 |
| UPerNet | 96.77 | 73.88 | 70.49 | 60.37 | 97.87 | 74.42 | 70.70 | 61.24 |
| LFD | 93.02 | 72.97 | 71.56 | 55.88 | 98.09 | 77.15 | 72.52 | 56.47 |
| Average | 95.26 | 71.51 | 69.31 | 56.79 | **97.72** | **73.59** | **69.86** | **57.66** |

## A.8 MORE ABLATION STUDIES

**Ablation on the Unbalanced Abnormal Text Prompt design**

In the design of the prompt for industrial anomaly image generation, we conduct experiments to validate the effectiveness of our Unbalanced Abnormal (UA) Text Prompt for each anomaly type of each product. We set the number of learnable $<\text{df}_n>$ to $N$, and the number of learnable $<\text{ob}_j>$ to $N'$. As shown in Tab. 20, by utilizing the UA Text Prompt, i.e.,

$$\mathcal{P} = \texttt{a <ob> with <df}_1\texttt{>,<df}_2\texttt{>, <df}_3\texttt{>, <df}_4\texttt{>}$$

we are able to provide high-fidelity and diverse images for downstream anomaly segmentation tasks, resulting in the best performance in segmentation metrics.

**Ablation on the Separation and Sharing Fine-tuning loss**

In the design of the DA loss and NA loss for the Separation and Sharing Fine-tuning, we conduct two sets of experiments: (a) We remove the second term in the DA loss (short for w/o ST in Tab. 21); (b) We replace the second term in DA loss with another term in the NA loss (short for with AT in Tab. 21), which aligns the background area with the token $<\text{ob}>$ according to the mask:

$$\mathcal{L}_{\mathbf{ob}} = \sum_{l=1}^{L}(||A_{\text{ob}}^l - (1 - M^l)||^2) + ||\epsilon_{\text{ob}} - \epsilon_\theta(\hat{z}_{\text{ob}}, t_{\text{ob}}, \mathbf{e}_{\text{ob}})||_2^2 \tag{9}$$

where $A_{\text{ob}}^l \in \mathbb{R}^{r \times r \times 1}$ is the cross-attention map corresponding to the normal token $<\text{ob}>$. As shown in Tab. 21, the experimental results demonstrate that, our adopted loss design achieves the best performance in downstream segmentation tasks.

Table 20: Ablation on the Unbalanced Abnormal Text Prompt design.

| Prompt | AUROC | AP | $F_1$-max | IoU |
|---|---|---|---|---|
| $N' = 1, N = 1$ | 96.48 | 63.69 | 62.50 | 52.02 |
| $N' = 1, N = 4$ **(Ours)** | **97.21** | **69.21** | **66.37** | **55.28** |
| $N' = 4, N = 4$ | 96.55 | 66.28 | 63.95 | 54.07 |

Table 21: Ablation on the Separation and Sharing Fine-tuning loss.

| Loss | AUROC | AP | $F_1$-max | IoU |
|---|---|---|---|---|
| w/o ST | 96.44 | 67.73 | 65.23 | 54.99 |
| with AT | 96.42 | 63.99 | 62.43 | 53.36 |
| **Ours** | **97.21** | **69.21** | **66.37** | **55.28** |

**Ablation on the minimum size requirement for training images**

In the few-shot setting, for a fair comparison, we follow the common setting in DFMGAN (Duan et al., 2023) and AnomalyDiffusion (Hu et al., 2024), i.e., using one-third abnormal image-mask pairs for each anomaly type in training. In this setting, the minimum number of abnormal training images is 2. Once we adopt a 3-shot setting, we need to reorganize the test set. To ensure that the test set is not reorganized for fair comparison, we take 1-shot and 2-shot settings for all anomaly types during training, i.e., $H = 1$ and $H = 2$, where $H$ is the image number. The results are shown in Tab. 22 and Fig. 15. Observably, the models trained by 1-shot and 2-shot settings still generate anomaly images with decent diversity and authenticity.

Table 22: Ablation on the minimum size requirement for training images.

| Size | IS | IC-L |
|---|---|---|
| $H = 1$ | 1.790 | 0.311 |
| $H = 2$ | 1.794 | 0.314 |
| $H = \frac{1}{3} \times H_0$ | **1.876** | **0.339** |

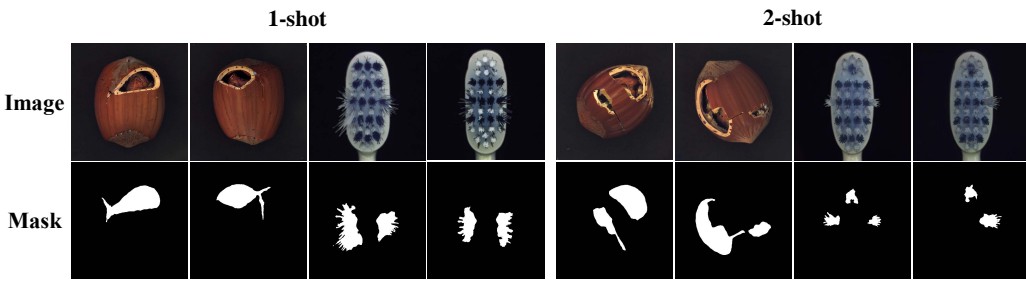

Figure 15: Visualization of the ablation study on the minimum size requirement for training images. In the figure, the first row is for generated images, the second row is for generated masks.

**Ablation on the training strategy of SeaS**

During each step of the fine-tuning process, we sample the same number of images from the abnormal training set $X_{\text{df}}$ and the normal training set $X_{\text{ob}}$. To investigate the efficacy of this strategy, we conduct three distinct sets of experiments: (a) prioritizing training with abnormal images followed by normal images (short for Abnormal-Normal in Tab. 23); (b) prioritizing training with abnormal images followed by anomaly images (short for Normal-Abnormal in Tab. 23); (c) training with a mix of both normal and abnormal images in each batch (short for Abnormal&Normal in Tab. 23). As shown in Tab. 23, SeaS yields superior performance in anomaly image generation, characterized by both high fidelity and diversity in the generated images.

**Ablation on the cross-attention maps for Decoupled Anomaly Alignment**

In Decoupled Anomaly Alignment (DA) loss, we leverage cross-attention maps from various layers of the U-Net encoder. Specifically, we investigate the impact of integrating different cross-attention maps, denoted as $A^1 \in \mathbb{R}^{64 \times 64}$, $A^2 \in \mathbb{R}^{32 \times 32}$, $A^3 \in \mathbb{R}^{16 \times 16}$ and $A^4 \in \mathbb{R}^{8 \times 8}$. These correspond to the cross-attention maps of the "down-1", "down-2", "down-3", and "down-4" layers of

Table 23: Ablation on training strategy of SeaS.

| Strategy | IS | IC-L |
|---|---|---|
| Abnormal-Normal | 1.53 | 0.28 |
| Normal-Abnormal | 1.70 | 0.32 |
| Abnormal&Normal (**Ours**) | **1.88** | **0.34** |

the encoder in U-Net respectively. As shown in Tab. 24, the experimental results demonstrate that, employing a combination of $\{A^2, A^3\}$ for DA loss, achieves the best performance in downstream segmentation tasks.

**Ablation on the features for Coarse Feature Extraction**

In the coarse feature extraction process, we extract coarse but highly-discriminative features for anomalies from U-Net decoder. Specifically, we investigate the impact of integrating different features, denoted as $F_1 \in \mathbb{R}^{16 \times 16 \times 1280}$, $F_2 \in \mathbb{R}^{32 \times 32 \times 1280}$, $F_3 \in \mathbb{R}^{64 \times 64 \times 640}$ and $F_4 \in \mathbb{R}^{64 \times 64 \times 320}$. These correspond to the output feature "up-1", "up-2", "up-3", and "up-4" layers of the encoder in U-Net respectively. As shown in Tab. 25, the experimental results demonstrate that, employing a combination of $\{F_2, F_3\}$ for coarse feature extraction, achieves the best performance in downstream segmentation task.

Table 24: Ablation on the cross-attention maps for Decoupled Anomaly Alignment.

| $A^l$ | AUROC | AP | $F_1$-max | IoU |
|---|---|---|---|---|
| $l = 1, 2, 3$ | 96.42 | 68.92 | 66.24 | 54.52 |
| $l = 2, 3, 4$ | 95.71 | 64.51 | 62.33 | 52.46 |
| $l = 2, 3$ (**Ours**) | **97.21** | **69.21** | **66.37** | **55.28** |

Table 25: Ablation on the features for Coarse Feature Extraction.

| $F_y$ | AUROC | AP | $F_1$-max | IoU |
|---|---|---|---|---|
| $y = 1, 2, 3$ | 94.35 | 63.58 | 60.54 | 52.36 |
| $y = 2, 3, 4$ | 96.93 | 67.42 | 64.26 | **55.31** |
| $y = 2, 3$ (**Ours**) | **97.21** | **69.21** | **66.37** | 55.28 |

**Ablation on the features of VAE for Refined Mask Prediction**

In the Refined Mask Prediction, we combine the high-resolution features of VAE decoder features with discriminative features from U-Net, to generate accurately aligned anomaly image-mask pairs. In addition, we can also use the VAE encoder features as high-resolution features. As shown in Tab. 26, the experimental results show that, using VAE decoder features achieves better performance in downstream segmentation tasks.

Table 26: Ablation on the features of VAE for Refined Mask Prediction.

| $F^{res}$ | AUROC | AP | $F_1$-max | IoU |
|---|---|---|---|---|
| VAE encoder | 96.14 | 66.26 | 63.48 | 54.99 |
| **VAE decoder** | **97.21** | **69.21** | **66.37** | **55.28** |

**Ablation on the normal image supervision for Refined Mask Prediction**

In the Refined Mask Prediction branch, we predict masks for normal images as the supervision for the mask prediction. We conduct two sets of experiments: (a) We remove the second and the fourth term in the loss for RMP, i.e., the normal image supervision (short for NIA in Tab. 27); (b) We use the complete form in RMP branch loss, i.e., we use the normal image for supervision, as in Eq. equation 10:

$$\mathcal{L}_{\mathcal{M}} = \mathcal{F}(\hat{M}_{\text{df}}, \mathbf{M}_{\text{df}}) + \mathcal{F}(\hat{M}_{\text{ob}}, \mathbf{M}_{\text{ob}}) + \mathcal{F}(\hat{M}'_{\text{df}}, \mathbf{M}'_{\text{df}}) + \mathcal{F}(\hat{M}'_{\text{ob}}, \mathbf{M}'_{\text{ob}}) \quad (10)$$

As shown in Tab. 27, the experimental results show that, using normal images for supervision achieves better performance in downstream segmentation tasks. We also provide further qualitative results of the effect of normal image supervision (short for NIA in Fig. 16) on MVTec AD.

Table 27: Ablation on the normal image supervision for Refined Mask Prediction.

| $F^{res}$ | AUROC | AP | $F_1$-max | IoU |
|---|---|---|---|---|
| w/o NIA | 96.20 | 66.03 | 64.09 | 53.97 |
| **with NIA (Ours)** | **97.21** | **69.21** | **66.37** | **55.28** |

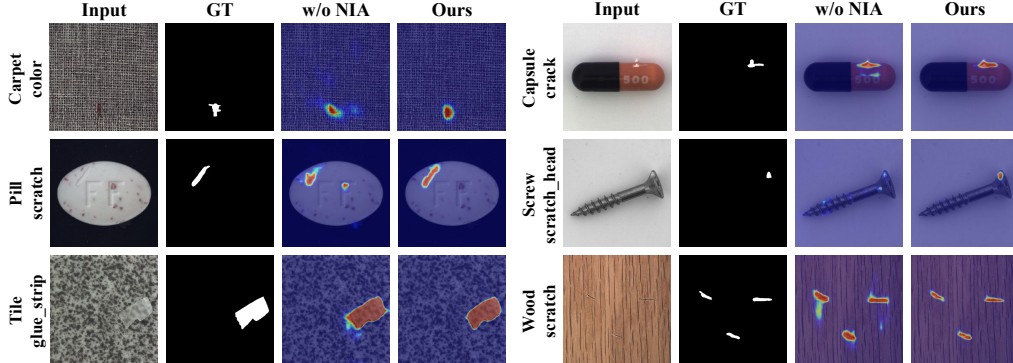

Figure 16: Qualitative results of the effect of normal image supervision on MVTec AD.

**Ablation on the Mask Refinement Module**

In the Refined Mask Prediction branch, the Mask Refinement Module (MRM) is utilized to generate refined masks. We devise different structures for MRM, as shown in Fig. 17, including Case a): those without conv blocks, Case b): with one conv blocks, and Case c): with chained conv blocks. As shown in Fig. 18, we find that using the conv blocks in Case b), which consists of two $1 \times 1$ convolutions and one $3 \times 3$ convolution, helps the model learn the features of the defect area more accurately, rather than focusing on the background area for using one convolution alone in Case a). Based on this observation, we further designed a chained conv blocks structure in Case c), and the acquired features better reflect the defect area. This one-level-by-one level of residual learning helps the model achieve better residual correction results for the defect area features. As shown in Tab. 28 in the Appendix, Case c) improves the performance by + 0.28% on AUROC, + 2.29% on AP and + 2.29% on $F_1$-max, + 0.32% on IoU compared with Case b). We substantiate the superiority of the MRM structures that we design, through the results of downstream segmentation experiments.

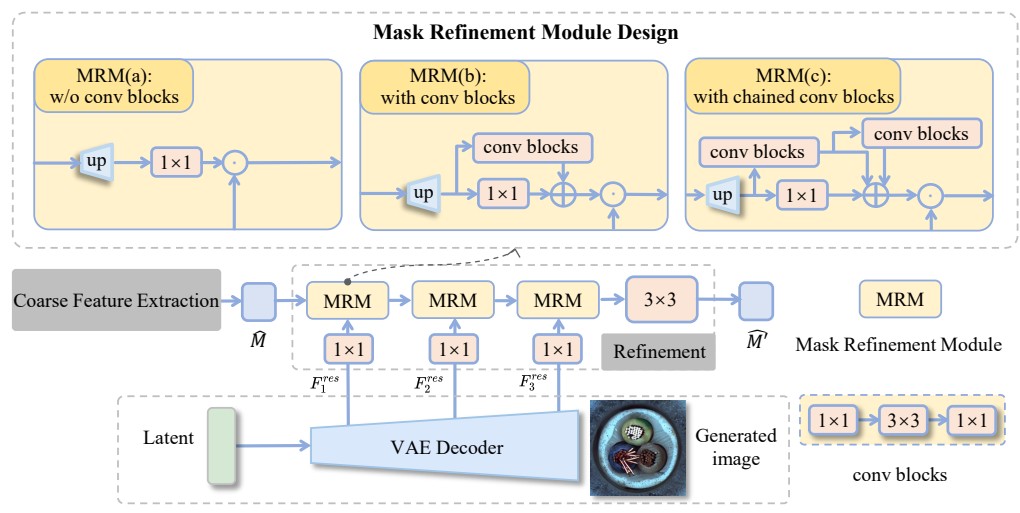

Figure 17: Different structure designs for the mask refinement module in the mask prediction branch.

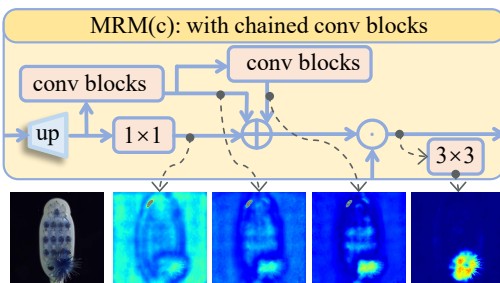

Figure 18: Visualization of the MRM module intermediate results. The top is for MRM structure diagram, and the bottom is sequentially for input image, feature maps of MRM intermediate process and the predicted mask.

Table 28: Ablation on the Mask Refinement Module.

| Model | AUROC | AP | $F_1$-max | IoU |
|---|---|---|---|---|
| with MRM (a) | 96.75 | 68.18 | 64.96 | **55.51** |
| with MRM (b) | 96.93 | 66.92 | 64.08 | 54.96 |
| **with MRM (c)** | **97.21** | **69.21** | **66.37** | 55.28 |

**Ablation on the threshold for mask binarization**

In the Refined Mask Prediction branch, we take the threshold $\tau$ for the second channel of refined anomaly masks $\hat{M}'_{\text{df}}$ to segment the final anomaly mask. We train segmentation models using anomaly masks with $\tau$ settings ranging from 0.1 to 0.5. As shown in Tab. 29, results indicate that setting $\tau = 0.2$ yields the best model performance.

Table 29: Ablation on the threshold for mask binarization.

| threshold | AUROC | AP | $F_1$-max | IoU |
|---|---|---|---|---|
| $\tau = 0.1$ | **97.56** | 65.33 | 63.38 | 52.40 |
| $\tau = 0.2$ **(Ours)** | 97.21 | **69.21** | **66.37** | **55.28** |
| $\tau = 0.3$ | 97.20 | 66.92 | 64.35 | 54.68 |
| $\tau = 0.4$ | 95.31 | 63.55 | 61.97 | 53.03 |
| $\tau = 0.5$ | 94.11 | 60.85 | 59.92 | 50.87 |

## A.9  MORE QUALITATIVE COMPARISON RESULTS OF SEGMENTATION MODELS TRAINED ON IMAGE-MASK PAIRS GENERATED BY DIFFERENT ANOMALY GENERATION METHODS

We provide further qualitative results with different anomaly generation methods on the MVTec AD dataset. We report the generation results of SeaS for varying types of anomalies in each category. Results are from Fig. 19 to Fig. 22.

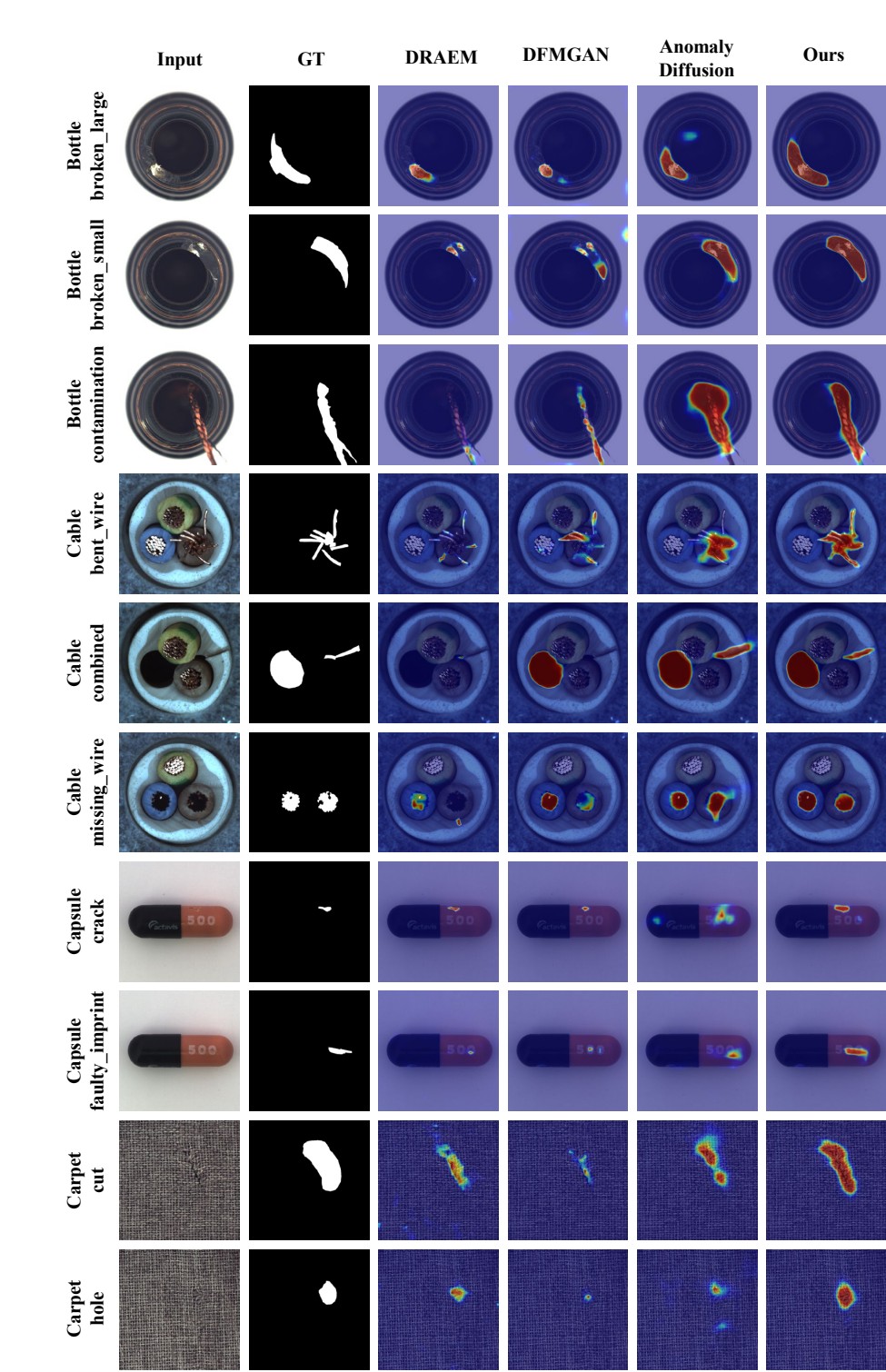

Figure 19: Comparison results with the anomaly segmentation models on MVTec AD. In the figure, from top to bottom are the results for *bottle*, *cable*, *capsule* and *carpet* categories.

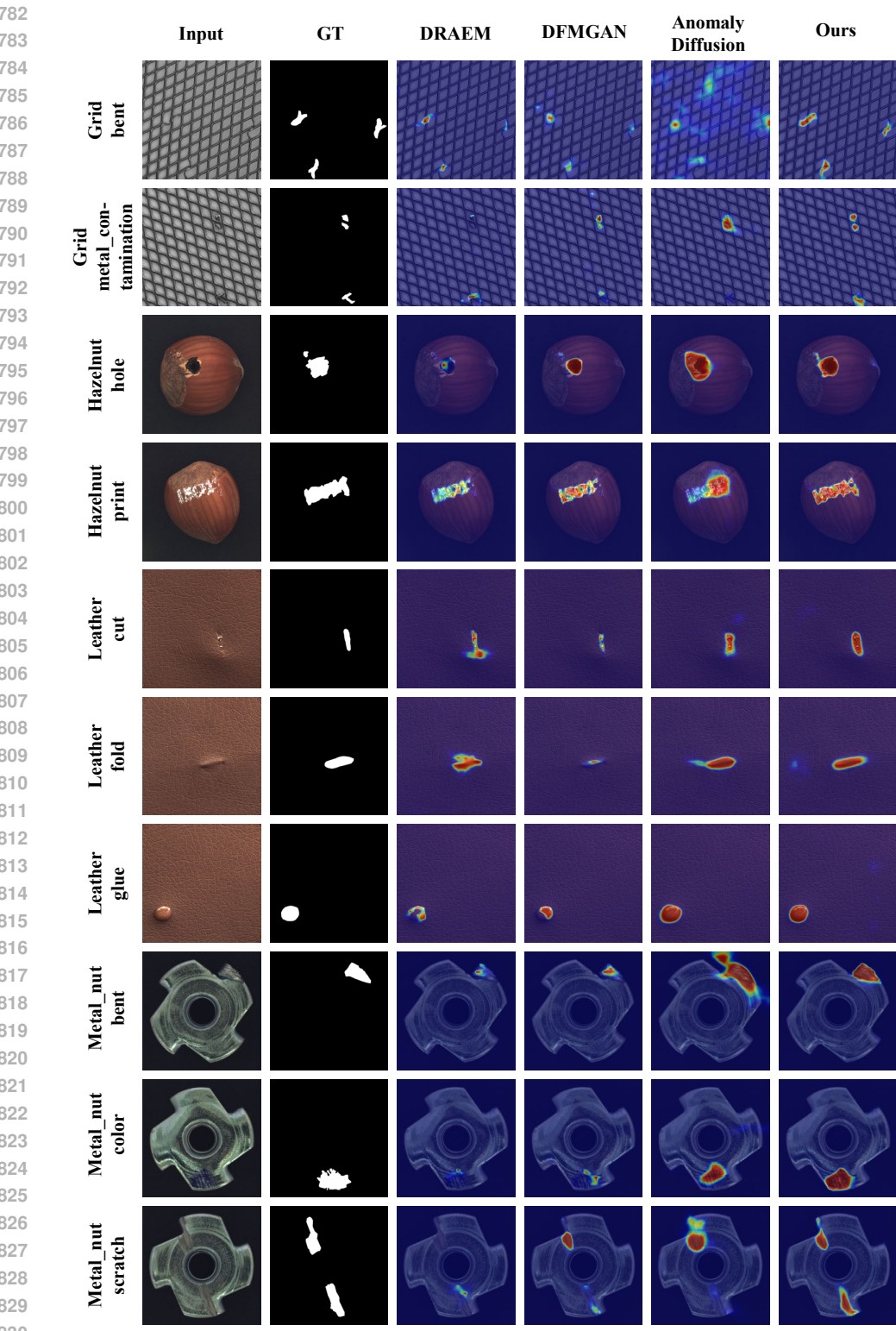

Figure 20: Comparison of the anomaly segmentation results on MVTec AD. In the figure, from top to bottom are the results for *grid*, *hazelnut*, *leather* and *metal_nut* categories.

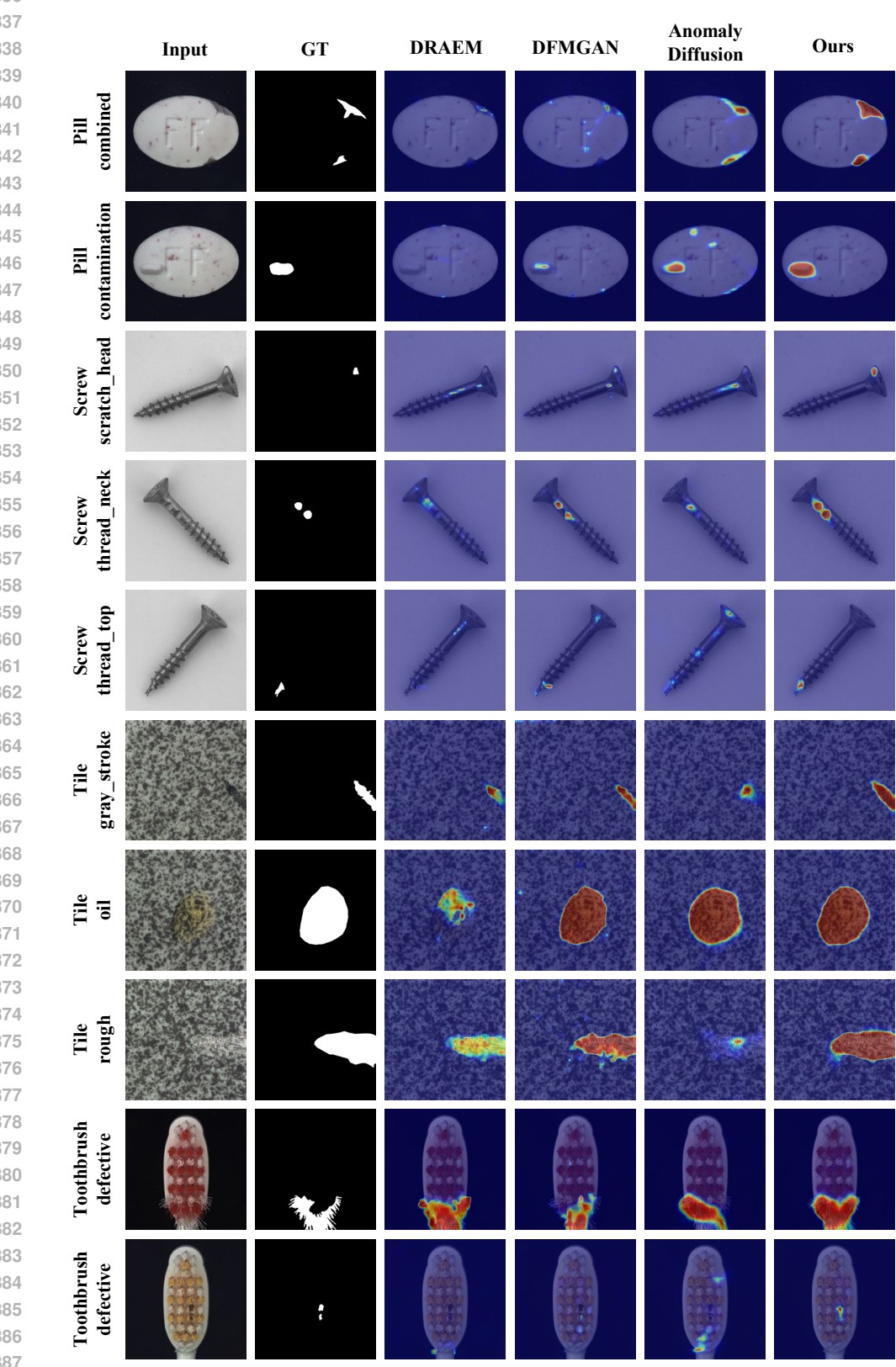

Figure 21: Comparison of the anomaly segmentation results on MVTec AD. In the figure, from top to bottom are the results for *pill*, *screw*, *tile* and *toothbrush* categories.

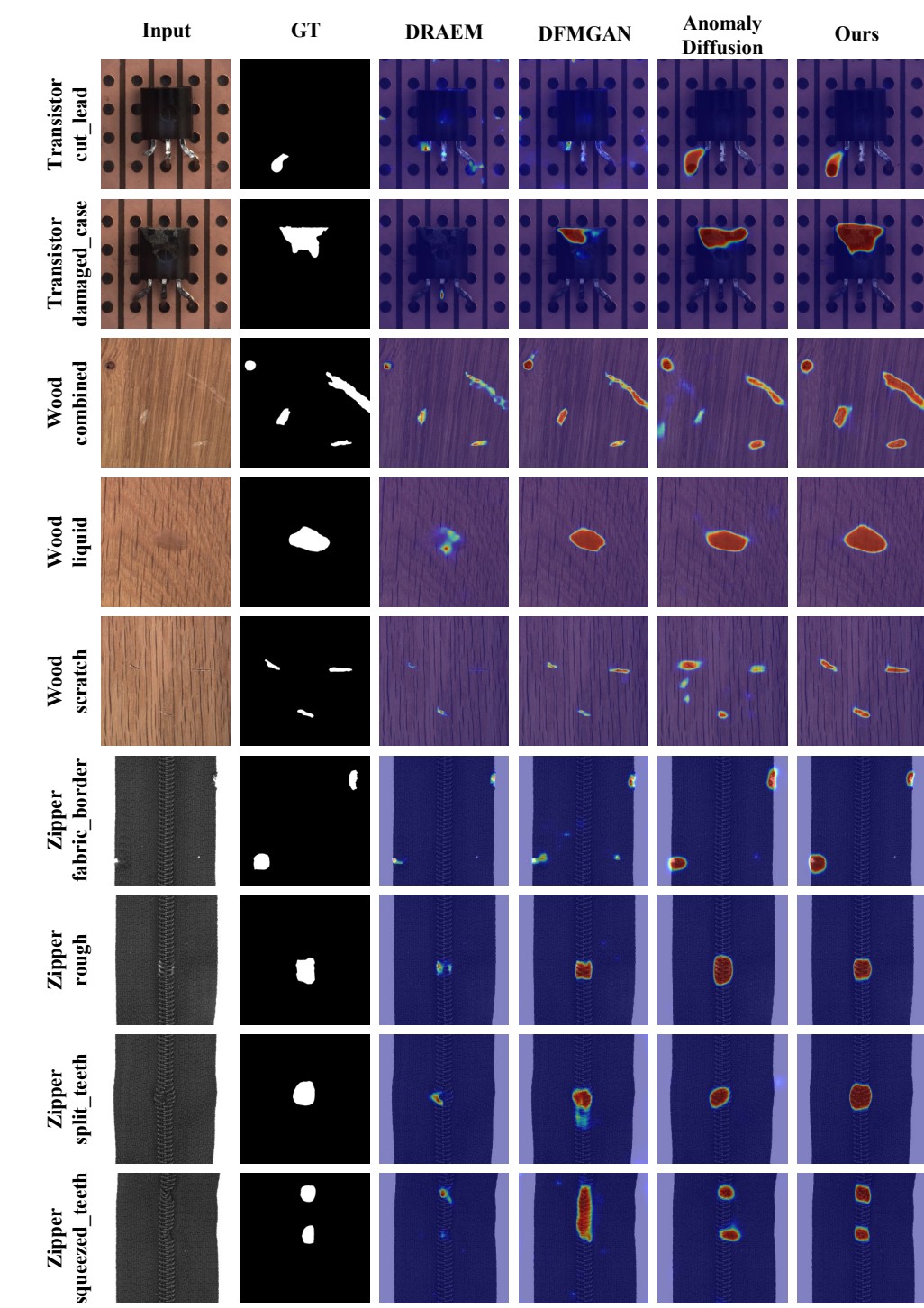

Figure 22: Comparison of the anomaly segmentation results on MVTec AD. In the figure, from top to bottom are the results for *transistor*, *wood* and *zipper* categories.

### A.10 MORE QUALITATIVE COMPARISON RESULTS OF DIFFERENT SEGMENTATION MODELS TRAINED ON IMAGE-MASK PAIRS GENERATED BY SEAS

In this section, we provide further qualitative results with different segmentation models on the MVTec AD dataset. We choose three models with different parameter quantity scopes (BiSeNet V2 (Yu et al., 2021): 3.341M, UPerNet (Xiao et al., 2018): 64.042M, LFD (Zhou et al., 2024a): 0.936M). We report the segmentation results of SeaS for varying types of anomalies in each category. Results are from Fig. 23 to Fig. 26.

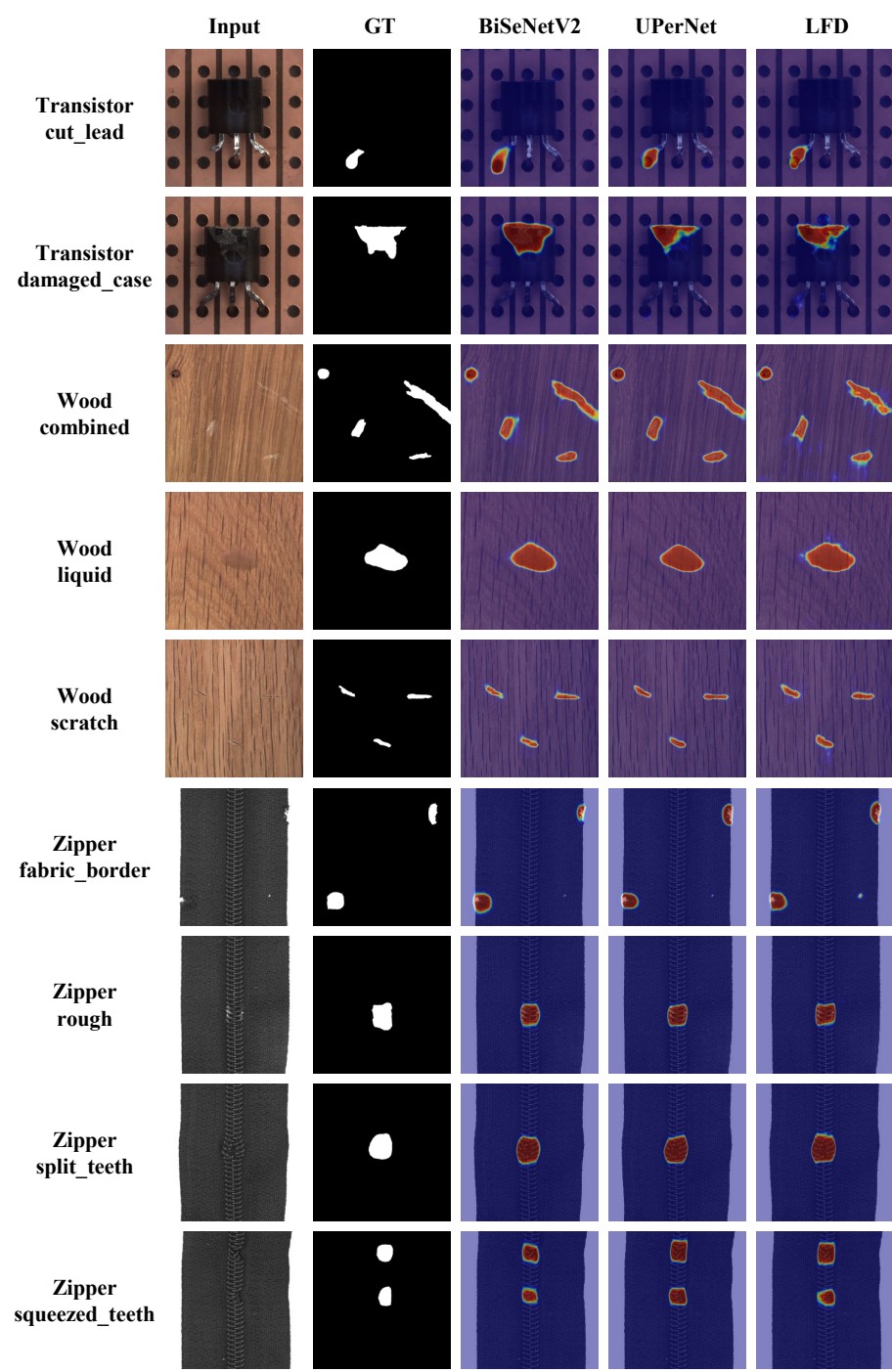

Figure 23: Qualitative comparison results with the segmentation models on MVTec AD. In the figure, from top to bottom are the results for *transistor*, *wood*, and *zipper* categories.

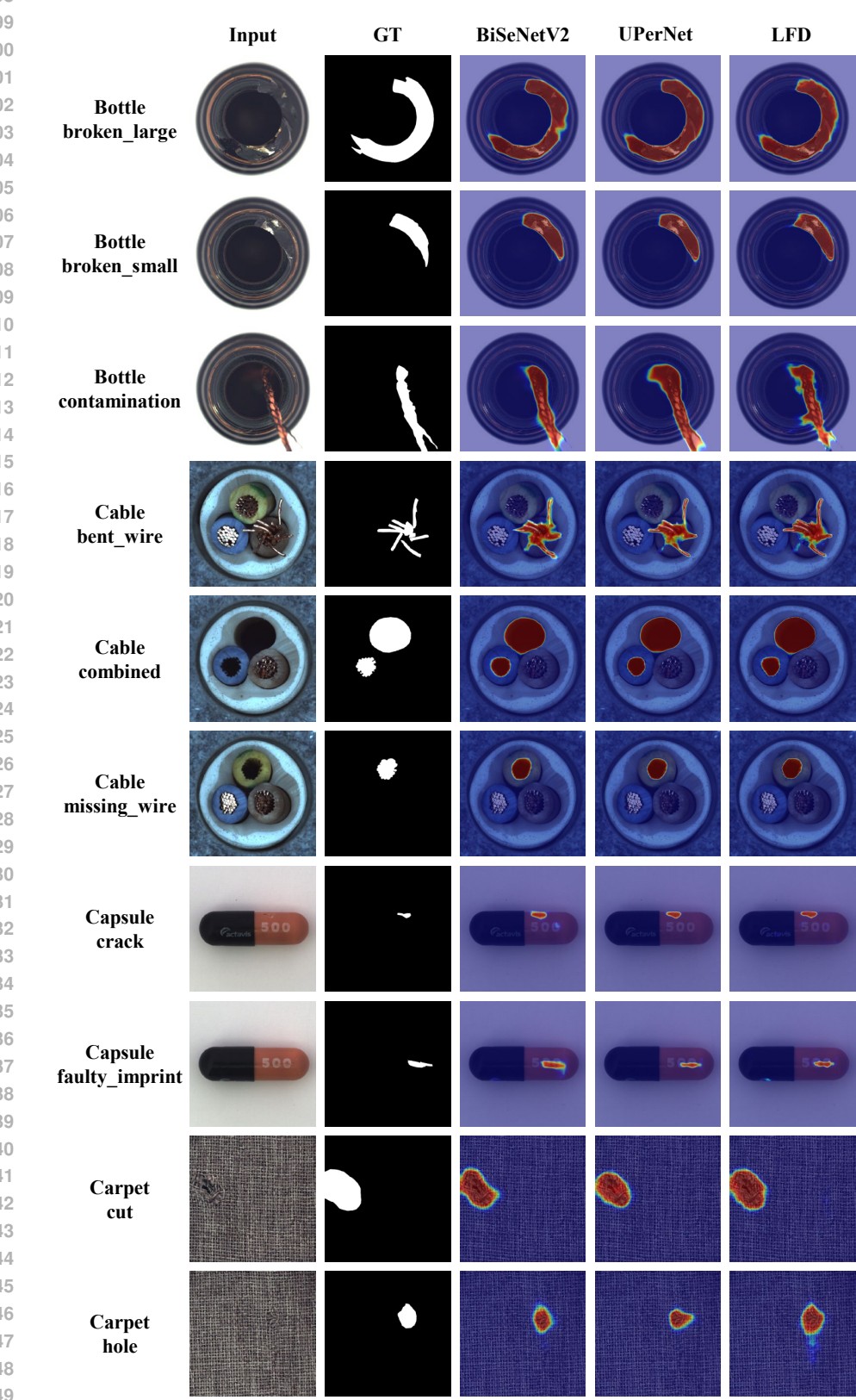

Figure 24: Qualitative comparison results with the segmentation models on MVTec AD. In the figure, from top to bottom are the results for *bottle*, *cable*, *capsule* and *carpet* categories.

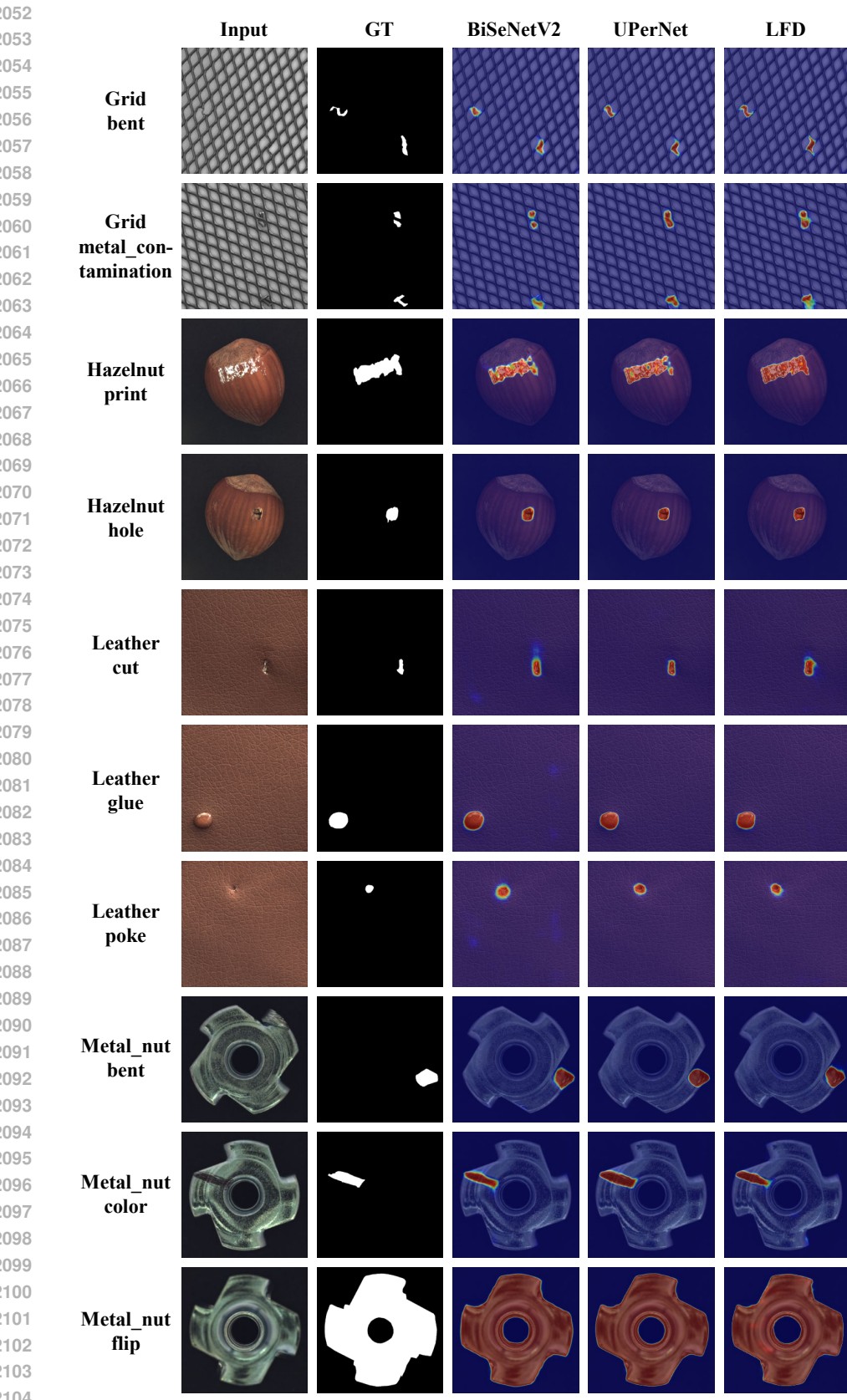

Figure 25: Qualitative comparison results with the segmentation models on MVTec AD. In the figure, from top to bottom are the results for *grid*, *hazelnut*, *leather* and *metal_nut* categories.

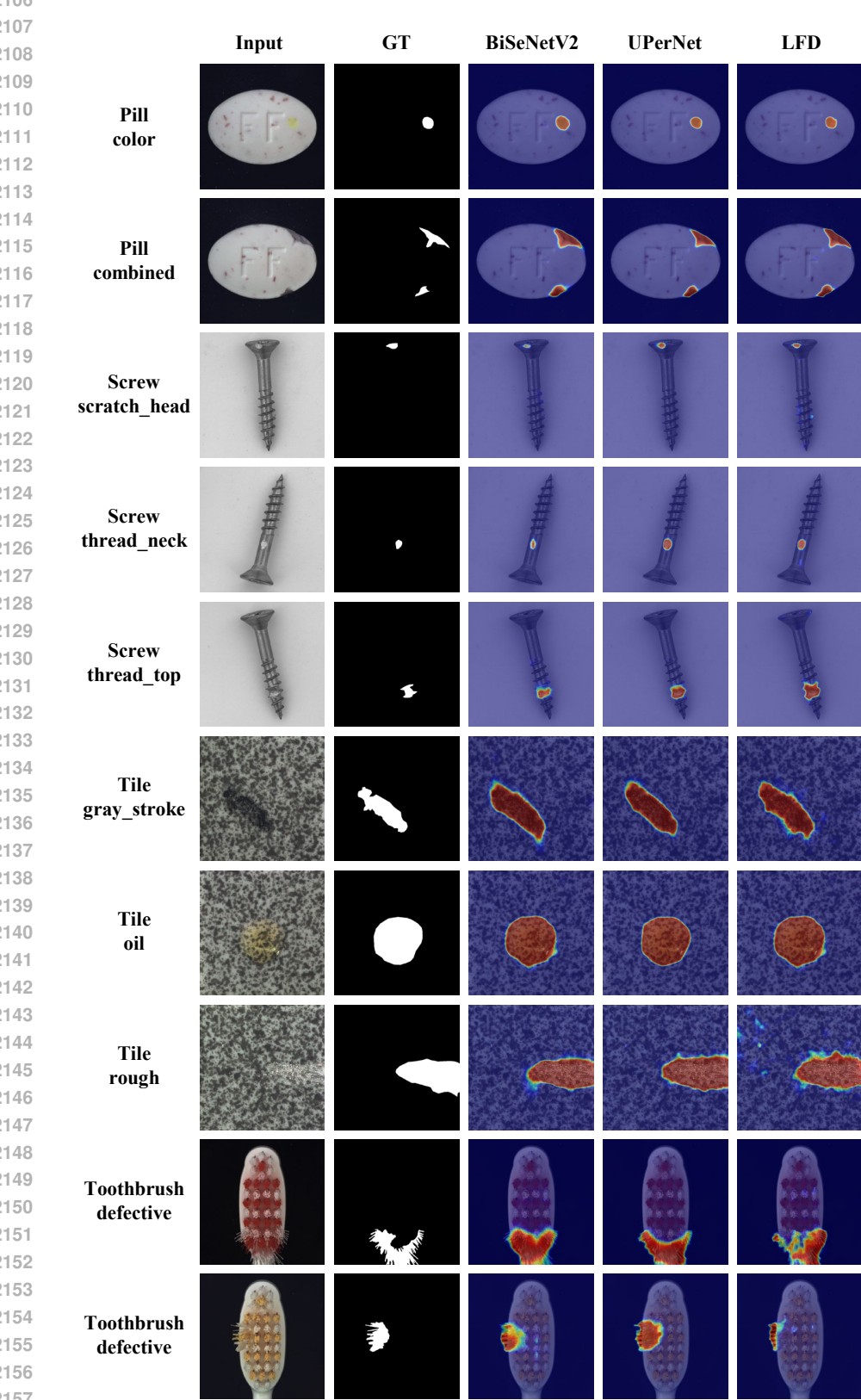

Figure 26: Qualitative comparison results with the segmentation models on MVTec AD. In the figure, from top to bottom are the results for *pill*, *screw*, *tile* and *toothbrush* categories.

### A.11 ADDITIONAL VISA DATASET RESULTS

We perform experimental evaluations on the images of the VisA Dataset (Zou et al., 2022), which includes 12 product categories, each with up to 9 different anomalies.

As shown in Tab. 30 and Fig. 27, SeaS generates anomaly images with higher fidelity and diversity. Tab. 31 shows the comparisons on downstream supervised segmentation trained by the generated images. It consistently demonstrates that our method outperforms others across all the segmentation models, with an 11.71% average improvement on IoU. We report the image-level metrics in Tab. 32 and our method achieve a 5.92% gain on image-AUROC. We show the segmentation anomaly maps in Fig. 28, by using our generated image-mask pairs to train BiSeNet V2, there are fewer false positives in *chewinggum* and fewer false negatives in *pcb1* and *pipe_fryum*.

Table 30: Comparison on IS and IC-LPIPS on VisA. Bold indicates the best performance.

| Method | DFMGAN (Duan et al., 2023) | | AnomalyDiffusion (Hu et al., 2024) | | Ours | |
|---|---|---|---|---|---|---|
| | IS ↑ | IC-L ↑ | IS ↑ | IC-L ↑ | IS ↑ | IC-L ↑ |
| Average | 1.25 | 0.25 | 1.26 | 0.25 | **1.27** | **0.26** |

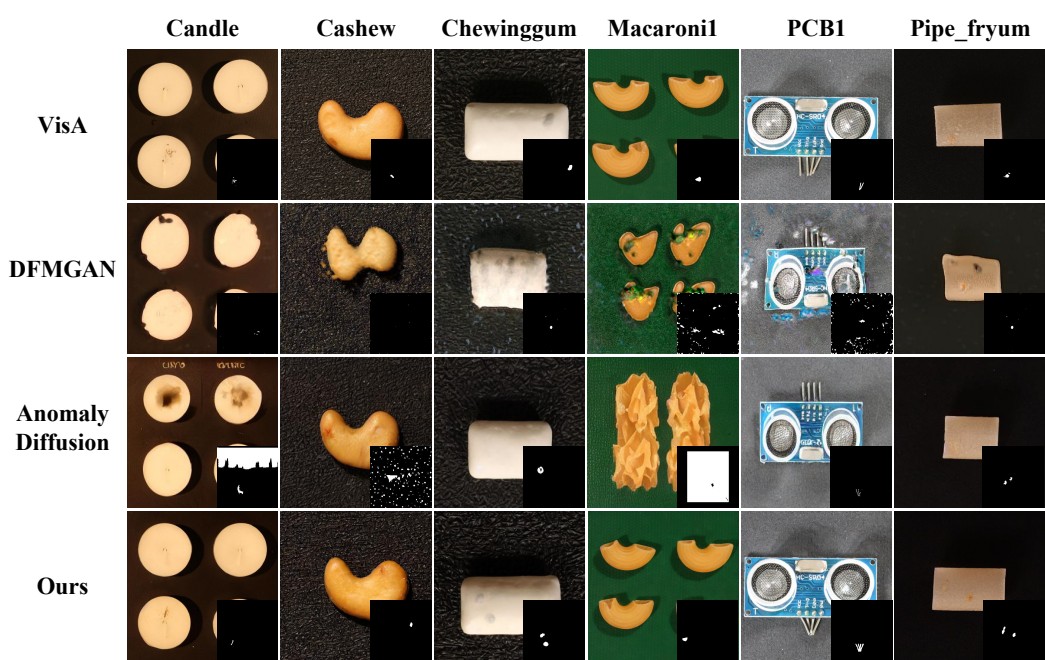

Figure 27: Visualization of the generation results on VisA. The sub-image in the lower right corner is the generated mask.

Table 31: Comparison on anomaly segmentation on VisA.

| Model | DFMGAN (Duan et al., 2023) | | | | | AnomalyDiffusion (Hu et al., 2024) | | | | | Ours | | | | |
|---|---|---|---|---|---|---|---|---|---|---|---|---|---|---|---|
| | AUROC | AP | $F_1$-max | PRO | IoU | AUROC | AP | $F_1$-max | PRO | IoU | AUROC | AP | $F_1$-max | PRO | IoU |
| BiSeNet V2 (Yu et al., 2021) | 75.91 | 9.17 | 15.00 | 21.49 | 9.66 | 89.29 | 34.16 | 37.93 | 28.09 | 15.93 | 96.03 | 42.80 | 45.41 | 61.29 | 25.93 |
| UPerNet (Xiao et al., 2018) | 75.09 | 12.42 | 18.52 | 27.38 | 15.47 | 95.00 | 39.92 | 45.37 | 44.90 | 20.53 | 97.01 | 55.46 | 55.99 | 58.90 | 35.91 |
| LFD (Zhou et al., 2024a) | 81.21 | 15.14 | 18.70 | 14.98 | 6.44 | 88.00 | 30.86 | 36.56 | 38.56 | 16.61 | 92.91 | 43.87 | 46.46 | 29.55 | 26.37 |
| Average | 77.40 | 12.24 | 17.41 | 21.28 | 10.52 | 90.76 | 34.98 | 39.95 | 37.18 | 17.69 | **95.32** | **47.38** | **49.29** | **49.91** | **29.40** |

Table 32: Comparison on image-level anomaly detection on VisA.

| Model | DFMGAN (Duan et al., 2023) | | | AnomalyDiffusion (Hu et al., 2024) | | | Ours | | |
|---|---|---|---|---|---|---|---|---|---|
| | AUROC | AP | $F_1$-max | AUROC | AP | $F_1$-max | AUROC | AP | $F_1$-max |
| BiSeNet V2 (Yu et al., 2021) | 63.07 | 62.63 | 66.48 | 76.11 | 77.74 | 73.13 | 85.61 | 86.64 | 80.49 |
| UPerNet (Xiao et al., 2018) | 71.69 | 71.64 | 70.70 | 83.18 | 84.08 | 78.88 | 90.34 | 90.73 | 84.33 |
| LFD (Zhou et al., 2024a) | 65.38 | 62.25 | 66.59 | 81.97 | 82.36 | 77.35 | 83.07 | 82.88 | 77.24 |
| Average | 66.71 | 65.51 | 67.92 | 80.42 | 81.39 | 76.45 | **86.34** | **86.75** | **80.69** |

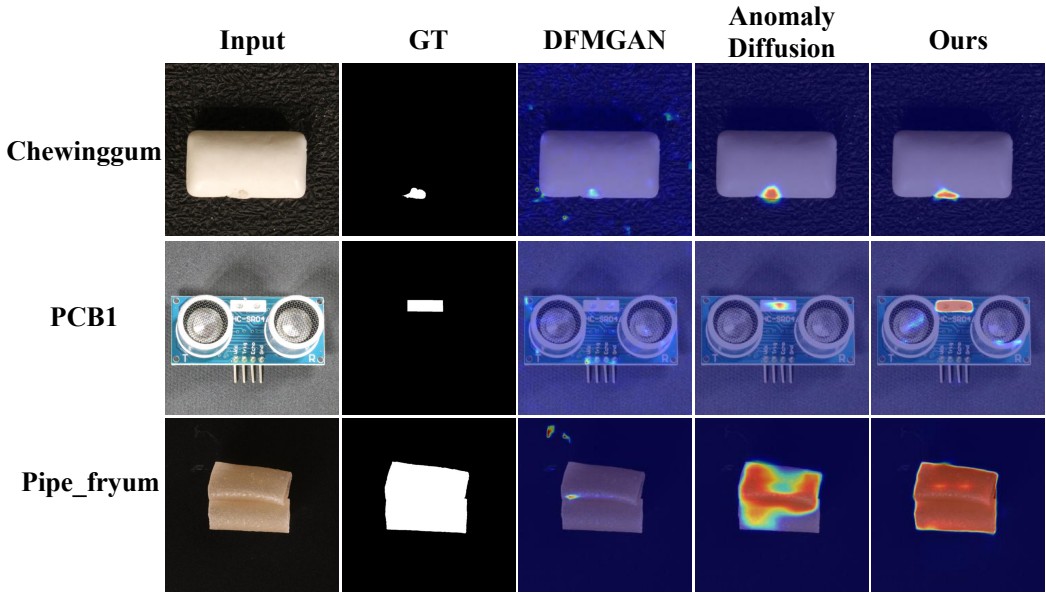

Figure 28: Qualitative anomaly segmentation results with BiSeNet V2 on VisA.

## A.12 EXPLANATION OF DISCRIMINATIVE FEATURES IN U-NET DECODER

The U-Net can learn the highly discriminative features of the defect area accurately. As shown in Fig. 29, we use the output features of the "up-2" and "up-3" layers of the decoder in U-Net, and apply convolution blocks and concatenation operations, then we can obtain the unified coarse feature $\hat{F} \in \mathbb{R}^{64 \times 64 \times 192}$, which can be used to predict masks corresponding to anomaly images.

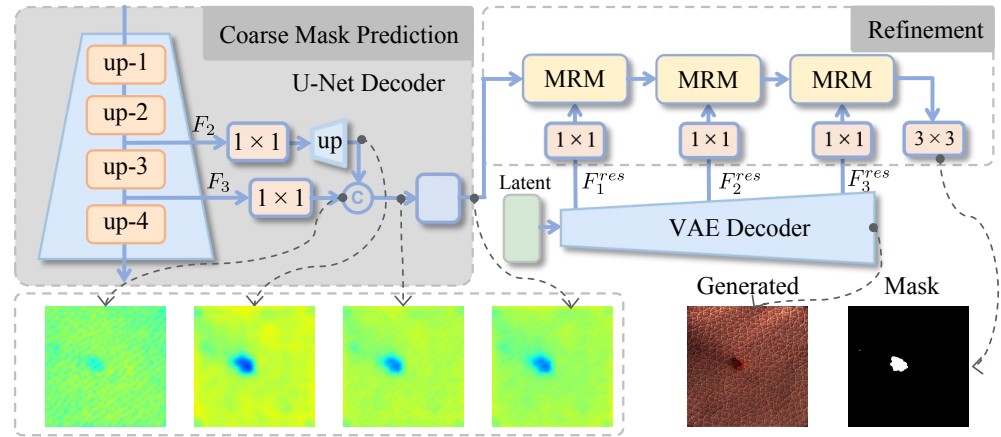

Figure 29: Visualization of the U-Net decoder features in mask prediction process.

## A.13 COMPARISON WITH THE TEXTUAL INVERSION

We conduct the experiment of only using the Textual Inversion (TI) (Gal et al., 2022) method to learn the product, and the generated images are shown in Fig. 30. The TI method struggles to generate images similar to the real product due to the limited number of learnable parameters. In contrast, for the AIG method, the products satisfy global consistency with minor variations in local details, while the anomalies hold randomness, so the generated products should be globally consistent with the real products. Therefore, unlike the AG method AnomalyDiffusion (Hu et al., 2024), where the TI method alone is sufficient to meet the anomaly generation needs, we fine-tunes the U-Net to ensure the global consistency of the generated products.

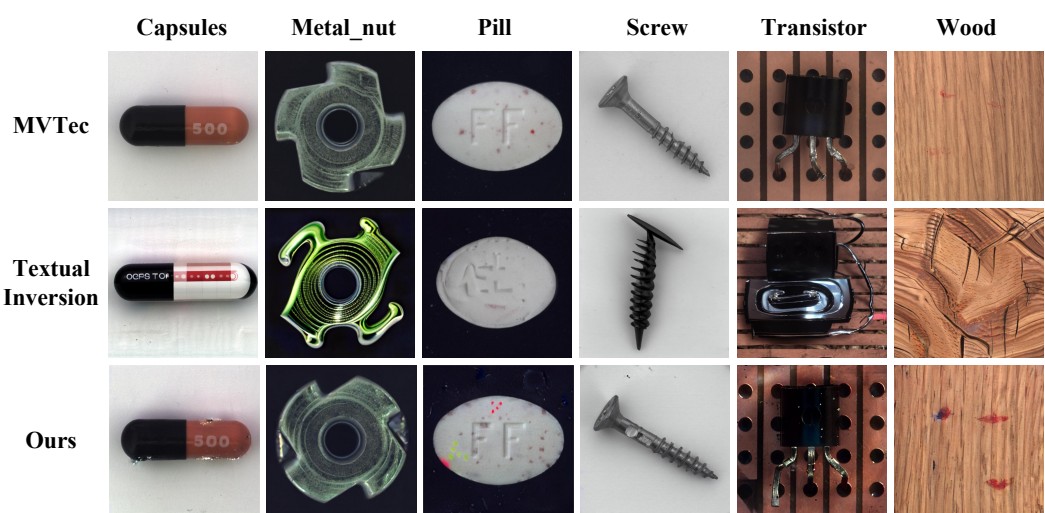

Figure 30: Qualitative comparison on the generation results with Textual Inversion.

### A.14 MORE EXPERIMENTS ON LIGHTING CONDITIONS

We choose one defect class from peach, a product in the MVTec3D dataset, that has significant variations in lighting conditions and backgrounds, to conduct experiments. Images with strong lighting conditions depict the top side of the peach, whereas those with weak lighting conditions show the bottom side. Consequently, the background in the images, whether the top or bottom of the peach, also differs. We selected three training sets with different lighting conditions for experiments: 1) only images from the top side with strong lighting condition, 2) only images from the bottom side with weak lighting condition, 3) half of the images from the top side with strong lighting condition, and a half from the bottom side with weak lighting condition. The generated images of different settings are shown in Fig. 31. It can be seen that SeaS is robust against lighting conditions and background variations.

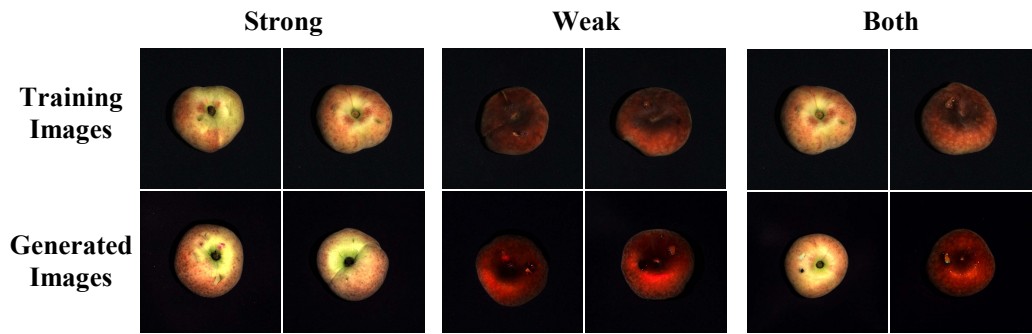

Figure 31: Visualization of the generation results on MVTec3D AD on different lighting conditions and backgrounds. In the figure, the first row is for the training images and the second row is for the generated images.

## A.15 MORE EXPERIMENTS ON REPLACING GENERATION STRATEGIES.

We replace the abnormal generation strategy in DRAEM (Zavrtanik et al., 2021) and BGAD (Yao et al., 2023b) with the proposed generation strategy, the results are given in Tab. 33. The segmentation result demonstrates that our method outperforms the exsiting anomaly detection methods.

Table 33: Comparison on replacing generation strategies with anomaly detection methods on MVTec AD.

| Model | Image-level | | | Pixel-level | | | |
|---|---|---|---|---|---|---|---|
| | AUROC | AP | $F_1$-max | AUROC | AP | $F_1$-max | IoU |
| DRAEM (Zavrtanik et al., 2021) | 98.00 | 98.45 | 96.34 | 97.90 | 67.89 | 66.04 | 60.30 |
| SeaS + DRAEM | **99.25** | **99.66** | 98.35 | 97.98 | **77.35** | 73.27 | **63.99** |
| BGAD (Yao et al., 2023b) | 98.31 | 98.05 | 98.27 | **99.26** | 73.85 | 77.89 | 60.60 |
| SeaS + BGAD | 98.44 | 98.18 | **99.08** | **99.26** | 73.85 | **77.93** | 60.81 |

## A.16 MORE VISUALIZATION RESULTS ON RECOMBINING THE DECOUPLED ATTRIBUTES FOR UNSEEN ANOMALIES.

We provide more examples in Fig. 32, where new anomalies are generated that significantly differ from the training samples in terms of color and shape. For example, we showcase *bottle_contamination*, *hazelnut_print*, and *tile_gray_stroke* with a novel shape, *wood_color* and *metal_nut_scratch* with a novel color, and *pill_crack* with a new shape, featuring multiple cracks where the training samples only exhibit a single crack. These examples demonstrate the the model's ability to create unseen anomalies based on recombining the decoupled attributes.

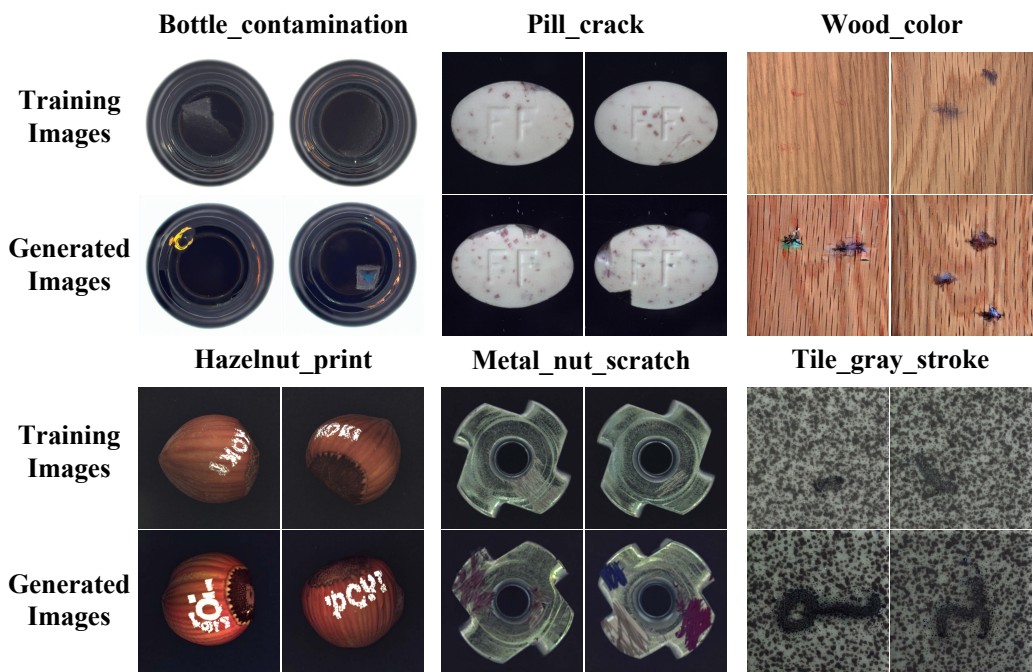

Figure 32: Visualization of the generation results for unseen anomalies on MVTec AD.

