# OpenReview forum: "SeaS: Few-shot Industrial Anomaly Image Generation with Separation and Sharing Fine-tuning"
_ICLR.cc/2025/Conference — Submitted to ICLR 2025_

### Official Review · Reviewer_EXr6 · 2024-11-01

**Soundness:** 2
**Presentation:** 3
**Contribution:** 1
**Rating:** 6
**Confidence:** 4

**Summary:**

This work proposes a few-shot industrial anomaly generation method named SeaS.  The authors use a technique similar to textual inversion to update the learnable prompt, and then use the prompt and latent diffusion to generate defects. Moreover, this work proposes an alignment (DA) loss to assign different attributes of the anomalies to different anomaly tokens, achieving a high diversity for anomalies generation.

**Strengths:**

1. This paper is well organized, and easy to follow.

2. Extensive experiments on MVTec AD and MVTec 3D demonstrate the effectiveness of proposed seaS.

3. The decoupled anomaly alignment loss makes sense, which improves the effect of texual inversion.

**Weaknesses:**

1. The design of "Unbalanced Abnormal Text Prompt" is similar with previous works [1, 2], which compromises the novelty.

2. While the paper demonstrates strong performance on MVTec datasets, it is unclear how well the SeaS method generalizes to other datasets or different types of unseen anomalies during training.

3. The paper should provide more details on the computational complexity, as the two training processes.

4. The scalability of the SeaS approach to handle a large number of anomaly types or a higher resolution of images is not discussed. It is an important consideration for industrial applications with diverse and high-resolution imagery.

5. As for the part of defect generation, the key lies in whether the generated defect samples can improve the model detection ability, rather than paying attentions on whether the samples are similar to real defects. However, the metrics used  in Table 1 are all about measures on the generated image.  Author should focus more on generating samples to enhance model detection performance.

6. The author should conduct experiments on more datasets. At present, some high-quality datasets are existing, such as VisA [4] and Real-IAD [3].

[1] Gal, Rinon, et al. "An image is worth one word: Personalizing text-to-image generation using textual inversion." arXiv preprint arXiv:2208.01618 (2022).

[2] Li, Xiaofan, et al. "Promptad: Learning prompts with only normal samples for few-shot anomaly detection." In CVPR,2024.

[3] Chengjie Wang, Wenbing Zhu, Bin-Bin Gao, Zhenye Gan, Jiangning Zhang, Zhihao Gu, Shuguang Qian, MingangChen, and Lizhuang Ma. Real-iad: A real-world multi-view dataset for benchmarking versatile industrial anomaly detection. In CVPR, 2024. 5

[4] Yang Zou, Jongheon Jeong, Latha Pemula, Dongqing Zhang, and Onkar Dabeer. Spot-the-difference self-supervised pretraining for anomaly detection and segmentation. In ECCV, 2022. 2, 5, 6

**Questions:**

Please report the time cost of the proposed method. To my knowledge, diffusion-based work takes long time. But in real industry applications, we need lower time cost.

---

> ### Author Response · Authors · 2024-11-25
>
> ***Q1: The design of "Unbalanced Abnormal Text Prompt" is similar with previous works [1, 2], which compromises the novelty.***
>
> >**A1**: Thanks for your comment! Here, we clarify the fundamental difference between the Unbalanced Abnormal (UA) Text Prompts in SeaS and existing prompts. This difference is one of the key reasons behind the improvement of SeaS. Below, the existing prompts are briefly outlined:
>
> >**Textual Inversion** [1] leverages the following prompt:
> "A photo of $S_*$"
> where $S_*$ is a placeholder string, representing the new concept we wish to learn. This approach  only learns one concept at a time, which is not sufficient for AIG tasks that require differentiation between anomalies and products. We added the experiment of only using the Textual Inversion (TI) method to learn the product, which can be shown in Fig. 30. Due to the limited number of learnable parameters, the TI method struggles to generate images similar to the real product.
>
> >**PromptAD** [2] leverages the following prompt:
> $$\begin{eqnarray}s^n & = & [P_1][P_2] . . . [P_{E_N} ][obj.] \end{eqnarray}$$
> $$\begin{eqnarray}s^m & = & [P_1][P_2] . . . [P_{E_N}][obj.][with][color][stain] \end{eqnarray}$$
> $$\begin{eqnarray}s^l & = & [P_1][P_2] . . . [P_{E_N} ][obj.][A_1] . . . [A_{E_A}]\end{eqnarray}$$
> where $E_N$ denotes the length of the learnable normal prefix in the learnable normal prompt and manual anomaly prompt, "[with][color][stain]" denotes anomaly suffixes from the anomaly labels of the datasets, and $E_A$ denotes the length of learnable anomaly suffix in learnable anomaly prompt. PromptAD incorporates predefined semantic words such as “bottle” and “color stain”. While this method incorporates anomaly labels, it relies on fixed, generic semantic terms that may not align well with a few training images that contain specific anomaly types.
>
> >In contrast, our **UA Text Prompts** approach allows the product and anomaly to be independently learned through embeddings, making it more flexible and capable of handling a wider variety of anomalies effectively. The unbalanced design is not just a structural difference, but is crucial for separately learning common characteristics of products and diverse characteristics of anomalies.
>
> >We hope this explanation clarifies the novelty and advantages of our approach. Thank you again for your feedback!
>
>
> ***Q2: While the paper demonstrates strong performance on MVTec datasets, it is unclear how well the SeaS method generalizes to other datasets or different types of unseen anomalies during training.***
>
> >**A2**: Thank you for the comment! As we mentioned in Lines 22-24 in the paper. "***Recombining the decoupled attributes may produce anomalies that have never been seen in the training dataset"***. Such a conclusion can  be observed in Fig. 3(c) in the paper, and unseen samples in the same category can be handled efficiently.
>
> >For the generalization to other datasets, as we mentioned in lines 407-409 in the paper, SeaS demonstrates strong performance on the RGB images of the MVTec 3D AD dataset. The MVTec 3D AD dataset contains some common challenges, such as lighting condition variations, product pose variations, and so on, which are very important to validate the generalizability and robustness of our method. Additionally, we have added experimental evaluations on the VisA dataset in the **General Response**.
>
> ***Q3: The paper should provide more details on the computational complexity, as the two training processes.Please report the time cost of the proposed method.***
>
> >**A3**: Thanks for your comment. As we mentioned in Appendix A.3 in the paper, SeaS trains a generation model on a single NVIDIA Tesla A100 40G GPU, which only requires about 20G GPU memory.
>
> >Comparisons of computational resource costs are given in Tab. 6 and Tab. 7 of the appendix. For the entire MVTecAD datasets with 73 anomaly types, our training takes 73 hours totally, which is only **30%** of the time of the AnomalyDiffusion (249 hours) and only **18%** of the time of DFMGAN (414 hours). In terms of inference time, SeaS costs 720 ms per image, which is only **19%** of the 3830 ms per image required by the diffusion-based method AnomalyDiffusion.

---

> > ### Author Response · Authors · 2024-11-25
> >
> > ***Q4: The scalability of the SeaS approach to handle a large number of anomaly types or a higher resolution of images is not discussed. It is an important consideration for industrial applications with diverse and high-resolution imagery.***
> >
> > >**A4**: Thanks for your comment! As far as we know, in various datasets, MVTec AD, VisA, and MVTec 3D AD, a single product has at most 8 to 9 anomaly categories. SeaS can handle this well, as shown in Fig. 9-10 and Fig. 12-14 for the generated image of multiple anomaly categories for each product. For products with over 9 anomaly categories, theoretically, SeaS can be compatible with a larger number of anomaly types by increasing the number of sets of the anomaly tokens in our unbalanced abnormal text prompt. Further verification is needed when a larger scale of anomaly  type product samples are available.
> >
> > >Regarding the high-resolution images, in order to match the input image size of the pretrained StableDiffusion model, we will resize the images of any resolution to 512 × 512 size before training. To generate higher resolution images, a potential way is to replace the pre-trained  model.
> >
> > ***Q5: As for the part of defect generation, the key lies in whether the generated defect samples can improve the model detection ability, rather than paying attentions on whether the samples are similar to real defects. However, the metrics used in Table 1 are all about measures on the generated image. Author should focus more on generating samples to enhance model detection performance.***
> >
> > >**A5**: Thanks for your comment! We fully agree with you, and have done many experiments to demonstrate the contribution of our generated samples to the detection performance of models.
> >
> > >First, SeaS performs better on enhancing the downstream segmentation performance compared with AnomalyDiffusion and DFMGAN. As we mentioned in Lines 33-36, ***"For the downstream task, by using our generated anomaly image-mask pairs, three common segmentation methods achieve an average 11.17% improvement on IoU on MVTec AD dataset, and a 15.49% enhancement in IoU on the MVTec 3D AD dataset."*** We use the generated image-mask pairs to train the unified segmentation models. The results, as shown in Tab. 2, Tab. 3, Tab. 9 and Tab. 10, demonstrate that the segmentation model trained with our generated data outperforms existing anomaly image generation methods.
> >
> > >Second, with the training samples generated by SeaS, the conventional lightweight segmentation models surpass existing AD methods, as shown in Tab. 11 and Tab. 12.
> >
> > >Third, training on the samples generated by SeaS, existing AD methods can achieve a higher performance on segmentation, as shown in **Tab. 33** in the appendix.
> >
> >
> > ***Q6: The author should conduct experiments on more datasets. At present, some high-quality datasets are existing, such as VisA [4] and Real-IAD [3].***
> >
> > >**A6**: Thanks for your suggestion! The new results on the VisA dataset are given in the **General Response**. Our method outperforms others across all the segmentation models, with an **11.71%** average improvement on IoU.

---

> > > ### Comment · Reviewer_EXr6 · 2024-11-28
> > >
> > > Thanks for your detailed reply. At present, augmentation-based work can be roughly divided into handcraft-based method and generative-model-based methods. Recently, GLASS [1] has manually generated defects using some simple image processing techniques, achieving a high performance of image-level AUROC 99.9 and pixel-level AUROC 99.3 on MVTec AD. Furthermore, the generation speed of handcraft-based method is much faster than diffusion-based methods (as you mentioned in your response, the cost of 73 hours for training on MVTec AD).  Existed generative-model-based works only generate synthetic samples that are similar to real defects, but do not effectively improve detection capabilities. Therefore, I am still concerned that generative focuses too much on some generated metrics, such as IS and IC-L, and ignores detection metrics. Based on the above considerations, I will continue to maintain my score of 5.
> > >
> > > [1] Chen Q, Luo H, Lv C, et al. A Unified Anomaly Synthesis Strategy with Gradient Ascent for Industrial Anomaly Detection and Localization. In ECCV2024.

---

> > > > ### Author Response · Authors · 2024-12-03
> > > >
> > > > ***Q7: At present, augmentation-based work can be roughly divided into handcraft-based method and generative-model-based methods. Recently, GLASS [1] has manually generated defects using some simple image processing techniques, achieving a high performance of image-level AUROC 99.9 and pixel-level AUROC 99.3 on MVTec AD. Furthermore, the generation speed of handcraft-based method is much faster than diffusion-based methods (as you mentioned in your response, the cost of 73 hours for training on MVTec AD).***
> > > >
> > > > > **A7**: Thank you for your insightful comment! As is well known, different metrics capture specific aspects of anomaly detection (AD) performance:
> > > > >
> > > > > - Pixel-level metrics provide more detailed evaluations than image-level metrics.
> > > > > - AUROC is sensitive to false negatives, while AP and F1-max are more affected by false positives.
> > > >
> > > > > While existing unsupervised AD methods excel in image-level AUROC (e.g., EfficientAD achieves 99.8 without auxiliary synthetic abnormal data), the main challenge lies in improving pixel-level AP and F1-max, i.e. key metrics in practical industrial applications. For instance, in mobile phone display manufacturing, precise anomaly localization is crucial for processes like demura on MURA defects. To address this, methods like DRAEM emphasize that incorporating synthetic anomalies helps define a decision boundary between normal and anomalous examples. Therefore, segmentation metrics, which assess localization accuracy, are more suitable for evaluating the quality of anomaly generation.
> > > >
> > > > > **AD method is a complex system, while SeaS only focuses on generating high-quality anomaly images**. Therefore, we also integrated SeaS with GLASS[1], a SOTA AD method. For a fair comparison, we replaced the pseudo-anomaly images in GLASS, which are generated by pasting anomalies, with anomaly images generated by SeaS, and keep the other modules unchanged. Tab. R11 shows that GLASS+SeaS outperforms the original GLASS, achieving +2.73% AP, +1.96% F1-max, and +0.31% IoU at the pixel level and slight improvements in image-level AUROC, AP, and F1-max, further proving SeaS’s effectiveness. The performance of GLASS differs from the original paper, as we mentioned in Lines 401-402, we use 2/3 of the anomaly images and all good images from the MVTec AD testing set, while the original results were based on the full test set.
> > > >
> > > > > In summary, as shown in Tab. 33 and Tab. R11, combining SeaS with existing SOTA AD menthods (like DRAEM, BGAD, and GLASS) leads to substantial performance gains in anomaly detection task, particularly in pixel-level accuracy, demonstrating its superiority and flexibility over traditional manual anomaly generation techniques.
> > > >
> > > > >**Table R11. Comparison of anomaly segmentation with manually generation method.**
> > > > |                | image-level |           |           | pixel-level |           |           |           |
> > > > | :------------: | :---------: | :-------: | :-------: | :---------: | :-------: | :-------: | :-------: |
> > > > |                |    AUROC    |    AP     |  F1-max   |    AUROC    |    AP     |  F1-max   |    IoU    |
> > > > |     GLASS      |    99.92    |   99.98   |   99.60   |    99.27    |   74.09   |   70.42   |   57.14   |
> > > > | **SeaS+GLASS** |  **99.97**  | **99.99** | **99.81** |  **99.29**  | **76.82** | **72.38** | **57.45** |
> > > >
> > > >
> > > >
> > > > ***Q8: Existed generative-model-based works only generate synthetic samples that are similar to real defects, but do not effectively improve detection capabilities. Therefore, I am still concerned that generative focuses too much on some generated metrics, such as IS and IC-L, and ignores detection metrics.***
> > > >
> > > > > **A8**: Dear reviewer, thanks for your comment! As we mentioned in **Response A5**, we have done many experiments to **demonstrate the significant contribution of our generated samples to the anomaly detection performance of models**.
> > > >
> > > > > The results in Tab. 2, 3, 9, 10, 13-18, 31, and 32 demonstrate that segmentation models trained with our generated data outperform existing anomaly generation methods **across multiple datasets**, including MVTec-AD, MVTec3D-AD, and VisA. Tab. 11 and 12 further show the enhanced anomaly detection capabilities of our approach compared to state-of-the-art methods. Additionally, Tab. 33 and **Tab. R11** indicates that replacing existing generation strategies with ours leads to superior performance. These extensive and varied results clearly highlight that the improvements we achieve in anomaly detection beyond the generated metrics (e.g., IS and IC-L). They provide strong evidence of the practical benefits of our method in enhancing downstream detection tasks. We hope this can address your concern and underscore the significant contribution our approach makes to improving anomaly detection.

---

> ### Comment · Reviewer_EXr6 · 2024-12-03
>
> Thank you again for your detailed response. Although I still have doubts about this type of work, considering the author's positive response and extensive experimentation, I will increase my score to 6.

---

> > ### Author Response · Authors · 2024-12-03
> >
> > Thank you very much for your approval! Thanks again for your insightful comments and suggestions!

---

### Official Review · Reviewer_5auC · 2024-11-02

**Soundness:** 3
**Presentation:** 3
**Contribution:** 2
**Rating:** 5
**Confidence:** 5

**Summary:**

This paper presents an innovative approach called SeaS (Separation and Sharing Fine-tuning). SeaS is specifically designed to generate high-quality and diverse industrial anomaly images while using a minimal number of training examples. This is particularly important in industrial contexts, where anomaly images are often scarce.

At the heart of the SeaS methodology is a unique strategy that leverages both normal and abnormal images. This is accomplished through an Unbalanced Abnormal Text Prompt, which distinguishes between product tokens and anomaly tokens. This differentiation enables the generation of anomalies with a variety of attributes, enhancing the practical use of these images in real-world situations.

Additionally, the SeaS method incorporates two key loss functions that are essential to its framework. The first, Decoupled Anomaly Alignment loss, aims to increase the diversity of the generated anomalies. The second, Normal-image Alignment loss, ensures that the product images remain consistent and high-fidelity. Together, these elements strengthen the SeaS methodology, making it effective for producing high-quality industrial anomaly images.

**Strengths:**

1. The SeaS method presents an approach to few-shot learning specifically tailored for the generation of industrial anomaly images, a task that poses significant challenges due to the limited availability of such images. This method addresses the pressing need for effective anomaly generation in industrial settings, where data scarcity can hinder performance and innovation.
2. Central to the SeaS methodology is the introduction of the Unbalanced Abnormal (UA) Text Prompt, along with a systematic separation of anomaly tokens from product tokens. This strategy not only enhances the diversity of the generated anomalies but also improves their fidelity, thereby ensuring that the resulting images are both varied and accurate representations of real-world anomalies.
3. The paper offers an in-depth examination of two key loss functions: Decoupled Anomaly Alignment (DA) loss and Normal-image Alignment (NA) loss. The DA loss is designed to promote greater diversity among the generated anomalies, while the NA loss ensures that the product images maintain their consistency and high quality.

**Weaknesses:**

1. The examples provided in the article primarily focus on straightforward instances of image generation. They do not explore more complex scenarios, such as the cable swap and cable missing categories found in the MVTec dataset. These intricate examples necessitate a more advanced approach to image generation that can effectively address complicated anomalies and their effects on overall product appearance.
2. The SeaS framework, while innovative in its application of few-shot learning for industrial anomaly image generation, does not fundamentally differ from the AnomalyDiffusion framework. Both aim to tackle the challenge of generating diverse and realistic anomaly images from limited data. However, SeaS introduces distinctive mechanisms, such as the Unbalanced Abnormal Text Prompt and the Separation and Sharing Fine-tuning strategy. These innovations give SeaS an edge in generating anomalies with greater fidelity and diversity. Nonetheless, both frameworks share a common goal of enhancing anomaly generation capabilities, indicating a gradual evolution in the field of generative models for industrial applications rather than a revolutionary shift.
3. While the paper demonstrates the effectiveness of SeaS with simpler anomalies, it would be valuable for the authors to discuss its performance with more complex anomalies, such as the cable swap and cable missing examples from the MVTec dataset. Are there specific challenges or limitations when generating these complex anomalies? If the method has been tested with these examples, presenting those results could significantly strengthen the paper. If not, outlining potential strategies for addressing these complexities would be advantageous.
4. The paper positions SeaS as a superior approach compared to AnomalyDiffusion. What are the key factors that make SeaS more effective in certain aspects of industrial anomaly image generation? Are there scenarios in which AnomalyDiffusion may still offer advantages? A detailed comparative analysis or a side-by-side case study featuring both frameworks could clarify their respective strengths and weaknesses.

**Questions:**

1. How does SeaS handle generalization to new, unseen anomaly types in industrial images? Does the framework require retraining, or can it adapt to new anomalies using the same training strategy?
2. What are the computational resources needed for training the SeaS model, and how does this compare to existing methods in terms of training and inference times? Are there any optimizations implemented to accommodate large-scale datasets?
3.  How robust is the SeaS framework to variations in image quality, such as changes in lighting conditions, backgrounds, or resolutions? Does the model maintain consistent performance across these variations?

---

> ### Author Response · Authors · 2024-11-25
>
> ***Q1: The examples provided in the article primarily focus on straightforward instances of image generation. They do not explore more complex scenarios, such as the cable swap and cable missing categories found in the MVTec dataset. Are there specific challenges or limitations when generating these complex anomalies? If the method has been tested with these examples, presenting those results could significantly strengthen the paper. If not, outlining potential strategies for addressing these complexities would be advantageous.***
>
> >**A1**: Thanks for your insightful comment! There are no specific challenges or limitations for SeaS to generate these complex anomalies. The results on cable_swap and missing_cable were given in Fig. 13 of Appendix A.6 in the paper and Tab. R8 in the rebuttal. SeaS performed well on them.
>
> >The reason is that the self-attention mechanism in U-Net enables the model to effectively capture the structural information of products. During fine-tuning with abnormal images, the model leverages self-attention to learn the relationship between anomaly regions and normal regions, as well as their impact on the overall product appearance. Complex anomalies, such as logic anomalies (e.g., "cable swap" or "cable missing"), present challenges due to masks that often mark empty areas or lack valid texture information. U-Net's self-attention is essential for achieving a global receptive field, facilitating the generation of these complex anomalies. Relying solely on anomaly tokens makes it difficult to associate tokens with anomalies, reducing generation effectiveness.
>
> >**Table R8. The generation results on the complex anomalies on MVTec AD.**
> ||IS|IC-LPIPS|
> |:---:|:---:|:---:|
> |cable_swap|2.21|0.43|
> |missing_cable|2.60|0.38|
>
>
> ***Q2: The SeaS framework, while innovative in its application of few-shot learning for industrial anomaly image generation, does not fundamentally differ from the AnomalyDiffusion framework. Both aim to tackle the challenge of generating diverse and realistic anomaly images from limited data. However, SeaS introduces distinctive mechanisms, such as the Unbalanced Abnormal Text Prompt and the Separation and Sharing Fine-tuning strategy. These innovations give SeaS an edge in generating anomalies with greater fidelity and diversity. Nonetheless, both frameworks share a common goal of enhancing anomaly generation capabilities, indicating a gradual evolution in the field of generative models for industrial applications rather than a revolutionary shift.***
>
> >**A2**: Thanks for your comment! Despite their shared goal of generating diverse and realistic anomaly images from limited data, SeaS introduces several key innovations that differentiate it from AnomalyDiffusion, achieving several advantages.
>
> >**Generating products and anomalies at the same time**. AnomalyDiffusion is essentially an image editing algorithm that does not require generating products. SeaS generates products and anomalies at the same time, which additionally ensures the uniqueness of the product in each output.
>
> >**No mask input**. SeaS does not require mask input, which reduces the cost of obtaining input data. However, AnomalyDiffusion requires a mask along with the RGB image, which may cause concerns about the misalignment between products and masks.
>
> ***Q3: The paper positions SeaS as a superior approach compared to AnomalyDiffusion. What are the key factors that make SeaS more effective in certain aspects of industrial anomaly image generation? Are there scenarios in which AnomalyDiffusion may still offer advantages? A detailed comparative analysis or a side-by-side case study featuring both frameworks could clarify their respective strengths and weaknesses.***
>
> >**A3**: Thanks for your comment! We clarify the key differences between SeaS and AnomalyDiffusion in terms of their overall framework, and application scenarios.
>
> > **Overall Framework.** As highlighted in Fig. 1 of the paper, the key difference between AnomalyDiffusion and SeaS is that **AnomalyDiffusion is an editing approach**, that needs the mask as input, and edits the anomalies onto a given normal product image based on the given mask, the SD is fixed. While **SeaS is a fine-tuning approach**, during training, it fine-tunes the U-Net in SD by using a few anomaly images. The ability to generate diverse anomalies and accurate alignment of anomalies and masks makes SeaS more effective for industrial anomaly image generation.
>
> > **Application Scenarios.** While SeaS is more versatile for generating a wide variety of anomaly images and is well-suited for training downstream segmentation models, AnomalyDiffusion may still be the preferred choice when the goal is precise anomaly editing. For example, if a user needs to insert a specific defect at a certain location on a product image, AnomalyDiffusion is better suited for this task due to its image editing framework.

---

> > ### Author Response · Authors · 2024-11-25
> >
> > ***Q4: How does SeaS handle generalization to new, unseen anomaly types in industrial images? Does the framework require retraining, or can it adapt to new anomalies using the same training strategy?***
> >
> > >**A4**: Thank you for the comment! As we mentioned in Lines 22-24 in the paper. ***"Recombining the decoupled attributes may produce anomalies that have never been seen in the training dataset"***.  Such a conclusion can also be observed in Fig. 3(c) in the paper. SeaS is able to synthesize never-seen-before anomalies by recombining decoupled attributes. For the new anomaly types, the framework requires retraining to learn a new set of anomaly tokens in our Unbalanced Abnormal Text Prompt corresponding to the new anomalies using the same training strategy.
> >
> >
> > ***Q5: What are the computational resources needed for training the SeaS model, and how does this compare to existing methods in terms of training and inference times? Are there any optimizations implemented to accommodate large-scale datasets?***
> >
> > >**A5**: Thanks for your comment. As we mentioned in Appendix A.3 in the paper, SeaS trains a generation model on a single NVIDIA Tesla A100 40G GPU, which only requires about 20G GPU memory.
> >
> > >The computational resource cost comparisons were given in Tab. 6 and Tab. 7 of the appendix. For the MVTecAD datasets with 73 anomaly types, our training takes 73 hours, which is only **30%** of the time of the AnomalyDiffusion (249 hours) and **18%** of the time of DFMGAN (414 hours). In terms of inference time, SeaS costs 720 ms per image, which is only **19%** of the 3830 ms per image required by the diffusion-based method AnomalyDiffusion. The inference time of the GAN-based method DFMGAN is 48ms per image. In addition, SeaS has not been optimized for handling large-scale datasets, and we will further optimize this in future work.
> >
> >
> > ***Q6: How robust is the SeaS framework to variations in image quality, such as changes in lighting conditions, backgrounds, or resolutions? Does the model maintain consistent performance across these variations?***
> >
> > >**A6**: Thanks for the comment. We chose one defect class from peach, a product in the MVTec3D dataset, that has significant variations in lighting conditions and backgrounds, to conduct experiments. Images with strong lighting conditions depict the top side of the peach, whereas those with weak lighting conditions show the bottom side. Consequently, the background in the images, whether the top or bottom of the peach, also differs. We selected three training sets with different lighting conditions for experiments: 1) only images from the top side with strong lighting condition, 2) only images from the bottom side with weak lighting condition,  3) half of the images from the top side with strong lighting condition, and a half from the bottom side with weak lighting condition. As shown in Tab. R9, the variations have a slight effect on the authenticity of generated images. The generated images of different settings are shown in Fig. 31 in the appendix. It can be seen that SeaS is robust against lighting conditions and background variations. Regarding the resolution change, to match the pre-trained StableDiffusion model dimensions, we  resize the image to 512 × 512 size before training.
> >
> >
> > >**Table R9. The image generation quality on lighting conditions.**
> > |Lighting Condition|Background|IS|IC-LPIPS|
> > |:---:|:---:|:---:|:---:|
> > |Strong|Top_side|1.53|0.25|
> > |Weak|Bottom_side|2.18|0.27|
> > |Both|Both|1.70|0.24|

---

### Official Review · Reviewer_78Lv · 2024-11-02

**Soundness:** 3
**Presentation:** 3
**Contribution:** 3
**Rating:** 6
**Confidence:** 4

**Summary:**

This paper proposes a network structure for simultaneously generating anomaly images and their corresponding mask annotations. This is achieved through the following elements within a shared U-Net architecture: 1. Specialized text tokens; 2. Distinct loss functions tailored for anomaly and normal images; and 3. A mask prediction module designed for precise anomaly mask generation. Together, these components ensure the randomness of anomalies, the global consistency of products, and high-quality mask refinement.

**Strengths:**

Originality: The paper presents an innovative approach to industrial anomaly image generation by designing specialized object and anomaly tokens. This allows a shared model to train across multiple objects and anomalies, which is a unique solution in the field.
Quality: The network structure is carefully designed, combining tailored loss functions for anomaly and normal images along with a mask prediction module.
Clarity: The paper is clearly written and easy to follow, with well-organized sections and explanations that make the complex methodology accessible to readers.
Significance: Extensive experiments substantiate the method’s effectiveness, showcasing significant improvements over prior approaches.

**Weaknesses:**

Clarity in Methodology: The current fixed anomaly token setup (N=4) does not allow explicit control over specific types of anomalies, particularly in mixed anomaly cases (e.g., mixed anomalies in cable). This limitation reduces the flexibility of the model in generating targeted anomaly types.
Effectiveness of the Method: Unlike AnomalyDiffusion [1], which leverages the text inversion technique [2] without needing to fine-tune the U-Net, this method requires U-Net fine-tuning. This approach might reduce the domain gap for anomaly data, potentially contributing to the improved quality of generated images. However, it would be helpful to clarify the effect of freezing versus fine-tuning the U-Net on image quality.
Ambiguity in the Inference Process: The inference process in this paper lacks clarity, making it difficult to understand how to specify a particular anomaly type.
[1] AnomalyDiffusion: Few-Shot Anomaly Image Generation with Diffusion Model
[2] An Image is Worth One Word: Personalizing Text-to-Image Generation using Textual Inversion

**Questions:**

1. The paper currently sets a fixed number of anomaly tokens (N=4). This appears to limit control over generating specific anomaly types, especially for mixed anomaly cases like those in the "cable" category. Is this interpretation correct, or is there a way to dynamically adjust or control anomaly types within this setup? Further clarification on this point would be very helpful.
2. Since this method requires fine-tuning the U-Net while AnomalyDiffusion [1] uses text inversion [2] without U-Net adjustments, could the authors clarify whether the observed quality improvement in anomaly generation stems from this fine-tuning? A comparison between freezing and fine-tuning the U-Net would be insightful
3. Could the authors provide additional details on the inference process, specifically on how one could specify or target a particular anomaly type during generation?

---

> ### Author Response · Authors · 2024-11-24
>
> ***Q1: The current fixed anomaly token setup (N=4) does not allow explicit control over specific types of anomalies, particularly in mixed anomaly cases (e.g., mixed anomalies in cable). This limitation reduces the flexibility of the model in generating targeted anomaly types. Is this interpretation correct, or is there a way to dynamically adjust or control anomaly types within this setup? Further clarification on this point would be very helpful.***
>
> >**A1**: Thanks for your meticulous comment! This interpretation needs to be recorrected as below:
>
> >Our method allows for explicit control over specific types of anomalies through the use of different condition text prompts, denoted as $\mathcal{P_n}$. As we mentioned in lines 292-293 in the paper, these prompts are designed with different sets of anomaly tokens, which guide the training and generation of each anomaly type. The prompt is defined as:
>
> >$$\mathcal{P_n} = \text{a <ob> with } \text{<}\text{df}_{4 × n-3}\text{>,}\text{<}\text{df}_{4 × n-2}\text{>,} \text{<}\text{df}_{4 × n-1}\text{>,} \text{<}\text{df}_{4 × n}\text{>}$$
>
> >where $n$ represents the index of the anomaly types in the product. For instance, for the *"cable"* product, the prompts would be  $\mathcal{P_1} = \text{a <ob> with } \text{<}\text{df}_{1}\text{>,}\text{<}\text{df}_{2}\text{>,} \text{<}\text{df}_{3}\text{>,} \text{<}\text{df}_{4}\text{>}$ for the training and generation of the first anomaly type "*bent_wire"*, and $\mathcal{P_2} = \text{a <ob> with } \text{<}\text{df}_{5}\text{>,}\text{<}\text{df}_{6}\text{>,} \text{<}\text{df}_{7}\text{>,} \text{<}\text{df}_{8}\text{>}$ for the second anomaly type "*cable_swap*". For the "*combined*" anomaly category of "*cable*", the training and inference methods are the same as for other types of anomalies. Thus, our method provides flexibility in adjusting and controlling the specific types of anomalies by tailoring the prompt set accordingly.
>
>
> ***Q2:Unlike AnomalyDiffusion [1], which leverages the text inversion technique [2] without needing to fine-tune the U-Net, this method requires U-Net fine-tuning. This approach might reduce the domain gap for anomaly data, potentially contributing to the improved quality of generated images. However, it would be helpful to clarify the effect of freezing versus fine-tuning the U-Net on image quality.
> Since this method requires fine-tuning the U-Net while AnomalyDiffusion [1] uses text inversion [2] without U-Net adjustments, could the authors clarify whether the observed quality improvement in anomaly generation stems from this fine-tuning? A comparison between freezing and fine-tuning the U-Net would be insightful.***
>
> >**A2**: Thanks for your insightful comment! We agree with your opinion that "Fine-tuning the U-Net contributes to the improved quality of generated images". Fine-tuning the U-Net is essential for the **AIG** method.
>
> >To investigate the role of fine-tuning, we conducted the experiment of only using the Textual Inversion (TI) method to learn the product without fine-tuning the U-Net. The results, shown in Fig. 30 in the appendix, demonstrate that the TI method struggles to generate images similar to the real product due to the limited number of learnable parameters. In contrast, when fine-tuning the U-Net as part of the **AIG** method, the generated images exhibit significant improvements in both quality and global consistency. This suggests that the observed improvement in image quality and anomaly generation can largely be attributed to the fine-tuning of the U-Net.
>
> >As we mentioned in Lines 83-84 of the paper, "*the products satisfy global consistency with minor variations in local details, while the anomalies hold randomness,*" the generated products should be globally consistent with the real products. Therefore, unlike the **AG** method Anomalydiffusion, which does not need to consider the generation of the product, we fine-tune the U-Net.
>
> ***Q3: The inference process in this paper lacks clarity, making it difficult to understand how to specify a particular anomaly type.
> Could the authors provide additional details on the inference process, specifically on how one could specify or target a particular anomaly type during generation?***
>
> >**A3**:  Thanks for your comment! During inference, we use different condition text prompts $\mathcal{P_n}$ to specify the particular anomaly type of the product. We explained the design of different text prompts in **Response A1**. For example, aiming to generate "*cable_swap"*, which is the second anomaly type for the *"cable"*  product, we have to set the condition prompt as  $\mathcal{P_2} = \text{a <ob> with } \text{<}\text{df}_{5}\text{>,}\text{<}\text{df}_{6}\text{>,} \text{<}\text{df}_{7}\text{>,} \text{<}\text{df}_{8}\text{>}$ during inference.

---

### Official Review · Reviewer_RGGH · 2024-11-04

**Soundness:** 2
**Presentation:** 3
**Contribution:** 3
**Rating:** 6
**Confidence:** 4

**Summary:**

This paper proposes a Separation and Sharing Fine-tuning (SeaS) approach for few-shot industrial anomaly image generation. The proposed Decoupled Anomaly Alignment (DA) loss and Normal-image Alignment (NA) loss achieve the generation of highly-diverse anomalies and globally-consistent products. Moreover, the author designs the Refined Mask Prediction (RMP) module to produce pixel-wise anomaly annotations. Extensive experiments show the effectiveness of the method.

**Strengths:**

In general, the proposed method is reasonable and the results are fine.

+ Both anomaly image generation and anomaly segmentation are considered.
+ This paper is clearly presented.
+ The proposed method achieves realistic and diverse generation of abnormal samples.

**Weaknesses:**

- The word “Unbalanced” in Unbalanced Abnormal Text Prompt is inappropriate. The author believes fixed generic semantic words may fail to align with a few training images that contain specific defect types. Therefore, the text prompt should be expressed as dynamic or learnable, etc.
- The author should add the results of Baseline in Table 4, such as generation of typical text prompt with fixed generic semantic words. Moreover, the result with different layers in UNet in RMP branch should be discussed.
- The authors should replace the abnormal synthesis strategy in the existing unsupervised methods with the proposed generation strategy to prove the effectiveness of the proposed method, such as DRAEM, because the methods in the table 2 are not specifically designed for anomaly detection.
- Implementation details of inference should be described, such as inference step and guidance scale, which is critical to the quality of the generation.
- The latest generation method [1] should be compared.
- More datasets, such as VisA or RealIAD, are suggested to used for further evaluation of the proposed method.


[1] Few-Shot Anomaly-Driven Generation for Anomaly Detection. 2024.

**Questions:**

- The word “Unbalanced” in Unbalanced Abnormal Text Prompt is inappropriate. The author believes fixed generic semantic words may fail to align with a few training images that contain specific defect types. Therefore, the text prompt should be expressed as dynamic or learnable, etc.
- The author should add the results of Baseline in Table 4, such as generation of typical text prompt with fixed generic semantic words. Moreover, the result with different layers in UNet in RMP branch should be discussed.
- The authors should replace the abnormal synthesis strategy in the existing unsupervised methods with the proposed generation strategy to prove the effectiveness of the proposed method, such as DRAEM, because the methods in the table 2 are not specifically designed for anomaly detection.
- Implementation details of inference should be described, such as inference step and guidance scale, which is critical to the quality of the generation.
- The latest generation method [1] should be compared.
- More datasets, such as VisA or RealIAD, are suggested to used for further evaluation of the proposed method.


[1] Few-Shot Anomaly-Driven Generation for Anomaly Detection. 2024.

---

> ### Author Response · Authors · 2024-11-24
>
> ***Q1: The word “Unbalanced” in Unbalanced Abnormal Text Prompt is inappropriate. The author believes fixed generic semantic words may fail to align with a few training images that contain specific defect types. Therefore, the text prompt should be expressed as dynamic or learnable, etc.***
>
> >**A1**: Thanks for your comment! As we mentioned in lines 214-215 of the paper, "unbalanced" means using multiple anomaly tokens while only employing one normal token, rather than referring to the learnability of the prompt.
>
>
> ***Q2: The author should add the results of Baseline in Table 4, such as generation of typical text prompt with fixed generic semantic words.***
>
> >**A2**: Thanks for your comment! We included the baseline result (short for *with TP* ) in Tab. 4 and Tab. R5 in the rebuttal. The results demonstrate that using a typical text prompt with fixed generic semantics words leads to a decrease in the fidelity and diversity of the generated images, as well as in segmentation performance. This outcome proves the effectiveness of the text prompt we proposed.
>
> >**Table R5. More ablation on the generation model.**
> ||IS|IC-L|AUROC|AP|F1-max|IoU|
> |:---:|:---:|:---:|:---:|:---:|:---:|:---:|
> |*with TP*|1.72|0.33|94.72|57.16|55.67|50.46|
> |**Ours**|**1.88**|**0.34**|**97.21**|**69.21**|**66.37**|**55.28**|
>
>
> ***Q3: Moreover, the result with different layers in UNet in RMP branch should be discussed.***
>
> >**A3**: Thanks for your comment! We have provided the results with different layers of U-Net in the RMP branch in Table 25 of Appendix A.8. Our experiments show that "employing a combination of {*F2*,*F3*} for coarse feature extraction, achieves the best performance in downstream segmentation task", which explains our selection of these layers for the RMP branch.
>
>
> ***Q4: The authors should replace the abnormal synthesis strategy in the existing unsupervised methods with the proposed generation strategy to prove the effectiveness of the proposed method, such as DRAEM, because the methods in the table 2 are not specifically designed for anomaly detection.***
>
> >**A4**: Thanks for your comment! We follow your instructions to replace the abnormal synthesis strategy in DRAEM with the proposed generation strategy, the results are given in Tab. R6.  The segmentation result demonstrates that our method outperforms DRAEM, with improvements of **+9.46 %** in AP, **+7.23%** in F1-max and **+3.69%** in IoU.
>
>
> >**Table R6. Comparison with unsupervised method on MVTec AD.**
> ||image-level|||pixel-level|||||
> |:---:|:---:|:---:|:---:|:---:|:---:|:---:|:---:|:---:|
> ||AUROC|AP|F1-max|AUROC|AP|F1-max|PRO|IoU|
> |DRAEM|98.00|98.45|96.34|97.90|67.89|66.04|91.49|60.30|
> |**SeaS**|**99.25**|**99.66**|**98.35**|**97.98**|**77.35**|**73.27**|**93.79**|**63.99**|
>
>
> ***Q5: Implementation details of inference should be described, such as inference step and guidance scale, which is critical to the quality of the generation.***
>
> >**A5**: Thanks for your comment! For all experiments, we use t = 1500 to perform diffusion forward on normal images to get the initial noise. We employ T = 25 steps of sampling with a classifier-free guidance strength set to 8. To generate anomalies of a specific category, we use the same condition corresponding to that specific category of anomalies during training.
>
>
> ***Q6: The latest generation method [1] should be compared.***
>
> >**A6**: Thanks for your comment! Following your instructions, we substituted the abnormal synthesis strategy in AnoGen [1] with the proposed generation strategy. The results, which are presented in Tab. R7, show that our method surpasses AnoGen, achieving improvements of **+5.75%** in AP, **+4.09%** in F1-max, and **+3.07%** in IoU.
>
> >**Table R7. Comparison with the latest generation method on MVTec AD.**
> ||image-level|||pixel-level|||||
> |:---:|:---:|:---:|:---:|:---:|:---:|:---:|:---:|:---:|
> ||AUROC|AP|F1-max|AUROC|AP|F1-max|PRO|IoU|
> |AnoGen|98.28|99.31|97.70|97.79|71.60|69.18|91.92|60.92|
> |**SeaS**|**99.25**|**99.66**|**98.35**|**97.98**|**77.35**|**73.27**|**93.79**|**63.99**|
>
>
>
> ***Q7: More datasets, such as VisA or RealIAD, are suggested to used for further evaluation of the proposed method.***
>
> >**A7**: Thanks for your suggestion! The new results on the VisA dataset are given in the ***General Response.*** Our method outperforms others across all the segmentation models, with an **11.71%** average improvement on IoU.

---

### Official Review · Reviewer_uoAo · 2024-11-04

**Soundness:** 2
**Presentation:** 3
**Contribution:** 2
**Rating:** 5
**Confidence:** 5

**Summary:**

This paper proposes a separation and sharing fine-tuning (SeaS) approach with a few abnormal and some normal images to produce anomalies and pixel-wise annotations. An unbalanced abnormal (UA) text prompt is introduced for anomaly generation, which consists of one product token and several anomaly tokens. For anomaly images, a decoupled anomaly alignment (DA) loss is used to bind the attributes of the anomalies to different anomaly tokens. For normal images, a normal-image alignment (NA) loss is used to learn the products’ key features that are used to synthesize products with both global consistency and local variations. Experiments on MVTec AD and MVTec 3D AD datasets (RGB images) show the effectiveness of the proposed method for anomaly image generation and detection.

**Strengths:**

+ The motivation is good. A shared generation model for multiple anomaly types is proposed to solve the problem of insufficient anomaly images.

+ The generated anomaly images seem more real than other GAN-based methods.

+ Some ablation studies are provided to facilitate the understanding of how the performance benefits from different components, including the DA loss, the NA loss, and the refined mask prediction branch.

**Weaknesses:**

- The experimental results are insufficient. Because the ultimate goal of generating abnormal images is to improve the performance of anomaly detection tasks, some SOTA anomaly detection methods should also be compared on image AUROC, pixel AUROC and PRO besides generative model-based anomaly detection methods, e.g., DiAD [1]. Although RealNet [2] is compared in appendix A.5, the proposed method does not significantly outperform RealNet, particularly AUROC, and RealNet does not use any anomaly samples during training.

- Since the proposed method is not specific to the multi-class anomaly detection setting, I would also wonder about the comparison with SOTA methods in the single-class anomaly detection setting, especially supervised/semi-supervised methods, e.g., PRN [3] or BGAD [4], because anomaly samples are used during training of the proposed method.

- More datasets are required to evaluate the proposed anomaly generation method on image AUROC, pixel AUROC and PRO, such as VisA dataset containing more diverse and tiny anomalies. It is more challenging to generate these anomalies.

- What does the “unbalanced” mean in Sec. 3.2? The paper does not clearly explain its meaning. The authors should clearly define or explain this term when it is first introduced, and discuss its significance to the overall approach.

- In Fig.5, for Wood color, the results generated by the proposed method do not seem better than the results of AnomalyDiffusion.

[1] H. He, et al., A diffusion based framework for multi-class anomaly detection, AAAI 2024
[2] X. Zhang, et al., Realnet: feature selection network with realistic synthetic anomaly for anomaly detection, CVPR 2024.
[3] H. Zhang, et al., Prototypical residual networks for anomaly detection and localization, CVPR 2023
[4] X. Yao, et al., Explicit boundary guided semi-push-pull contrastive learning for supervised anomaly detection, CVPR 2023.

**Questions:**

1. Because the ultimate goal of generating abnormal images is to improve the performance of anomaly detection tasks, some SOTA anomaly detection methods should also be compared on image AUROC, pixel AUROC and PRO besides generative model-based anomaly detection methods, e.g., DiAD [1]. Although RealNet [2] is compared in appendix A.5, the proposed method does not significantly outperform RealNet, particularly on AUROC, and RealNet does not use any anomaly samples during training.

2. Since the proposed method is not specific to the multi-class anomaly detection setting, I would also wonder about the comparison with SOTA methods in the single-class anomaly detection setting, especially supervised/semi-supervised methods, e.g., PRN [3] or BGAD [4], because anomaly samples are used during training of the proposed method.

3. More datasets are required to evaluate the proposed anomaly generation method on image AUROC, pixel AUROC and PRO, such as VisA dataset containing more diverse and tiny anomalies. It is more challenging to generate these tiny anomalies.

4. What does the “unbalanced” mean in Sec. 3.2? The authors should clearly define or explain this term when it is first introduced, and discuss its significance to the overall method.

5. In Fig.5, for Wood color, the results generated by the proposed method do not seem better than the results of AnomalyDiffusion.  Are there particular challenges with the Wood color anomaly type for the proposed method?

---

> ### Author Response · Authors · 2024-11-25
>
> ***Q1: Some SOTA anomaly detection methods should also be compared besides generative model-based anomaly detection methods, e.g., DiAD [1]. Although RealNet [2] is compared in appendix A.5, the proposed method does not significantly outperform RealNet.***
>
> >**A1**: Thanks for your comment!
>
> >First, with the training samples generated by SeaS, the conventional lightweight segmentation models surpass existing SOTA anomaly detection methods, as shown in Tab. 11 and Tab. 12. Moreover, we added the comparison with the non-generative model-based anomaly detection method in Tab. 12,  the result of *"SeaS + UperNet"* outperforms the DiAD [1] method on all metrics. For RealNet [2], as we mentioned in Lines 1177-1179 in Appendix A.5, *"Real-Net contains 591M parameters, around **177** times larger than BiSeNet V2, and **631** times larger than LFD, while the pixel-level AP and IoU measures of Real-Net are even worse than those of BiSeNet V2 and LFD. Although the pixel-level AUROC metric, which is more sensitive to false negatives than to false positives, is slightly higher for Real-Net, we observe that it generates a high number of false positives, substantially reducing pixel-level AP, F1-max, and IoU scores. "* For effective industrial anomaly detection, a method should be small-scale, and balance false positives and false negatives.
>
> >Additionally, in Tab. R10, we compare the results of training BiSeNet V2 using only the original data from MVTec AD with using the generated anomaly image-mask pairs from SeaS. The results show significant improvements across all metrics when using the proposed generation approach, further demonstrating that our method greatly enhances the performance of downstream tasks.
>
>
> >**Table R10. Comparison with only using original training data on MVTec AD.**
> ||image-level|||pixel-level||||
> |:---:|:---:|:----:|:---:|:---:|:---:|:---:|:---:|
> ||AUROC|AP|F1-max|AUROC|AP|F1-max|IoU|
> |BiSeNet V2|68.89|80.51|82.22|70.35|23.02|28.72|15.96|
> |SeaS + BiSeNet V2|**96.00**|**98.14**|**95.43**|**97.21**|**69.21**|**66.37**|**55.28**|
>
> ***Q2: Since the proposed method is not specific to the multi-class anomaly detection setting, I would also wonder about the comparison with SOTA methods in the single-class anomaly detection setting, especially supervised/semi-supervised methods, e.g., PRN [3] or BGAD [4], because anomaly samples are used during training of the proposed method.***
>
> >**A2**: Thanks for your comment! In terms of the downstream supervised segmentation task, as we mentioned in Lines 475-476 in the paper, we train a **single unified model** for the  segmentation task in a **multi-class** setting. While our method isn't specifically tailored for single-class anomaly detection, we have conducted comparisons with existing state-of-the-art methods and demonstrated that our model outperforms them.
>
> >To elaborate, since the official source code for PRN [3] is unavailable, we compared our method using the reported metrics from Tab. 2 and Tab. 3 of AnomalyDiffusion. We included this comparison in Tab. 12 of our paper, where our method shows superior performance compared to PRN. As for BGAD [4], we replaced its pseudo-anomaly generation strategy with our proposed generation approach. Experimental results, presented in Tab. R4, show that our approach outperforms BGAD as well.
>
>
> >**Table R4. Comparison with supervised method on MVTec AD.**
> ||image-level|||pixel-level||||
> |:---:|:---:|:---:|:---:|:---:|:---:|:---:|:---:|
> ||AUROC|AP|F1-max|AUROC|AP|F1-max|IoU|
> |BGAD|98.31|98.05|98.97|**99.26**|**73.85**|77.89|60.60|
> |SeaS+BGAD|**98.44**|**98.18**|**99.08**|**99.26**|**73.85**|**77.93**|**60.81**|
>
>
> ***Q3: More datasets are required to evaluate the proposed anomaly generation method on image AUROC, pixel AUROC and PRO, such as VisA dataset containing more diverse and tiny anomalies. It is more challenging to generate these tiny anomalies.***
>
> >**A3**: Thanks for your suggestion! The new results on the VisA dataset are given in the **General Response.** Our method outperforms others across all the segmentation models, with an **11.71%** average improvement on IoU.

---

> > ### Author Response · Authors · 2024-11-25
> >
> > ***Q4: What does the “unbalanced” mean in Sec. 3.2? The paper does not clearly explain its meaning. The authors should clearly define or explain this term when it is first introduced, and discuss its significance to the overall approach.***
> >
> > >**A4**: Thanks for your comment! As we mentioned in Lines 214-215 of the paper, "unbalanced" means using multiple Anomaly Tokens while only employing one Nomal Token. Moreover, the significance of the design of the unbalanced abnormal text prompt is given in Lines 238-240 of the paper, *"Experimental observations indicate that one $<\text{ob}>$  is sufficient to express the normal product, while multiple $<\text{df}>$ are necessary for controlling the generation of the anomalies."*
> >
> > >In the revised manuscript, we have highlighted the meaning of "unbalanced" when it is first introduced.
> >
> >
> >
> > ***Q5: In Fig.5, for Wood color, the results generated by the proposed method do not seem better than the results of AnomalyDiffusion. Are there particular challenges with the Wood color anomaly type for the proposed method?***
> >
> > >**A5**: Thanks for your comment! This type of anomaly does not pose any particular challenge for SeaS. As is shown in Lines 469-470 of the paper, *"SeaS can generate images with different types, colors, and shapes of anomalies rather than overfitting to the training images (e.g., wood color and pill crack)"*.  We chose to showcase such generated images of *wood color* in Fig. 5 to illustrate that our model is capable of **generating more diverse anomalies**, which helps to improve the generalization capability of downstream segmentation models. In contrast, the defect appearance and masks of the *wood color* image generated by AnomalyDiffusion in Fig. 5 are nearly identical to those of the training samples shown in the first column, lacking diversity. Such overfitting to the training images could adversely affect the generalization ability of downstream segmentation models.
> >
> > >Furthermore, as shown in Fig. 3 (c)(i), SeaS is also capable of generating anomalies that are similar to the training data.

---

### Official Review · Reviewer_B56D · 2024-11-05

**Soundness:** 3
**Presentation:** 2
**Contribution:** 3
**Rating:** 8
**Confidence:** 4

**Summary:**

The paper proposes a new method named Seas to solve the few-shot anomaly image generation problem. It leverages the stable diffusion model and VAE to generate anomaly images with accurate annotations. The experiment results show the effectiveness of their method.

**Strengths:**

1.	Leveraging text prompts to guide the model in decoupling the generation of abnormal regions and objects.
2.	Using VAE to generate high-resolution annotations is a good direction.

**Weaknesses:**

1.	The relationship between anomaly tokens and training different types of anomalies is not clear.
2.	The paper has not discussed how to control the type of exceptions generated during inference.
3.	The paper does not explain why the U-net used to predict noise in the Refined Mask Prediction branch has a highly discriminative feature.

**Questions:**

1. I want to clarify the relationship between exception marking and training different types of exceptions.
2. I want to know how to control the types of exceptions generated during inference.
3. I wonder why U-net, which is used to predict noise, has highly discriminative features in the fine mask prediction branch.

---

> ### Author Response · Authors · 2024-11-24
>
> ***Q1: The relationship between anomaly tokens and training different types of anomalies is not clear. I want to clarify the relationship between exception marking and training different types of exceptions.***
>
> >**A1**: Thanks for your comment! As we mentioned in Lines 1433-1436 in Appendix A.8, we use 4 $<\text{df}>$ as a set for each anomaly type, and 1 $<{\text{ob}}>$ for each product class.
> >Taking the product "*cable*" with 8 anomaly types as an example, there are 1 <ob> and 8 different token sets for $<\text{df} _ n>$ , i.e., {$<\text{df} _ 1>$$<\text{df} _ 2>$,$<\text{df} _ 3>$,$<\text{df} _ 4>$, {$<\text{df} _ 5>$,$<\text{df} _ 6>$,$<\text{df} _ 7>$,$<\text{df} _ 8>$},...,{$<\text{df} _ {29}>$,$<\text{df} _ {30}>$,$<\text{df} _ {31}>$,$<\text{df} _ {32}>$}. Each token set contains 4 tokens corresponding to one anomaly type of "*cable*".
>
>
> ***Q2: The paper has not discussed how to control the type of exceptions generated during inference.***
>
> >**A2**:  Thanks for your insightful comment! During inference, for a particular type of anomaly, we use the Unbalanced Abnormal (UA) Text Prompt $\mathcal{P _ n}$ with different sets of anomaly tokens as the condition to generate the specified type of anomaly.
>
> >$\mathcal{P_n} = \text{a <ob> with } \text{<}\text{df}_{4 × n-3}\text{>,}\text{<}\text{df}_{4 × n-2}\text{>,} \text{<}\text{df}_{4 × n-1}\text{>,} \text{<}\text{df}_{4 × n}\text{>}$
>
> >where $n$ represents the index of the anomaly types in the product. For example, for the *"cable"* product, the prompts would be $\mathcal{P_1} = \text{a <ob> with } \text{<}\text{df}_{1}\text{>,}\text{<}\text{df}_{2}\text{>,} \text{<}\text{df}_{3}\text{>,} \text{<}\text{df}_{4}\text{>}$ for the training and generation of the first anomaly type "*bent_wire"*, $\mathcal{P_2} = \text{a <ob> with } \text{<}\text{df}_{5}\text{>,}\text{<}\text{df}_{6}\text{>,} \text{<}\text{df}_{7}\text{>,} \text{<}\text{df}_{8}\text{>}$ for the second anomaly type "*cable_swap".*
>
>
> ***Q3: The paper does not explain why the U-net used to predict noise in the Refined Mask Prediction branch has a highly discriminative feature.***
>
> >**A3**: Thanks for your comment! During the fine-tuning, we use Decoupled Anomaly Alignment loss to align the anomaly embeddings with anomaly region features, while simultaneously preventing the normal region features from aligning with these anomaly embeddings. This approach ensures that the features of the anomaly region are distinguishable from those of the normal region.
> >This can be observed in Fig. 29 of the appendix, we use the output features of the “up-2” and “up-3” layers in U-Net. After applying convolutional blocks and concatenation operations, these output features exhibit distinct responses between anomaly and normal regions, thereby demonstrating the high discriminability of the U-Net features.

---

> > ### Comment · Reviewer_B56D · 2024-11-27
> >
> > I have reviewed your supplementary experiments and appendix again. In addition to the concerns we mentioned earlier, you may need to consider more about the control of the type of anomalies generated by the model. For example, the idea of "generating unprecedented anomalies by recombining" is not enough just through some visualization, we would like to know where the bottom line of this model is. Because even though the variety of existing anomalies to synthesize is often limited, we may need more examples to support your idea of generating more "invisible new anomalies".
> >
> > In addition, I increased my score based on your added experimental points and explanations, but emphasized that you needed to further explain my concerns.

---

> ### Comment · Reviewer_B56D · 2024-11-27
>
> Thanks to the author for your response!
> For Q1 I suggest putting them in the main text to make it easier for readers to understand. I have no more questions about Q2.
>
> Also in Q3 I would like you to use more words to explain why such an approach does lead to "more discrimination" than before. In other words, I understand that you want to create "more discriminative feature representations, especially between normal and abnormal features". However, I would like you to re-emphasize the motivation of your model in this regard, especially why your model does better in this regard, to help the reader understand.
>
> A few others that don't matter:
> 1) Some of the images in the appendix may affect the visual impression, for example, the sample selection in Fig.7 is not correct, perhaps we would prefer to see different results of each model on the same sample.
> 2) We know that the depth map is a single-channel grayscale image, but we can actually stack it into a 3-channel image. I am curious what effect will be produced on the depth map of MVTec3D-AD? Are RGB and depth maps similar for the same sample? This is important for multimodal anomaly detection. Maybe you can show a few examples in the appendix to help people understand.
>
> Good luck again!
>
> I apologize for delaying the discussion due to my personal reasons, and I hope you will reply as soon as possible.

---

> > ### Author Response · Authors · 2024-11-29
> >
> > ***Q1: I suggest putting them in the main text to make it easier for readers to understand.***
> >
> > >**A1**: Thank you for your suggestion! We appreciate your feedback and have integrated them into the main text to improve clarity for readers.
> >
> >
> > ***Q3: I would like you to use more words to explain why such an approach does lead to "more discrimination" than before. In other words, I understand that you want to create "more discriminative feature representations, especially between normal and abnormal features". However, I would like you to re-emphasize the motivation of your model in this regard, especially why your model does better in this regard, to help the reader understand.***
> >
> > >**A3**: Thank you for your insightful comment! As we discussed in Lines 209-211, one of the unique challenges in industrial scneario is that the limited number of training images containing specific defect types may not align well with the fixed, generic semantic words. This misalignment limits the expressiveness of the model's learned features, as the model may struggle to map the fixed words to the subtle and context-dependent characteristics of the defects.
> >
> > >To address this challenge, our method leverages a **Decoupled Anomaly Alignment loss**, to specifically aligns anomaly tokens with abnormal areas in the images, which ensures that the **learnable anomaly embeddings are directly bound to the relevant anomaly features**. Additionally, the **unbalanced design** of using a single normal token and multiple anomaly tokens further enhances the discriminative power of the anomaly features. By emphasizing the anomaly features in this way, the model is able to differentiate between normal and abnormal regions, leading to more distinct and discriminative feature representations.
> >
> >
> > ***Q4: Some of the images in the appendix may affect the visual impression, for example, the sample selection in Fig.7 is not correct, perhaps we would prefer to see different results of each model on the same sample.***
> >
> > >**A4**: Thank you for your comment! Regarding the comparison of different models, we used the same set of training samples for evaluation, and the generated data in each column corresponds to the same defect category of the product. To provide a clearer comparison, we have replaced the current set of images in Fig. 7 to better illustrate the differences between these methods.
> >
> >
> > ***Q5: We know that the depth map is a single-channel grayscale image, but we can actually stack it into a 3-channel image. I am curious what effect will be produced on the depth map of MVTec3D-AD? Are RGB and depth maps similar for the same sample? This is important for multimodal anomaly detection. Maybe you can show a few examples in the appendix to help people understand.***
> >
> > >**A5**: Thank you for your insightful comment! Our SeaS focuses on 2D image generation, so we only use the RGB modal in the MVTec 3D-AD dataset. Although the 3D depth map is a single-channel grayscale map, each pixel represents the distance from each point of the point cloud to the 3D industrial sensor, which has a large gap compared with the meaning of 2D RGB image. So when directly using the StableDiffusion pre-trained by RGB images, the 3D data is mapped to a different latent space than 2D data, which results in 2D generation methods not working well in the 3D data. Thank you for your very enlightening suggestion again. In our future work, we will consider designing some new techniques to generate 3D data.

---

> ### Author Response · Authors · 2024-11-29
>
> ***Q6: In addition to the concerns we mentioned earlier, you may need to consider more about the control of the type of anomalies generated by the model. For example, the idea of "generating unprecedented anomalies by recombining" is not enough just through some visualization, we would like to know where the bottom line of this model is. Because even though the variety of existing anomalies to synthesize is often limited, we may need more examples to support your idea of generating more "invisible new anomalies".***
>
> >**A6**: Thank you for your meticulous comment! We fully acknowledge the importance of controlling the types of anomalies generated by the model. In future work, we plan to explore attribute decoupling and recombination principles more deeply, with the goal of enabling precise control over individual anomaly attributes such as color, shape, and texture.
>
> >To clarify the potential of generating "invisible new anomalies", we have provided additional examples in Fig. 32 of the appendix, where the model generates anomalies that differ significantly from the training data in terms of both color and shape.
> For instance, we present new examples such as *bottle\_contamination*, *hazelnut\_print*, and *tile\_gray\_stroke*,  which introduce novel shapes. We also show *wood\_color* and *metal\_nut\_scratch*, which differ in color, and *pill\_crack*, which introduces a new shape with multiple cracks, whereas the training samples contained only single cracks. These examples illustrate the model’s capacity to generate unseen anomalies by recombining learned attributes in innovative ways, going beyond the limitations of the original anomaly diversity in the training set.

---

> ### Comment · Reviewer_B56D · 2024-11-29
>
> Thank you for your answer. I'm glad you answered or practiced Q1 and Q4. I still recommend writing more clearly and explaining the motivation, including putting Q3 into the body of the text and describing it in more detail (although this is a suggestion that does not affect the rating). On the other hand, for Q5 I am just a suggestion, maybe the model can adapt to the two modalities of RGBD, and the generated exceptions can correspond to one of the challenges that our field needs to face.
>
> I may have some objections to Q6, just as you said, "recombining way" to generate exceptions, which may cause some misunderstandings. For example, this way is easy to be misunderstood as the weighted generation of existing exceptions, I hope to explain further in the appendix.
>
> There are also some simple questions:
> As other reviewers have said, although you perform well on some anomaly generation metrics, your main task still needs to focus on anomaly detection. Your performance on some anomaly detection metrics is not so impressive. Maybe you need to balance the two points more.
>
> Based on the 8 score I've improved, I stay the same.

---

> > ### Author Response · Authors · 2024-12-03
> >
> > ***Q3: I still recommend writing more clearly and explaining the motivation, including putting Q3 into the body of the text and describing it in more detail (although this is a suggestion that does not affect the rating).***
> >
> > > **A3**: Thank you very much for your helpful suggestion! We will revise the manuscript to clarify the motivation behind Q3 and integrate it more effectively into the main text of the camera ready version.
> >
> > ***Q5: On the other hand, for Q5 I am just a suggestion, maybe the model can adapt to the two modalities of RGBD, and the generated exceptions can correspond to one of the challenges that our field needs to face.***
> >
> > > **A5**: Thank you for the insightful suggestion! We appreciate the idea of extending the model to handle RGBD modalities. In our future work, we will consider this extension in our ongoing research.
> >
> > ***Q6: I may have some objections to Q6, just as you said, "recombining way" to generate exceptions, which may cause some misunderstandings. For example, this way is easy to be misunderstood as the weighted generation of existing exceptions, I hope to explain further in the appendix.***
> >
> > > **A6**: Thank you for your thoughtful comment. We appreciate your attention to this matter. As we mentioned in Lines 241-244, the term "recombining" refers specifically to our Decoupled Anomaly Alignment (DA) loss, in which different tokens exhibit distinct responses in abnormal regions, thereby **capturing different attributes of the anomalies**. By averaging the cross-attention maps corresponding to different anomaly tokens, we are able to generate previously unseen anomalies. A more detailed analysis of the learning process of the DA loss is given in Appendix A.2. Additionally, we will provide a more detailed explanation to further clarify this point in the camera ready version.
> >
> > ***Q7: As other reviewers have said, although you perform well on some anomaly generation metrics, your main task still needs to focus on anomaly detection. Your performance on some anomaly detection metrics is not so impressive. Maybe you need to balance the two points more.***
> >
> > > **A7**: Thank you very much for your valuable feedback! We understand that while our approach shows strong performance in anomaly generation metrics, the primary task remains anomaly detection (AD). As is well known, different metrics capture specific aspects of anomaly detection (AD) performance:
> > >
> > > - Pixel-level metrics provide more detailed evaluations than image-level metrics.
> > > - AUROC is sensitive to false negatives, while AP and F1-max are more affected by false positives.
> >
> > > While existing unsupervised AD methods excel in image-level AUROC (e.g., EfficientAD achieves 99.8 without auxiliary synthetic abnormal data), the main challenge lies in improving pixel-level AP and F1-max, i.e. key metrics in practical industrial applications. For instance, in mobile phone display manufacturing, precise anomaly localization is crucial for processes like demura on MURA defects. To address this, methods like DRAEM emphasize that incorporating synthetic anomalies helps define a decision boundary between normal and anomalous examples. Therefore, segmentation metrics, which assess localization accuracy, are more suitable for evaluating the quality of anomaly generation.
> >
> > > **AD method is a complex system, while SeaS only focuses on generating high-quality anomaly images**. Therefore, we also integrated SeaS with GLASS[1], a SOTA AD method recommended by ***Reviewer EXr6***. For a fair comparison, we replaced the pseudo-anomaly images in GLASS, which are generated by pasting anomalies, with anomaly images generated by SeaS, and keep the other modules unchanged. Tab. R11 shows that GLASS+SeaS outperforms the original GLASS, achieving +2.73% AP, +1.96% F1-max, and +0.31% IoU at the pixel level and slight improvements in image-level AUROC, AP, and F1-max, further proving SeaS’s effectiveness. The performance of GLASS differs from the original paper, as we mentioned in Lines 401-402, we use 2/3 of the anomaly images and all good images from the MVTec AD testing set, while the original results were based on the full test set.
> >
> > > In summary, as shown in Tab. 33 and Tab. R11, combining SeaS with existing SOTA AD menthods (like DRAEM, BGAD, and GLASS) leads to substantial performance gains in anomaly detection task, particularly in pixel-level accuracy, demonstrating its superiority and flexibility over traditional manual anomaly generation techniques.

---

> > > ### Comment · Reviewer_B56D · 2024-12-03
> > >
> > > Your reasons are sound, and I'm glad you took your suggestion to change the text! I have no motivation to raise more doubts, I will keep my score, thank you for your positive response.

---

> > > > ### Author Response · Authors · 2024-12-03
> > > >
> > > > Thank you very much for your approval! Thanks again for your insightful comments and suggestions!

---

### Author Response · Authors · 2024-11-24
**General Response**

***Dear Reviewers and Area Chair:***

Thank you very much for reviewing our paper. The insightful suggestions from all of you are very helpful in making our paper more clear. Below, we provide the overall answers to several common questions. Please do not hesitate to let us know if you have any further questions. We are willing to solve them in time.


***Comparisons on the additional dataset.***

>Thanks for the valuable comments! We have added experimental evaluations on the VisA Dataset, recommended by ***Reviewer uoAo, Reviewer RGGH, and Reviewer EXr6***. We chose it because it contains  more diverse and tiny anomalies, and it is more challenging to generate these tiny anomalies to validate the effectiveness of our method, which is pointed out by ***Reviewer uoAo***. Note that the experimental setting is the same as those in our original paper.

>In terms of compared approaches, since existing state-of-the-art approaches, e.g., DFMGAN, and AnomalyDiffusion, conducted the experimental evaluations only on the MVTec AD dataset, we evaluate them on the VisA Dataset using their official source codes. As shown in Tab. R1 and Fig. 27 of the appendix, SeaS generates anomaly images with higher fidelity and diversity. Tab. R2 shows the comparisons on downstream supervised segmentation trained by the generated images. It consistently demonstrates that our method outperforms others across all the segmentation models, with an **11.71%** average improvement on IoU. We report the image-level metrics in Tab. R3 and our method achieve a **5.92%** gain on image-AUROC. We show the segmentation anomaly maps in Fig. 28 of the appendix, by using our generated image-mask pairs to train BiSeNet V2, there are fewer false positives in *"chewinggum"* and fewer false negatives in *"pcb1"* and *"pipe_fryum"*.

>**Table R1. Comparison of image generation on VisA.**
|     Methods      |    IS    | IC-LPIPS |
| :--------------: | :------: | :------: |
|       SeaS       | **1.27** | **0.26** |
|      DFMGAN      |   1.25   |   0.25   |
| AnomalyDiffusion |   1.26   |   0.25   |


>**Table R2. Comparison of anomaly segmentation trained using generated images on VisA.**
|            |          |       | DFMGAN|       |       |          |       | AnomalyDiffusion |       |       |           |           |   SeaS   |           |           |
| :--------: | :----: | :---: | :----: | :---: | :---: | :--------------: | :---: | :----: | :---: | :---: | :-------: | :-------: | :-------: | :-------: | :-------: |
|            | AUROC  |  AP   | F1-max |  PRO  |  IoU  |      AUROC       |  AP   | F1-max |  PRO  |  IoU  |   AUROC   |    AP     |  F1-max   |    PRO    |    IoU    |
| BiSeNet V2 | 75.91  | 9.17  | 15.00  | 21.49 | 9.66  |      89.29       | 34.16 | 37.93  | 28.09 | 15.93 |   96.03   |   42.80   |   45.41   |   61.29   |   25.93   |
|  UPerNet   | 75.09  | 12.42 | 18.52  | 27.38 | 15.47 |      95.00       | 39.92 | 45.37  | 44.90 | 20.53 |   97.01   |   55.46   |   55.99   |   58.90   |   35.91   |
|    LFD     | 81.21  | 15.14 | 18.70  | 14.98 | 6.44  |      88.00       | 30.86 | 36.56  | 38.56 | 16.61 |   92.91   |   43.87   |   46.46   |   29.55   |   26.37   |
|  Average   | 77.40  | 12.24 | 17.41  | 21.28 | 10.52 |      90.76       | 34.98 | 39.95  | 37.18 | 17.69 | **95.32** | **47.38** | **49.29** | **49.91** | **29.40** |


>**Table R3. Comparison of image-level anomaly detection trained using generated images on VisA.**
|            |       |DFMGAN |        |      |AnomalyDiffusion|    |    |  SeaS    |           |
| :--------: | :----: | :---: | :----: | :--------------: | :---: | :----: | :-------: | :-------: | :-------: |
|            | AUROC  |  AP   | F1-max |      AUROC       |  AP   | F1-max |   AUROC   |    AP     |  F1-max   |
| BiSeNet V2 | 63.07  | 62.63 | 66.48  |      76.11       | 77.74 | 73.13  |   85.61   |   86.64   |   80.49   |
|  UPerNet   | 71.69  | 71.64 | 70.70  |      83.18       | 84.08 | 78.88  |   90.34   |   90.73   |   84.33   |
|    LFD     | 65.38  | 62.25 | 66.59  |      81.97       | 82.36 | 77.35  |   83.07   |   82.88   |   77.24   |
|  Average   | 66.71  | 65.51 | 67.92  |      80.42       | 81.39 | 76.45  | **86.34** | **86.75** | **80.69** |

---

### Meta-Review · Area_Chair_uzvy · 2024-12-22

**Metareview:**

This paper proposes the Separation and Sharing Fine-tuning (SeaS) approach, which uses only a few abnormal and some normal images to handle Few-shot Industrial Anomaly Image Generation. Six reviewers provided mixed feedback on the paper. Following discussions between the reviewers and authors, and after the rebuttal, two reviewers raised their scores—one from 3 to 5, and another from 5 to 6. Ultimately, four reviewers gave positive scores, while two remained negative.

After carefully reviewing the feedback and discussions, the Area Chair (AC) finds that the rebuttal has not fully addressed all the concerns raised. For example:

- The authors claim that combining SeaS with existing state-of-the-art anomaly detection methods (such as DRAEM, BGAD, and GLASS) results in substantial performance gains, but in some cases, these gains are marginal (see Table R11).

- Reviewer uoAo also believes that the paper needs additional experiments to fully validate the effectiveness of the proposed method, and that it is not yet ready for publication at ICLR. Reviewer Exr6 continues to have reservations about this type of work.

- Regarding novelty, the differences between existing methods and the proposed "unbalanced abnormal text prompt" are not adequately explained.

From the extensive and intense discussions, it is clear that several areas of the paper require further improvement. Based on the aforementioned reasons, I am inclined to reject this paper.

**Additional Comments On Reviewer Discussion:**

Six reviewers comment on this paper. The main concerns are Segmentation Metrics for Evaluating Anomaly Generation Quality, Results on More Datasets, and the Help of SOTA  Anomaly Detection Methods. After the rebuttal phase, most concerns are addressed, and some reviewers raise their scores; however, the final scores are still mixed. The AC carefully checked the discussions and rebuttals and rejected this paper.

---

### Decision · Program_Chairs · 2025-01-22

Reject